Manuscript prepared for Geosci. Model Dev.
with version 2014/09/16 7.15 Copernicus papers of the LATEX class copernicus.cls.
Date: 5 March 2021

# JULES-CN: a coupled terrestrial Carbon-Nitrogen Scheme (JULES vn5.1)

Andrew J. Wiltshire[1,2], Eleanor J. Burke[1], Sarah E. Chadburn[3], Chris D. Jones[1], Peter M. Cox[3], Taraka Davies-Barnard[3], Pierre Friedlingstein[3], Anna B. Harper[3], Spencer Liddicoat[1], Stephen Sitch[2], and Sönke Zaehle[4]

[1]Met Office Hadley Centre, Exeter, Devon, UK EX1 3PB
[2]College of Life and Environmental Sciences, University of Exeter, Exeter, EX4 4RJ
[3]College of Engineering, Mathematics, and Physical Sciences, University of Exeter, Exeter, EX4 4QE
[4]Biogeochemical Signals Department, Max Planck Institute for Biogeochemistry, 07745 Jena, Germany

*Correspondence to:* Andrew Wiltshire (andy.wiltshire@metoffice.gov.uk)

**Abstract.** Understanding future changes in the terrestrial carbon cycle is important for reliable projections of climate change and impacts on ecosystems. It is well known that nitrogen (N) could limit plants' response to increased atmospheric carbon dioxide and is therefore important to include a representation of the N cycle in Earth System Models. Here we present the implementation of the terrestrial nitrogen cycle in the Joint UK Land Environment Simulator (JULES) - the land surface scheme of the UK Earth System Model (UKESM). Two configurations are discussed - the first one (JULES-CN) has a bulk soil biogeochemical model and the second one is a development configuration which resolves the soil biogeochemistry with depth (JULES-CN$_{layer}$). In JULES the nitrogen (N) cycle is based on the existing carbon (C) cycle and represents all the key terrestrial N processes in a parsimonious way. Biological N fixation is dependent on net primary productivity, and N deposition is specified as an external input. Nitrogen leaves the vegetation and soil system via leaching and a bulk gas loss term. Nutrient limitation reduces carbon-use efficiency (CUE - ratio of net to gross primary productivity) and can slow soil decomposition. We show that ecosystem level N limitation of net primary productivity (quantified in the model by the ratio of the potential amount of C that can be allocated to growth and spreading of the vegetation compared with the actual amount achieved in its natural state) falls at the lower end of the observational estimates in forests (approximately 1.0 in the model compared with 1.01 to 1.38 in the observations). The model shows more N limitation in the tropical savanna and tundra biomes consistent with the available observations. Simulated C and N pools and fluxes are comparable to the limited available observations and model derived estimates. The introduction of a N cycle improves the representation of interannual variability of global net ecosystem exchange which was more pronounced in the C cycle only versions of JULES

(JULES-C) than shown in estimates from the Global Carbon Project. It also reduces the present-day CUE from a global mean value of 0.45 for JULES-C to 0.41 for JULES-CN and 0.40 for JULES-$CN_{layer}$ all of which fall within the observational range. The N cycle also alters the response of the

C fluxes over the twentieth century and limits the $CO_2$-fertilisation effect, such that the simulated current-day land C sink is reduced by about 0.5 Pg C $yr^{-1}$ compared to the version with no N limitation. JULES-$CN_{layer}$ additionally improves the representation of soil biogeochemistry including turnover times in the northern high latitudes. The inclusion of a prognostic land N scheme marks a step forward in functionality and realism for the JULES and UKESM models.

**1    Introduction**

Terrestrial ecosystems absorb around 25% of anthropogenic carbon emissions (Le Quéré et al., 2018; Friedlingstein et al., 2020), and changes in the future land carbon (C) sink will feedback to climate via the proportion of the emissions remaining in the atmosphere. Under projected climate change, the primary mechanism for increased terrestrial sequestration is an increase in plant productivity and

biomass, which relies on sufficient availability of nitrogen (N) within the soil-plant system. Therefore the availability of N impacts the land C sink, both in the present and in a higher atmospheric carbon dioxide ($CO_2$) future.

Nitrogen exists in the terrestrial system in organic and inorganic forms and is continually cycled.

In a stable climate the external inputs to the coupled vegetation and soil system–biological N fixation and N deposition–are balanced by the losses from this system–N leaching and N gas loss associated primarily with denitrification processes. Depending on the nutrient status of the vegetation and soil, changes in the balance of the inputs and outputs of N can drive adjustments in vegetation biomass and soil organic matter. Within the system organic N is transferred from the vegetation to the soil

through the production of litter and disturbance. The litter decomposes into soil organic matter and in turn is mineralised into inorganic N. Both inorganic and organic N may become available for plant uptake, although the amount of organic N uptake by plants is small and typically not included in models (Weintraub and Schimel, 2005).

Any increase in atmospheric $CO_2$ drives an increase in the land C uptake and hence an increase in the gross primary productivity (GPP). This results in an extra demand for N which could potentially limit the increase in future C stocks. For example, Zaehle (2013a) suggest that, in some areas, N could limit any future C sink by up to 70%. N cycling also tends to reduce the sensitivity of land C uptake to temperature. Warmer conditions lead to increased plant respiration and soil respiration,

which tends to reduce the land C sink. However, the increased soil respiration also leads to accelerated N mineralisation and increased N availability to plants, which may provide a counteracting

increase in GPP. This latter effect is absent from models that do not include a N cycle. As a result of neglecting these important effects, land-surface models without an interactive N cycle tend to overestimate both $CO_2$ and temperature effects on the land C sink (Wenzel et al., 2016; Cox et al., 2013). In addition, climate projections assessed by IPCC using CMIP5 Earth System Models that lacked terrestrial carbon cycle Ciais et al. (2014) have been shown to exhibit a major and systematic bias in their future projection of land carbon sink Zaehle et al. (2015); Wieder et al. (2015b). An increasing number of land surface and climate models now include constraints on the land C sink caused by N limitation (Zaehle et al., 2014; Wania et al., 2012; Smith et al., 2014). In fact, recent simulations have generated a range of estimates for the sensitivity of the C cycle to N availability (Meyerholt et al., 2020a; Davies-Barnard et al., 2020; Arora et al., 2019). For example, Meyerholt et al. (2020a) used a perturbed model ensemble to show that N limitation reduces both the projected future increase in land C store due to $CO_2$ fertilisation and the projected loss in land C caused by climate change. The inclusion of nitrogen cycle processes in many CMIP6 models has been a major advance Arora et al. (2019). Jones and Friedlingstein (2020) show how CMIP6 models have a much reduced spread in their simulation of airborne fraction than in CMIP5 and this is attributable to the inclusion of N-cycle in about half of these latest generation models. But process understanding and evaluation of these model is still in its infancy (Davies-Barnard et al., 2020).

The purpose of this paper is to describe and evaluate the implementation of coupled C and N cycles within the Joint UK Land-Environment Simulator (Best et al., 2011; Clark et al., 2011) (JULES at vn5.1 - http://jules-lsm.github.io/vn5.1/release_notes/JULES5.1.html). JULES is the land surface component of the UK Earth System Model (UKESM) (Sellar et al., 2019). The addition of the N cycle to JULES described in this paper was carried out alongside other developments such as improved plant physiology and an extended number of plant functional types (Harper et al., 2018), an enhanced representation of surface exchange and hydrology (Wiltshire et al., 2020) and a new module for land management (Robertson et al., in prep.). These separate components combine to make the land surface component of UKESM and were used for the most recent Global Carbon Budget annual assessment (Friedlingstein et al., 2020).

The philosophy behind the developments described here is to produce a parsimonious model to capture the established first order emergent response of N addition on growth which translates into leaf area index (LAI) and biomass. Our approach is to simulate the large-scale role of N limitation on vegetation C use efficiency (CUE - ratio of net to gross primary productivity) and soil C turnover. This is achieved by extending the implicit representation of N in the existing dynamic vegetation and plant physiology modules to be fully interactive. At the core of surface exchange in JULES is a coupled stomatal conductance photosynthesis scheme with a dependency on the leaf N concentration. Similarly, plant maintenance respiration has a dependency on leaf, root and stem N concentration

(Cox et al., 1998, 1999; Cox, 2001; Clark et al., 2011). Implicit within JULES, even in simulations excluding the N cycle is the parameterisation of plant tissue level N concentrations and associated allometry (discussed further in Section 3.1.3 and by Wiltshire et al. (2020)). Simulations with an interactive C cycle therefore assume that enough N is available to meet vegetation growth and turnover. Here, we simply limit growth if not enough N is available. To do this requires a full representation of the N cycle in the land surface including a coupled soil C and N organic and soil inorganic N scheme.

At the ecosystem level, the C and N cycles are closely coupled with each flux of C associated with a corresponding flux of N linked via the C to N ratios. In JULES nutrient limitation operates through two mechanisms. Firstly, the vegetation cannot uptake as much C – any C that the plants cannot uptake is denoted excess C. Secondly the decomposition of litter C is slowed because there is insufficient N present. This is achieved by explicitly representing the demand for N within the vegetation and soil modules and then reducing plant net primary productivity to match available nutrients. In the soil module an additional decomposition rate modifier is introduced that slows respiration by microbes to match available nutrients. The current structure of the TRIFFID dynamic vegetation model (Cox, 2001), in particular the fixed allometry and C allocation, is largely unchanged. As the aim of this scheme is to capture the impact on terrestrial C stores, N loss terms are aggregated and not speciated. The model's reduced uptake of vegetation C due to N limitation is designed to have only a minor impact on the GPP. The emergent impact of the N scheme is modelled by reducing NPP and hence the carbon use efficiency (CUE) of the vegetation. In reality the C the plants are unable to use because of insufficient N (defined as $\Psi$) becomes to non structural carbohydrates, root exudates or biogenic volatile organic compounds (Collalti and Prentice, 2019). However, to simplify the carbon balance in JULES-CN, it is added to the autotrophic respiration.

A key assumption in the JULES representation of vegetation, and common amongst complex DGVMs (Meyerholt and Zaehle, 2015) is of fixed plant stoichiometry (mass ratio of C to N atoms or C:N ratio). The implication is that leaf-level photosynthetic capacity does not vary with available N. This is consistent with field experiments enhancing N fertilisation that find increases in growth but no corresponding change in photosynthetic capacity (Brix and Ebell, 1969; Wang et al., 2012; Field and Mooney, 1986; McGuire et al., 1995). However, more recent analyses do make the link between nutrient availability and leaf level N concentrations (e.g. Mao et al. (2020)). In general, models make different assumptions about the tightness of the coupling mechanism between the C and N cycles leading to substantial uncertainty in their projections (Zaehle and Dalmonech, 2011). Within the fully coupled Earth Systems Models used in the Coupled Climate Carbon Cycle Model Intercomparison Project (C4MIP) for quantifying C feedbacks only four out of eleven models include a N cycle representation and only two include both N and dynamic vegetation of which JULES is one of them (Arora et al., 2020). The representation of the N cycle in the full complexity Earth System

Models remains challenging and there is clearly a need for simple models capturing the first order responses. This is the first time a N cycle has been incorporated in JULES and it is expected to be improved and developed with time as the knowledge of how important processes can be represented in existing frameworks improves.


## 2 Introduction to JULES

JULES is the land surface component of the new UK community Earth System model, UKESM (Sellar et al., 2019). JULES can also be run offline forced by observed meteorology globally, region-ally or at a single location. A full description of the main components of JULES is provided by Best

et al. (2011) and Clark et al. (2011). In particular, JULES represents the surface energy balance, a dynamic snowpack model (one dimensional), vertical heat and water fluxes, soil freezing, large scale hydrology, and C fluxes and C storage in both vegetation and soil. Typically JULES represents four soil layers down to a total depth of 3m. Within JULES, C stocks and fluxes in and between the soil and vegetation along with competition between different vegetation types are modelled by the

Top-Down Representation of Interactive Foliage and Flora Including Dynamics (TRIFFID) (Cox, 2001). In this version of TRIFFID, five plant functional types (PFTs) are represented: broadleaf trees, needleleaf trees, $C_3$ grasses, $C_4$ grasses and shrubs. The soil C model is based on the RothC model (Clark et al., 2011). Recently, Burke et al. (2017) and Chadburn et al. (2015) added a represen-tation of permafrost soil processes to JULES, including a representation of the vertical distribution

of soil organic C which we build upon here. JULES-C is the standard carbon cycle configuration (a configuration defines a specific set of switches and parameters) and was used in the Global Carbon Budget annual assessment in 2018 (Le Quéré et al., 2018).

What follows is a description of the extension of the C cycle process modelled by the JULES-

C configuration to include an interactive N cycle. This results in two new model configurations: JULES-CN and JULES-CN$_{layer}$. The soil biogeochemistry is represented by a single bulk layer in JULES-CN whereas it varies as a function of depth in JULES-CN$_{layer}$. As standard JULES-C in-cludes an implicit representation of N which has been extended to be fully interactive. The N cycle processes are added to the TRIFFID dynamic vegetation and RothC soil C models. For clarity we

include here a full description of the C and N cycle including the existing TRIFFID and RothC mod-els and highlight where and how their processes have been modified.

## 3 JULES developments

### 3.1 Vegetation C and N

The TRIFFID dynamic vegetation model provides the core of the vegetation module (Cox, 2001). TRIFFID represents the vegetation cover at each location in terms of the fractional area covered, and the leaf area index (LAI) and canopy height of each plant functional type (PFT). In JULES the C fluxes are calculated at the model timestep (typically 0.5 - 1 hour) prior to any N limitation (if configured). These fluxes are then aggregated to the timestep required for running TRIFFID (once every

10 days in the current implementation) so that allocation of C can take place. TRIFFID employs fixed allometry such that the split of vegetation carbon between leaf, root and stem is defined by a single state prognostic variable that defines the total biomass. Biomass density increases via growth and is reduced by litter production and competition with other PFTs (Clark et al., 2011). Biomass can also increase by spreading through an increase in covered area. N limitation reduces growth and

spreading such that the change in vegetation N cannot exceed the N uptake rate.

This section documents the vegetation model starting with the structure of the vegetation (Section 3.1.1) including the additional complexity of labile N (Section 3.1.2). The following subsection describes how growth and spreading is limited by N availability (Section 3.1.3). The final subsection

describes how vegetation C and N is turned over by disturbance and competition and aggregated from PFTs to the gridcell level (Section 3.1.4). Biological N fixation is input directly into the soil inorganic N pool and is described later in Section 3.3.1.

#### 3.1.1 Vegetation Structure

availThe mean canopy height per PFT $i$ is converted via allometric equations into a maximum or balanced leaf area index for each PFT ($\mathcal{L}_{b,i}$ in $m^2 m^{-2}$). $\mathcal{L}_{b,i}$ is the prognostic variable used in JULES to describe the vegetation and is functionally the equivalent of the potential leaf area. Given $\mathcal{L}_{b,i}$, leaf, root and wood pools are diagnosed for each PFT as introduced in Cox (2001). The balanced leaf area index is updated interactively following the C balance and is coupled to the surface exchange

via surface albedo, roughness and heat capacity. This section is included to fully document the new scheme, but the equations can also be found in Clark et al. (2011).

The vegetation C density per PFT ($C_{v,i}$ in $kg\,[C]\,m^{-2}$) can be separated into leaf ($L_{c,i}$ in $kg\,[C]\,m^{-2}$), fine root ($R_{c,i}$ in $kg\,[C]\,m^{-2}$) and total stem plus coarse root ($W_{c,i}$ in $kg\,[C]\,m^{-2}$) pools, each of

which is related allometrically to the balanced leaf area ($\mathcal{L}_{b,i}$). Each component is then related to $\mathcal{L}_{b,i}$. Root C is set equal to leaf C, which is itself a linear function of $\mathcal{L}_{b,i}$, and total stem C is related

to $\mathcal{L}_{b,i}$ by a power law (Enquist et al., 1998):

$$C_{v,i} = L_{c,i} + R_{c,i} + W_{c,i} \tag{1}$$

$\quad L_{c,i} = \sigma_{l,i} \mathcal{L}_{b,i} \tag{2}$

$$R_{c,i} = L_{c,i} \tag{3}$$

$$W_{c,i} = a_{wl,i} (\mathcal{L}_{b,i})^{b_{wl,i}} \tag{4}$$

$\quad$ Where $\sigma_{l,i}$ (kg [C] m$^{-2}$), $a_{wl,i}$ (kg [C] m$^{-2}$) and $b_{wl,i}$ (dimensionless) are PFT dependent allo-
metric parameters defined in Table 1. By definition $\mathcal{L}_{b,i}$ does not have an explicit seasonal cycle but
responds to changes in the vegetation C on both short (seasonal) and long (centennial) timescales. A
high $\mathcal{L}_{b,i}$ is related to a high C density and tall canopies. It should be noted that leaf seasonality is
represented by a separate phenology model and is not directly affected by N availability. TRIFFID
combines Equation 4 with a "pipe model" approach (Shinozaki et al., 1964a, b) to obtain the canopy
height for PFT $i$ ($h_i$ in m):

$$h_i = \frac{W_{c,i}}{a_{ws,i}\eta_{sl,i}} \left( \frac{a_{wl,i}}{W_{c,i}} \right)^{1/b_{wl,i}} \tag{5}$$

where $\eta_{sl,i}$ (kg [C] m$^{-2}$ per unit LAI) relates respiring stem to leaf C (Table 1) and $a_{ws,i}$ is the
ratio of total stem C to respiring stem C. We can combine Equations 4 and 5 to relate ($\mathcal{L}_{b,i}$) to
canopy height ($h_i$) and these two variables can be used interchangeably to describe the state of the
vegetation. During a simulation the C pools are updated interactively and the canopy height and
balanced leaf area diagnosed for each PFT. This representation allows changes in vegetation C to
feedback to surface exchange.

$\quad$ The root and total stem N pools are defined using stoichiometric relationships as a function of the
C pools. These stoichiometric functions already exist in the model and are used in the calculation of
plant maintenance respiration (Clark et al., 2011). We extend their use to explicitly define N pools
as part of the new scheme:

$$R_{n,i} = \mu_{rl,i} \, n_{l0,i} \, R_{c,i} \tag{6}$$


$$W_{n,i} = \mu_{sl,i} \, n_{l0,i} \, W_{c,i} \tag{7}$$

**Table 1.** Default values of PFT-specific parameters for allometry, allocation and vegetation N and C stoichiometry in the JULES-CN and JULES-CN$_{layer}$ configurations. The subscript ($i$) is present to show that it is a PFT-specific value. $n_{l0,i}$ is the N concentration at the top of the canopy but is shown here as $1/n_{l0,i}$ so that it is comparable to expected C:N ratios from the literature.

| Symbol (units) | Definition | Broadleaf tree | Needleleaf tree | C$_3$ grass | C$_4$ grass | Shrub |
|---|---|---|---|---|---|---|
| $\sigma_{l,i}$ (kg [C] m$^{-2}$) | Specific density of leaf C | 0.0375 | 0.1000 | 0.0250 | 0.0500 | 0.0500 |
| $a_{wl,i}$ (kg [C] m$^{-2}$) | Allometric coefficient | 0.65 | 0.65 | 0.005 | 0.005 | 0.10 |
| $a_{ws,i}$ (-) | Ratio total C to respiring stem C | 10.0 | 10.0 | 1.0 | 1.0 | 10.0 |
| $b_{wl,i}$ (-) | Allometric exponent | 1.667 | 1.667 | 1.667 | 1.667 | 1.667 |
| $\eta_{sl,i}$ (kg [C] m$^{-2}$ per unit LAI) | Live stemwood coefficient | 0.01 | 0.01 | 0.01 | 0.01 | 0.01 |
| $\mu_{rl,i}$ (-) | Ratio root N to top leaf N | 1.0 | 1.0 | 1.0 | 1.0 | 1.0 |
| $\mu_{sl,i}$ (-) | Ratio stem N to top leaf N | 0.1 | 0.1 | 1.0 | 1.0 | 0.1 |
| $1/n_{l0,i}$ ((kg [C])(kg [N])$^{-1}$) | C:N ratio at canopy top | 21.7 | 30.3 | 13.7 | 16.67 | 16.67 |
| $k_{n,i}$ (-) | N profile coefficient | 0.78 | 0.78 | 0.78 | 0.78 | 0.78 |
| $\kappa_{r,i}$ (-) | Root N retranslocation coef. | 0.2 | 0.2 | 0.2 | 0.2 | 0.2 |
| $\kappa_{l,i}$ (-) | Leaf N retranslocation coef. | 0.5 | 0.5 | 0.5 | 0.5 | 0.5 |
| $\mathcal{L}_{min,i}$ (-) | Minimum balanced LAI | 3.0 | 3.0 | 1.0 | 1.0 | 1.0 |
| $\mathcal{L}_{max,i}$ (-) | Maximum balanced LAI | 9.0 | 9.0 | 4.0 | 4.0 | 4.0 |
| $f_{DPM,i}$ (-) | Decomposable litter fraction | 0.25 | 0.25 | 0.67 | 0.67 | 0.33 |

where $\mu_{rl,i}$ and $\mu_{sl,i}$ are dimensionless stoichiometric parameters linking the top leaf N concentration ($n_{l0,i}$ in kg [N] kg [C]$^{-1}$) to the total stem and root N concentration via $n_{l0,i}$. The leaf N pool ($L_{n,i}$ in kg [N] m$^{-2}$) has an additional dependency on phenological state (Section 3.1.2) and assumed distribution of N in the canopy. Following Equation 1 the total vegetation N store per PFT ($N_{v,i}$ in kg [N] m$^{-2}$) is given by:

$$N_{v,i} = L_{n,i} + R_{n,i} + W_{n,i} \tag{8}$$

The C:N ratio of the root and stem pools are fixed in time and leaf pool C:N ratio only varies with phenological state. However, the relative proportions of each pool vary with total biomass resulting in the whole plant C:N ratio increasing with total vegetation C for woody PFTs (Figure 1). This is due to the relatively greater proportion of stem C at higher biomass. Grasses show less variation with biomass due to their comparatively small amount of structural C relative to leaf area, which also results in woody PFTs having higher C:N ratios. Equations 1-8 show that the total vegetation N increases with canopy height and biomass (Figure 2).

### 3.1.2 Labile C and N: Phenology and Mobilisation

The total leaf C pool per PFT ($L_{c,i}$, Equation 2) varies allometrically with the vegetation C state on both short (seasonal) and long (centennial) timescales but not with changes in phenological state. Implicit within TRIFFID is a labile leaf C pool that acts as a reserve of C during spring and a store during fall. $L_{c,i}$ therefore includes a labile pool from which C can be mobilised during leaf out plus an allocated pool representing the actual LAI. The labile pool is zero at full leaf out and at the allometrically defined maximum during the no leaf period. As part of the N coupling we introduce the ability for plants to retranslocate some of the allocated N to the labile N pool according to the phenology. The new parameterisation of retranslocation and labile N is therefore dependent on the leaf phenological state as well as the fixed stoichometry. In JULES, leaf phenology is controlled by a second state variable ($p_i$) which relates the LAI ($\mathcal{L}_i$) at any moment in time to the balanced leaf area index ($\mathcal{L}_{b,i}$).

$$\mathcal{L}_i = p\mathcal{L}_{b,i} \tag{9}$$

where $p_i$ is a scalar between 0 and 1 that describes the phenological state of the system (Clark et al., 2011). For evergreen plants $p_i$ is a constant of 1. The two state variables $\mathcal{L}_{b,i}$ and $p_i$ combine to define the vegetation phenological state for each PFT $i$. Using the phenological state we extend the equivalent approach to leaf C such that the leaf N pool ($L_{n,i}$) has fixed allometry dependent on the phenological state and the magnitude of leaf retranslocation. We introduce this simple parameterisation under the assumption that higher leaf retranslocation during autumn implies a higher labile N store. The leaf N pool therefore becomes:

$$L_{n,i} = p_i n_{lc,i} L_{c,i} + (1 - p_i)(\frac{1 + \kappa_{l,i}}{2}) n_{lc,i} L_{c,i} \tag{10}$$

where $\kappa_{l,i}$ is the dimensionless leaf N retranslocation coefficient and $n_{lc,i}$ is the mean canopy N concentration (defined in Equation 11 in kg [N] kg [C] $^{-1}$). Here $\kappa_{l,i}$ is set to 0.5 for all PFTs (Zaehle and Friend, 2010). The formulation of the labile pool, in this configuration, means that around half of the N required for full leaf-out is taken from leaf retranslocation with a further quarter acquired during the dormant phase while the rest is acquired during the leaf-out period.

JULES assumes a process-based scaling-up of leaf level photosynthesis to the the canopy level. In both the JULES-CN and JULES-CN$_{layer}$ configurations, to be consistent with the JULES-C model, we assume a multi-level canopy with leaf N decreasing exponentially through the canopy (*CanRad-Mod 5*). The plant physiology routines uses this assumed distribution to calculate penetration through the canopy and photosynthesis on individual layers before scaling back to the canopy (Clark et al., 2011). In the application here, we use this distribution to be fully consistent with the physiology. The vertical distribution of leaf N concentration ($n_{lc,i}(d)$ in kg [N] kg [C] $^{-1}$) in the canopy is described

by (Mercado et al., 2007):

$$n_{lc,i}(d) = n_{l0,i}e^{-k_{n,i}d} \tag{11}$$

where $k_{n,i}$ is a constant representing the profile of N density and $d$ represents the fraction of canopy above the layer. Based on observed N profiles in the Amazon basin (Carswell et al., 2000), a value of 0.78 for $k_{n,i}$ was found (Mercado et al., 2007). Equation 11 is independent of leaf area and therefore equates to a constant of proportionality relating PFT-specific top leaf N to the mean canopy N concentration.

### 3.1.3 Vegetation Growth and Allocation

The previous section describe how the vegetation C ($C_{v,i}$, Equation 1) and vegetation N ($N_{v,i}$, Equation 8) for each PFT vary with vegetation size and phenological state. This section describes how growth and spreading are limited by available N. Growth is the increase in C density and spreading is the increase in vegetation cover from recruitment and reproduction.

Net Primary Productivity (NPP) in JULES-C is simply the difference between GPP and autotrophic respiration ($R_a$). In JULES-CN the potential NPP or $NPP_{pot}$ is defined in the same way as the NPP in JULES-C before the explicit N cycle was included, i.e. the potential amount of C that can be allocated to growth ($g$) and spreading by TRIFFID. In JULES-CN and in order for the NPP to achieve its potential it needs to be able to uptake sufficient inorganic N. If not enough inorganic N is available, the system is N limited and an additional term is required in the C balance representing C which cannot be assimilated into the plant due to lack of available N ($\Psi$ in kg [C] m$^{-2}$). A positive $\Psi$ results in a reduction of carbon use efficiency (CUE).

The C balance per PFT $i$ is given by:

$$\frac{dC_{v,i}}{dt} = (1 - \lambda_i)\Pi_i - \Lambda_{c,i} - \Psi_{g,i} \tag{12}$$

where $\Pi_i$ is the potential NPP per unit area of PFT in (kg [C] m$^{-2}$ s$^{-1}$ (prior to nutrient limitation) and $\Lambda_{c,i}$ (kg [C] m$^{-2}$ s$^{-1}$) is the PFT specific litterfall rate (Section 3.1.4). Any excess C from growth ($\Psi_{g,i}$) is considered an additional plant respiration term and at the end of the TRIFFID timestep is used to reduce the potential NPP for each PFT to its actual value. $\lambda_i$ is the coefficient for partitioning the NPP between growth and spreading. $\lambda_i$ is utilised in increasing the fractional coverage of the vegetation and $(1-\lambda_i)$ increases the C of the existing vegetated area. $\lambda_i$ is a function of the vegetation C which itself is a function of the balanced LAI for PFT $i$ ($\mathcal{L}_{b,i}$):

$$\lambda_i = \begin{cases} 1 & \mathcal{L}_{b,i} > \mathcal{L}_{max,i} \\ \frac{\mathcal{L}_{b,i} - \mathcal{L}_{min,i}}{\mathcal{L}_{max,i} - \mathcal{L}_{min,i}} & \mathcal{L}_{min,i} < \mathcal{L}_{b,i} \leq \mathcal{L}_{max,i} \\ 0 & \mathcal{L}_{b,i} \leq \mathcal{L}_{min,i} \end{cases} \tag{13}$$

The equivalent N balance per PFT is given by:

$$\frac{dN_{v,i}}{dt} = (1 - \lambda_i)\Phi_i - \Lambda_{n,i} \tag{14}$$

where $\Phi_i$ (kg [N] m$^{-2}$ s$^{-1}$) is the PFT specific N uptake (see Equation 19) and $(1-\lambda_i)\Phi_i$ is equal to $\Phi_{g,i}$, the N uptake available for growth. $\Lambda_{n,i}$ is the PFT N litter flux after taking into account the retranslocation of N from leaves or roots. The N available for spreading is a fraction $\lambda_i$ of the total available N with a fraction $(1 - \lambda_i)$ available for growth.

Litter is produced by the turnover of the leaf, wood and root pools for each PFT, defined as

$$\Lambda_{c,i} = \gamma_{l,i}L_{c,i} + \gamma_{r,i}R_{c,i} + \gamma_{w,i}W_{c,i} \tag{15}$$

and

$$\Lambda_{n,i} = (1 - \kappa_{l,i})\gamma_{l,i}L_{n,i} + (1 - \kappa_{r,i})\gamma_{r,i}R_{n,i} + \gamma_{w,i}W_{n,i} \tag{16}$$

for litter C ($\Lambda_{c,i}$ in kg [C] m$^{-2}$ s$^{-1}$) and litter N ($\Lambda_{n,i}$ in kg [N] m$^{-2}$ s$^{-1}$) respectively. $\gamma_{r,i}$ and $\gamma_{w,i}$ are turnover rates in $s^{-1}$ (Table 6 of Clark et al. (2011)). The leaf turnover rate ($\gamma_{l,i}$) is a temperature dependent turnover rate consistent with the phenological state and defined in Clark et al. (2011). The equivalent term for N allows for retranslocation of N from leaves into the labile store and a reduced N cost of maintaining fine roots. $\kappa_{l,i}$ and $\kappa_{r,i}$ are the dimensionless coefficients for the retranslocation of leaf and root N shown in Table 1 (Zaehle and Friend, 2010).

In JULES-CN the N available for plant uptake for each PFT $i$ ($N_{avail,i}$ in $kg\,[N]\,m^{-2}$) is the the inorganic soil N pool ($N_{in}$ in $kg\,[N]\,m^{-2}$) split equitably between the PFTs assuming there is no differential ability between PFTs to acquire N and the whole pool is available for uptake during the model timestep. The available N in JULES-CN$_{layer}$ is more complicated and takes into account the soil profile. This is discussed in Section 3.3.2.

Equations 12 and 14 have two remaining unknowns for each PFT: the plant N uptake for growth ($\Phi_{g,i}$) and the excess C from growth ($\Psi_{g,i}$). The litter fluxes are functions of the total vegetation pool and therefore can be solved at the same time. Solving for the case where $\Psi_{g,i} = 0.0$ gives the total vegetation N demand for growth. If the N demand is less than the available N in a given timestep ($\Delta t$) ($\Phi_{g,i} < (1-\lambda_i)\,N_{avail,i}/\Delta t$) then growth is unlimited and the fluxes can be updated accordingly. Where N is limiting, growth N uptake is set equal to the available N ($\Phi_{g,i} = (1-\lambda_i)$ $N_{avail,i}/\Delta t$) and the excess C for growth $\Psi_{g,i}$ can be derived. Following the solution of $\frac{dN_{v,i}}{dt}$ the C store and balanced LAI ($\mathcal{L}_{b,i}$) are updated and the leaf, root and wood pools for each PFT can be derived following the allometric equations (Equations 2-4).

The remaining proportion ($\lambda_i$) of NPP and N is allocated to spreading. The N demand for spreading is equal to the C allocated to spreading scaled by the whole plant stoichiometry:

$$\Phi_{s,i} = \frac{N_{v,i}}{C_{v,i}} \left( \Pi_i - \frac{dC_{v,i}}{dt} - \Psi_{s,i} \right) \tag{17}$$

where $\Psi_{s,i}$ (or $\lambda_i \Psi_i$) is the excess C term from spreading and $\frac{N_{v,i}}{C_{v,i}}$ is the inverse of the the whole plant C:N ratio. As with growth limitation, Equation 17 is first solved to find the N demand for spreading ($\Psi_{s,i} = 0.0$). If the arising demand is less than the available N ($\Phi_{s,i} < \lambda_i \, N_{avail,i}/\Delta t$) spreading is unlimited. If N demand is in excess of that available, the uptake is set equal to the available N flux ($\Phi_{s,i} = \lambda_i \, N_{avail,i}/\Delta t$) and the excess ($\Psi_{s,i}$) assimilate solved for.

Total excess C per PFT $i$ ($\Psi_i$) is therefore the combination of that from growth plus spreading:

$$\Psi_i = \Psi_{s,i} + \Psi_{g,i} \tag{18}$$

Similarly total N uptake per PFT $i$ ($\Phi_i$) is therefore the combination of N uptake used for growth plus N uptake used for spreading:

$$\Phi_i = \Phi_{s,i} + \Phi_{g,i} \tag{19}$$

The PFT level N uptake and excess C are weighted by the fraction of coverage of each PFT in a grid cell ($v_i$) and summed to get the grid averaged values:

$$\Phi = \sum_i v_i \Phi_i \tag{20}$$

$$\Psi = \sum_i v_i \Psi_i \tag{21}$$

This excess C ($\Psi$) is considered an additional plant respiration term and at the end of the TRIFFID timestep is used to reduce the potential NPP to its actual value.

The C and N allocated to spreading allow the vegetation to expand onto bare ground. Where space is limiting the PFTs compete for space. The competition is handled in the Lotka-Volterra competition routines (see Clark et al. (2011) for full details). N only indirectly affects competition through the PFT specific allometric relationships. The competition code subsequently updates the fractional coverage of model PFTs ($v_i$).

### 3.1.4 Vegetation Turnover and Total Litter Production

The previous sections describe how N interacts to limit both growth and spreading of vegetation in the dynamic vegetation model. This final section describes the turnover of C and N through large-scale disturbance and competition.

Turnover is aggregated across PFTs to provide a grid box mean litter flux term to the soil biogeo-chemistry processes which acts at a grid box level. Total litter C ($\Lambda_c$, kg [C] m$^{-2}$ s$^{-1}$) is made-up of the area-weighted sum of the litter C from each PFT ($\Lambda_{c,i}$), along with large-scale PFT-dependent disturbance rate, and a density dependent component from intra-PFT competition for space. Large-scale disturbance is implemented in TRIFFID as a constant disturbance rate per PFT and captures processes such as wind-throw and other mortality events. Density dependent litter production arises through competition for space with increased turnover when space is limiting and plants are competing for space and light.

$$\Lambda_c = \sum_i v_i \left( \Lambda_{c,i} + \gamma_{v,i} C_{v,i} + (\Pi_i - \Psi_i) \sum_j c_{i,j} v_j \right) \tag{22}$$

where $c_{i,j}$ are the competition coefficents describing the effect of PFT $i$ on PFT $j$, $\gamma_{v,i}$ is a large scale disturbance term of PFT $i$, $v_i$ is the vegetation fraction of PFT $i$, $\Pi_i$ is defined in Equation 12 and $\Psi_i$ in Equation 18. The effect of N limitation on the litter C flux is captured in the excess C term per PFT ($\Psi_i$). Similarly to the total litter C, total litter N ($\Lambda_n$, kg [N] m$^{-2}$ s$^{-1}$) is given by:

$$\Lambda_n = \sum_i v_i \left( \Lambda_{n,i} + \gamma_{v,i} N_{v,i} + \Phi_i \sum_j c_{i,j} v_j \right) \tag{23}$$

Both $\Lambda_c$ and $\Lambda_n$ vary according to the vegetation type and the relative amount of stem, leaves and roots being turned over. This means that the C:N ratio also varies in time and space.

## 3.2 Soil Biogeochemistry

Here we describe the addition of a prognostic soil N model for JULES-CN that extends the Roth-C soil C model used in JULES-C (Jenkinson et al., 1990; Jenkinson and Coleman, 1999).

The original Roth-C soil C model represents four C pools ($p$) for each grid box. Plant litter input is split between two C pools of decomposable ($DPM$) and resistant ($RPM$) plant material, with the fraction that goes to each depending on the overlying vegetation PFT and parameterised via $f_{DPM,i}$. Grasses provide a higher fraction of decomposable litter input and trees provide a higher fraction of resistant litter input. The other two C pools are microbial biomass ($BIO$) and long-lived humified ($HUM$) pools. The $DPM$ and $RPM$ pools can be characterised as representing litter and $BIO$ and $HUM$ as representing soil organic matter. C from decomposition of all of the pools is partly released to the atmosphere, and the remaining fraction ($\beta_R$) enters the $BIO$ and $HUM$ pools. The C pools are updated according to:

$$\frac{dC_{DPM}}{dt} = \sum_i (v_i f_{DPM,i} \Lambda_{c,i}) - R_{DPM} \tag{24}$$

$$\frac{dC_{RPM}}{dt} = \sum_i \left(v_i(1 - f_{DPM,i})\Lambda_{c,i}\right) - R_{RPM} \tag{25}$$

$$\frac{dC_{BIO}}{dt} = 0.46\beta_R R_{tot} - R_{BIO} \tag{26}$$

$$\frac{dC_{HUM}}{dt} = 0.54\beta_R R_{tot} - R_{HUM} \tag{27}$$

where $t$ is the time in s; $C_p$ are the C pools in kg [C] m$^{-2}$ (where $p$ is one of $DPM$, $RPM$, $BIO$, $HUM$); $\Lambda_{c,i}$ is the litter input for PFT $i$ in kg [C] m$^{-2}$ s$^{-1}$ (term in brackets in Equation 22); $f_{DPM,i}$ represents the fraction of litter from each PFT $i$ that goes into $DPM$ with the rest $(1 - f_{DPM,i})$ going into the $RPM$ pool (dependent on amount of woody vegetation); and $R_{tot}$ is the total turnover in kg [C] m$^{-2}$ s$^{-1}$, where the $R_p$ represent the turnover of each C pool:

$$R_{tot} = R_{DPM} + R_{RPM} + R_{BIO} + R_{HUM} \tag{28}$$

The soil respiration to the atmosphere ($r_h$ in kg [C] m$^{-2}$ s$^{-1}$) is given by:

$$r_h = (1 - \beta_R)R_{tot} \tag{29}$$

where $\beta_R$ depends on soil clay content ($clay$ in %) and ranges from 0.25 for a soil with no clay content to 0.15 for a clay soil:

$$\beta_R = \frac{1}{4.09 + 2.67e^{(-0.079clay)}} \tag{30}$$

For each C pool there is an equivalent N pool with the N pools following a similar structure to the C pools:

$$\frac{dN_{DPM}}{dt} = \sum_i \left(v_i f_{DPM,i}\Lambda_{n,i}\right) - M_{DPM} \tag{31}$$

$$\frac{dN_{RPM}}{dt} = \sum_i \left(v_i(1 - f_{DPM,i})\Lambda_{n,i}\right) - M_{RPM} \tag{32}$$

$$\frac{dN_{BIO}}{dt} = 0.46I_{tot} - M_{BIO} \tag{33}$$

$$\frac{dN_{HUM}}{dt} = 0.54I_{tot} - M_{HUM} \tag{34}$$

Inputs into the litter pools ($DPM$, $RPM$) are from the litter N flux ($\Lambda_{n,i}$ in $\mathrm{kg\,[N]\,m^{-2}\,s^{-1}}$, Equation 23) and losses are determined by the pool specific mineralisation of organic N into inorganic N ($M_p$ in $\mathrm{kg\,[N]\,m^{-2}\,s^{-1}}$). Following the framework of the Roth-C model, input into both the $BIO$ and $HUM$ N pools is from the total immobilisation of inorganic N into organic N ($I_{tot}$ in $\mathrm{kg\,[N]\,m^{-2}\,s^{-1}}$):

$$I_{tot} = I_{DPM} + I_{RPM} + I_{BIO} + I_{HUM} \tag{35}$$

For each soil C pool ($p$), the potential turnover - i.e. the turnover rate when the N in the system is not limiting - is given by ($R_{p,pot}$):

$$R_{p,pot} = k_p C_p F_T(T_{soil}) F_\theta(\theta) F_v(v) \tag{36}$$

where the $k_p$ are fixed constants in $\mathrm{s^{-1}}$ (Clark et al., 2011). The functions of temperature ($F_T(T_{soil})$) and moisture ($F_\theta(\theta)$) depend on the temperature ($T_{soil}$) and moisture content ($\theta$) near the soil surface. The function $F_v(v)$ depends on the vegetation cover fraction ($v$) (Clark et al., 2011). The potential mineralisation of organic N when the system is not N limited ($M_{p,pot}$) is related to the potential turnover rates by the C to N ratio of each pool ($CN_p$):

$$M_{p,pot} = \frac{R_{p,pot}}{CN_p} \tag{37}$$

Similarly, the potential immobilisation of inorganic N into the organic N pools ($I_{p,pot}$) is related to pool potential turnover ($R_{p,pot}$), the retained fraction of respiration ($\beta_R$), and the C to N ratio of the destination pool in the decomposition chain:

$$I_{p,pot} = \beta_R \frac{R_{p,pot}}{CN_{soil}} \tag{38}$$

Where $CN_{soil}$ is a model parameter that fixes the C to N ratios of the two destination soil organic pools ($HUM$ and $BIO$) and has a default value of 10. The C to N ratio of the $DPM$ and $RPM$ litter pools is a function of litter quality and varies temporally and spatially depending on the contributions of the different PFTs within the grid cell. Potential mineralisation ($M_p$) and potential immobilisation ($I_p$) fluxes are defined before any N limitation is applied and take values that maintain the constant C:N ratio for the $HUM$ and $BIO$ pools.

When N is limiting, the turnover of the two litter pools ($DPM$ and $RPM$) into the soil organic matter pools is additionally limited by the availability of N.

$$R_p = k_p C_p F_T(T_{soil}) F_\theta(\theta) F_v(v) F_N \tag{39}$$

where $p$ is one of $RPM$ or $DPM$. The nitrogen limited mineralisation and immobilisation of the $DPM$ and $RPM$ pools (Equations 41 and 37) are now effectively a function of $R_p$.

$F_N$ is the litter decomposition rate modifier and is given by the ratio of the N available in the soil to the N required by decomposition (Equation 40). $F_N$ is limited to a range of 0.0 to 1.0. When $F_N$ is equal to 1, the decomposition, mineralisation and immobilisation take place at the potential rate and the system is not N limited. Where $F_N$ is less than 1, the availability of N limits the decomposition of litter into soil organic matter. This limitation is because respiration is carried out by microbes who require sufficient N to convert the $RPM$ and $DPM$ pools into $BIO$ and $HUM$ pools. $F_N$ is given by:

$$F_N = \frac{(M_{BIO} + M_{HUM} - I_{BIO} - I_{HUM})\Delta t + N_{in}}{(D_{DPM} + D_{RPM})\Delta t} \tag{40}$$

where $N_{in}$ is the total soil inorganic N pool in $\text{kg}\,[\text{N}]\,\text{m}^{-2}$ (discussed in Section 3.3 and defined in Equation 51) and $\Delta t$ is the time step. $D_{DPM}$ and $D_{RPM}$ are the net demand associated with decomposition of each of the litter pools:

$$D_p = I_{p,pot} - M_{p,pot} \tag{41}$$

where $p$ is one of $RPM$ or $DPM$. This demand is always positive because the C to N ratio of soil is very much less than the C to N ratio of the $DPM$ and $RPM$ pools. When the net demand is in excess of the available inorganic N, the system is N limited and $F_N < 1.0$. This available N is mainly the net mineralised N from the turnover of $BIO$ and $HUM$ pools but also from the inorganic N pool. N limitation reduces the soil respiration, mineralisation and immobilisation of the two litter pools ($RPM$ and $DPM$). The C:N ratio of these two pools are variable in time and are represented as prognostic variables. The other two organic matter pools ($BIO$ and $HUM$) always respire and are mineralised and immobilised at the potential rate (so $F_N$ is effectively 1.0).

If the net mineralisation is positive some of the N is emitted as gas, according to:

$$N_{gas} = f_{gas}(M_{tot} - I_{tot}) \tag{42}$$

where $N_{gas}$ is one component of the gas emission in $\text{kg}\,[\text{N}]\,\text{m}^{-2}\text{s}^{-1}$, $f_{gas}$ is a parameter that sets the fraction of the N flux that is emitted as gas to the atmosphere. Following Thomas et al. (2013a), it is assumed that 1% of net mineralisation is emitted as gas ($f_{gas}$ is set to 0.01). $M_{tot}$ is the the total mineralisation flux in $\text{kg}\,[\text{N}]\,\text{m}^{-2}\,\text{s}^{-1}$:

$$M_{tot} = M_{DPM} + M_{RPM} + M_{BIO} + M_{HUM} \tag{43}$$

If pool sizes become too small $N_{gas}$ could become negative to ensure N is conserved.

### 3.2.1 Vertical discretisation

The vertical discretisation of the soil C and N follows Burke et al. (2017). There is a set of four soil C and N pools ($DPM$, $RPM$, $BIO$, $HUM$) in every soil model layer. As in Burke et al. (2017) the

turnover rate is determined for each soil layer depending on the temperature, moisture conditions and N availability in that layer. An extra reduction of turnover with depth ($z$) is included to account for factors that are currently missing in the model such as priming effects, anoxia, soil mineral surface and aggregate stabilisation. The potential turnover of each layer is given by:

$$R_{p,pot}(z) = k_p C_p(z) F_T(T_{soil}(z)) F_\theta(\theta(z)) F_v(v) \exp(-\xi_{resp}z) \tag{44}$$

$F_T(T_{soil}(z))$, $F_\theta(\theta(z))$ and $C_p(z)$ are now all dependent on depth. $T_{soil}(z)$ and $\theta(z)$ are the simulated layered soil temperature and soil moisture content and $C_p(z)$ is the simulated soil C content for each layer and pool $p$. The additional reduction of turnover with depth is exponential, with $\xi_{resp}$ an empirical parameter (in m$^{-1}$) that controls the magnitude of the reduction (Burke et al., 2017). The larger the value of $\xi_{resp}$, the more inhibited the respiration is with increasing depth. Here $\xi_{resp}$ was tuned to give a realistic estimate of soil C in a vertically resolved version of JULES-C as in Burke et al. (2017). When N is limiting, the respiration of the $DPM$ and $RPM$ pools are reduced by a factor of $F_N(z)$ which is also now a function of depth and dependent on the available N in the relevant layer. $M_p$ and $I_p$ are also calculated as a function of depth based on their relationship with respiration.

The vertical mixing of each soil N pool follows that of the soil C pools:

$$\frac{\partial N_{DPM}(z)}{\partial t} = \frac{\partial}{\partial z}\left(D(z)\frac{\partial N_{DPM}(z)}{\partial z}\right) + \sum_i (v_i f_{DPM,i} \Lambda_{n,i} f_{lit}(z)) - M_{DPM}(z) \tag{45}$$

$$\frac{\partial N_{RPM}(z)}{\partial t} = \frac{\partial}{\partial z}\left(D(z)\frac{\partial N_{RPM}(z)}{\partial z}\right) + \sum_i (v_i(1 - f_{DPM,i}) \Lambda_{n,i} f_{lit}(z)) - M_{RPM}(z) \tag{46}$$

$$\frac{\partial N_{BIO}(z)}{\partial t} = \frac{\partial}{\partial z}\left(D(z)\frac{\partial N_{BIO}(z)}{\partial z}\right) + 0.46 I_{tot}(z) - M_{BIO}(z) \tag{47}$$

$$\frac{\partial N_{HUM}(z)}{\partial t} = \frac{\partial}{\partial z}\left(D(z)\frac{\partial N_{HUM}(z)}{\partial z}\right) + 0.54 I_{tot}(z) - M_{HUM}(z) \tag{48}$$

$I_{tot}(z)$ is the total immobilisation in kg [N] m$^{-2}$ s$^{-1}$ in each layer (following Equation 35). $D(z)$ is the diffusivity in m$^2$ s$^{-1}$ and varies both spatially and with depth ($z$) (Burke et al., 2017):

$$D(z) = \begin{cases} D_o & ; & z \leq 1m \\ \frac{D_o}{2}(3-z) & ; & 1m < z < 3m \\ 0.0 & ; & z \geq 3m \end{cases} \tag{49}$$

$D_o$ (m$^2$ s$^{-1}$) varies spatially depending on the freeze/thaw state of the soil. In regions without permafrost, $D_o$ represents a bioturbation mixing rate equivalent to 1 cm$^2$ year$^{-1}$. When permafrost is

present, $D_o$ represents cryoturbation and increases to a value equivalent to 5 cm$^2$ year$^{-1}$. The pa-
rameterisation of $D(z)$ in Equation 49 means that the soil organic pools can transfer between the
active layer and the permanently frozen soils in a steady state climate albeit at a relatively slow
rate. The PFT dependent litter inputs ($f_{lit}(z)\Lambda_{n,i}$ for litter N) are distributed so that they decline
exponentially with increasing depth. Here $f_{lit}(z)$ is independent of the PFT type and hence the root
distribution:

$$f_{lit}(z) = \frac{e^{-\xi_{lit}z}}{\sum\limits_{z=0}^{z_{max}} e^{-\xi_{lit}z}dz} \tag{50}$$

Where $\xi_{lit}$ is the parameter to reduce the litter input with increasing depth and is set to 0.2 m or 5
m$^{-1}$ and z is the mid-point of each layer.

The mineralised gas emissions are now a function of depth ($N_{gas}(z)$) and are calculated by re-
peating Equation 42) for each soil layer. Similarly, the litter decomposition rate modifier ($F_N$) is
calculated by repeating a slightly modified version of Equation 40 for each soil layer. In the verti-
cally resolved version of Equation 40, if the soil layer is frozen $N_{in}$ is not available so effectively
zero.

### 3.3 Inorganic Nitrogen

The changes in the inorganic N pool result from deposition, fixation, immobilisation losses, min-
eralisation inputs, gridbox mean plant uptake and inorganic N losses through leaching and gaseous
emission. For the bulk layer case (JULES-CN), these terms are simply added together:

$$\frac{dN_{in}}{dt} = N_{dep} + \sum_i v_i BNF_i - \sum_i v_i \Phi_i + M_{net} - N_{leach} - N_{gasI} \tag{51}$$

where $N_{in}$ is the inorganic N in kg [N] m$^{-2}$, $N_{dep}$ is prescribed N deposition in kg [N] m$^{-2}$ s$^{-1}$ and
$v_i$ the fractional cover of each PFT $i$. The biological N fixation ($BNF_i$) for each PFT $i$ is described
in Section 3.3.1 below and plant uptake ($\Phi_i$) for each PFT $i$ is described in Section 3.1.3. $M_{net}$ is
the net mineralisation which is the difference between $M_{tot}$ (Equation 43) and $I_{tot}$ (Equation 35)
reduced by $N_{gas}$ (Equation 42). The loss of N from the system via the inorganic pool is the sum
of leaching ($N_{leach}$ in kg [N] m$^{-2}$ s$^{-1}$) plus an additional gas loss to the atmosphere ($N_{gasI}$ in
kg [N] m$^{-2}$ s$^{-1}$):

$$N_{gasI} = \gamma_n N_{in} \tag{52}$$

where $\gamma_n$ is a tunable parameter (in s$^{-1}$). The total N gas loss is the sum of $N_{gasI}$ above and $N_{gas}$
from Equation 42 with $N_{gasI}$ representing approximately 90% of the total gas loss. This additional
gas loss term ($N_{gasI}$) represents missing processes relating to the gaseous loss of inorganic N and

limits the effective mineral N pool size. Including $N_{gasI}$ ensures that available N does not increase excessively, potentially due to excessive biological N fixation in regions where the NPP is very close or equal to the $NPP_{pot}$. In the current model configuration $\gamma_n$ is set to 0.0028 day$^{-1}$ such that the whole pool turns over once every model year.


The leaching of N ($N_{leach}$ in kg [N] m$^{-2}$ s$^{-1}$) through the profile is assumed to be a function of the net flux of moisture through the soil profile, the concentration of inorganic N, and a parameter ($\alpha$, dimensionless) representing the effective solubility of N. $\alpha$ is assumed to have a value of 0.1 and in JULES-CN represents the combined sorption of all inorganic N species (Wania et al., 2012).

$$N_{leach} = \alpha(N_{in}/\theta_{1m})Q_{subs} \tag{53}$$

where $\theta_{1m}$ is the soil water content in the top 1m of soil in kg m$^{-2}$ (so the inorganic N is assumed to occupy the top 1m of soil), and $Q_{subs}$ is the total subsurface runoff in kg m$^{-2}$ s$^{-1}$.

### 3.3.1   Biological Nitrogen Fixation ($BNF$)

Biological nitrogen fixation ($BNF$) is the largest natural supplier of N to the terrestrial ecosystem. Following the secondary model of Cleveland et al. (1999), N fixation is determined as a linear proportion of the net primary production before N limitation of each PFT $i$ ($NPP_{pot,i}$):

$$BNF_i = \zeta NPP_{pot,i} \tag{54}$$

$NPP_{pot,i}$ is defined in the same way as the net primary productivity in JULES before the ex-
plicit N cycle was included, i.e. before the excess carbon ($\Psi$) is removed. $BNF$ as a function of NPP is an established method used and assessed in other models (Meyerholt et al., 2016; Wieder et al., 2015a; Thomas et al., 2013b). While some models utilise more complex $BNF$ representations (Fisher et al., 2010), a lightweight approach is preferred here while the benefits of extra computational expense on $BNF$ are not yet established, and evidence is lacking that a different simple
representation (e.g. evapotranspiration) would perform better (Davies-Barnard and Friedlingstein, 2020). However, changes in NPP may not accurately reflect changes in $BNF$ with forcings such as elevated atmospheric carbon dioxide (Liang et al., 2016) or additional N (Thomas et al., 2013b; Ochoa-Hueso et al., 2013).

The rate of fixation ($\zeta$) is set such that global present day net primary productivity of approximately 60 Pg C yr$^{-1}$ results in approximately 100 Tg N yr$^{-1}$ fixation (0.0016 kg [N] kg C $^{-1}$), within the range of recent global observation-based estimates of $BNF$ (Davies-Barnard and Friedlingstein, 2020; Vitousek et al., 2013). The parameterisation based on NPP results in a latitudinal gradient with the highest rates of fixation in the tropics and lowest in boreal forests and arctic tundra which is con-
sistent with some estimates of $BNF$ (Houlton et al., 2008; Cleveland et al., 1999) though not recent

observational meta-analyses (Davies-Barnard and Friedlingstein, 2020).

In JULES-CN which has a bulk soil biogeochemistry parameterisation the $BNF$ is directly trans-
ferred into the single inorganic soil N pool and becomes available as inorganic N. However, in
JULES-CN$_{layer}$ the $BNF$ is distributed vertically in the soil depending on the fraction of roots in
each layer. If a soil layer is frozen there is no $BNF$ into that layer. If the whole soil is frozen, fixed
N goes into the inorganic N pool in the top layer.

### 3.3.2 Vertical discretisation of inorganic nitrogen

In JULES-CN$_{layer}$ there is an inorganic N pool in each soil layer. The dynamics are very similar to
Equation 51, but most of the components now vary with depth:

$$\frac{dN_{in}(z)}{dt} = \sum_i v_i BNF_i f_{R,i}(z) - \sum_i v_i \Phi_i f_{I,i}(z) + M_{net}(z) - N_{flux}(z) - N_{gasI}(z) \tag{55}$$

Any N deposition ($N_{dep}$) is added to the top layer of the soil only. The modifications to each term to
ensure they vary appropriately with depth are discussed below. The additional parameters in Equa-
tion 55 are $f_{R,i}(z)$ - the fraction of roots in each layer for PFT $i$ (Equation 56); $f_{I,i}(z)$ - the fraction
of available inorganic N in each layer for PFT $i$ (Equation 60) and $N_{flux}(z)$ - the transport of inor-
ganic N through the layer by the soil water fluxes (Equation 61).

As in Equation 51 the net mineralisation flux ($M_{net}(z)$) is the difference between $M_{tot}(z)$ and
$I_{tot}(z)$ reduced by $N_{gas}(z)$ for each layer (see Section 3.2.1). Inputs from biological N fixation from
PFT $i$ are distributed according to the root profile of the PFT under consideration ($f_{R,i}(z)$):

$$f_{R,i}(z) = \frac{f_{root,i}(z)}{\sum\limits_{z=0}^{z_{max}} f_{root,i}(z)dz} \tag{56}$$

where $f_{root,i}(z)$ is the volumetric root fraction of PFT $i$ at a given soil level and $z_{max}$ is the
maximum depth of the soil in m. Gas loss from the inorganic N ($N_{gasI}(z)$) occurs in each layer,
but with an additional exponential decay term which is a function of depth (similar to that used in
Equation 44 for the soil decomposition). This term empirically represents the factors that reduce soil
activity with depth. The additional gas loss term thus becomes:

$$N_{gasI}(z) = \gamma_n N_{in}(z) e^{-\xi_{resp} z} \tag{57}$$

This leaves two terms in Equation 55: the plant uptake term ($\sum\limits_i v_i \Phi_i f_{I,i}(z)$) which is PFT depen-
dent and the $N_{flux}$(z) term, which replaces the leaching term from Equation 51. These have a more
process-based representation in the layered case. When calculating the plant uptake term we assume
that plants cannot access all the inorganic N. Firstly, if a soil layer is frozen then plants cannot uptake

any of the N in that layer. Secondly, we assume that they only have direct access to a certain fraction of the soil, according to their root fraction, $f_{root,i}(z)$ (which reduces with depth). So for each PFT, $i$, the available amount of the inorganic N pool ($N_{avail,i}(z)$ in kg [N] m$^{-2}$) at equilibrium that could potentially be extracted by the vegetation is given by:

$$N_{avail,i}(z) = f_{root,i}(z)N_{in}(z)T(z) \tag{58}$$

Where $T(z)$ is zero when the soil temperature is $0^oC$ or colder and 1 when it is above $0^oC$. However, the system is not necessarily in equilibrium - as N is taken up from this availible pool around the roots, there will be a delay in this volume getting 're-filled'. We assume that the inorganic N is constantly diffusing back to the equilibrium state where the concentration is constant both horizontally and vertically within each layer, and thus after the extraction of inorganic N by the plants on each TRIFFID timestep we additionally update the available N pool according to:

$$\frac{N_{avail,i}(z)}{dt} = \gamma_{dif}(f_{root,i}(z)N_{in}(z) - N_{avail,i}(z)) \tag{59}$$

where $\gamma_{dif}$ is the rate of diffusion back to the equilibrium, set by default to 0.28 day$^{-1}$ or approximately 100 year$^{-1}$. $N_{avail,i}(z)$ is then multiplied by $T(z)$ to incorporate the frozen soil effect. Any biological N fixation goes directly into the available pool. Plant uptake is extracted entirely from the available N pool, and the dependence on depth is according to the same profile as the available N, i.e.

$$f_{I,i}(z) = \frac{N_{avail,i}(z)}{\sum\limits_{z=0}^{z_{max}} N_{avail,i}(z)dz} \tag{60}$$

All of the other fluxes are simply added in such a manner so as to maintain the ratio of the available to total inorganic N pools that would be present if the available and total pools were in equilibrium. Therefore the only two processes which take the available and total pools out of equilibrium are biological N fixation and uptake.

Leaching is now done in a process-based manner, where the inorganic N is transported through the soil profile by the soil water fluxes. For any given soil layer of thickness $\delta z$, the inorganic N flux ($N_{flux}$) is given by:

$$N_{flux}(z) = \alpha \delta z \frac{d}{dz}\left(W_{flux}(z)\frac{N_{in}(z)}{\theta(z)}\right) \tag{61}$$

where $\theta(z)$ is the soil water content of the layer in kg m$^{-2}$ and $W_{flux}(z)$ is the flow rate of the water through the layer in kg m$^{-2}$ s$^{-1}$. Multiplying by $\delta z$ gives the change in N content for each layer. The total leaching is then the sum of all N that leaves the soil both laterally from each layer or from the bottom of the soil profile.

| Variable | Value | Description | Equation |
|---|---|---|---|
| **Bulk soil nitrogen** | | | |
| $\zeta$ | 0.0016 kg [N] kg [C] $^{-1}$ | Rate of $BNF$ | Equation 54 |
| $CN_{soil}$ | 10 kg [C] kg [N] $^{-1}$ | CN ratio of $BIO$ and $HUM$ pools | Equation 41 |
| $f_{gas}$ | 0.01 (proportion) | Fraction of net mineralisation emitted as gas to atmosphere | Equation 42 |
| $\gamma_n$ | 3.215e-08 s $^{-1}$ | Imposed turnover coefficient to determine $N_{gasI}$ release from $N_{in}$ | Equation 52 |
| $\alpha$ | 0.1 (proportion) | Effective solubility of nitrogen in water | Equation 53 |
| **Vertically resolved soil carbon** | | | |
| $\xi_{resp}$ | 0.8 m $^{-1}$ | Parameter to control reduction of respiration with depth | Equation 44 |
| $D_o$ | bioturbation - 0.001 m$^2$s$^{-1}$ cryoturbation - 0.005 m$^2$s$^{-1}$ | Soil carbon and nitrogen mixing rate | Equation 49 |
| $\xi_{lit}$ | 5 m $^{-1}$ | Parameter to control reduction of litter input with depth | Equation 50 |
| **Vertically resolved soil carbon and nitrogen** | | | |
| $\gamma_{dif}$ | 100 per 360 days | Rate of diffusion transferring the inorganic nitrogen from $N_{in}$ to $N_{avail}$ | Equation 59 |

**Table 2.** A summary of the extra parameters required for the soil biogeochemistry component of JULES-CN and JULES-CN$_{layer}$.

Table 3.3.2 summarises the extra parameters required for the soil biogeochemistry component of JULES-CN and JULES-CN$_{layer}$ alongside their values.

## 4 Historical simulations

Global transient simulations were carried out following the protocol for the S2 experiments in TRENDY (Sitch et al., 2015) which include time varying climate, $CO_2$, and N deposition but pre-industrial land use. Time-varying $CO_2$, and climate were from the from the CRU-NCEP data-set (v4, 1901-2012, Viovy N. 2011 CRU-NCEPv4. CRUNCEP dataset). The fraction of agriculture in each grid cell was set to the pre-industrial value defined by (Hurtt et al., 2011). N deposition was taken from a ACCMIP multi-model data set interpolated to annual fields (Lamarque et al., 2013). The model resolution was N96 (1.875° longitude x 1.25° latitude).

Results from three different JULES model configurations are presented here:

– JULES-CN includes the newly developed soil and vegetation coupled C and N cycle.

– JULES-C is shown for comparison purposes and represents the soil and vegetation C cycle as used in Le Quéré et al. (2018).

– JULES-CN$_{layer}$ is a version of JULES-CN which has identical above ground processes to JULES-CN but additionally includes vertically discretised soil biochemistry.

In each case five PFTs were used: broadleaf trees, needleleaf trees, C3 and C4 grass and shrubs. Plant competition was allowed, with TRIFFID updating vegetation fractions on a 10 day time step. These three configurations of JULES adopt the standard 4 layer soils with a maximum depth of 3
m. However it should be noted that Burke et al. (2017) and Chadburn et al. (2015) adopt a configuration which increases both the maximum soil depth and number of soil layers - a configuration which is recommended for detailed scientific study of northern high latitudes. The sole difference between JULES-C and JULES-CN is the inclusion of the N cycle. JULES-CN$_{layer}$ additionally has vertically discretised soil biogeochemistry. There are no differences in any of the shared model pa-
rameters which were initially tuned for the JULES-C configuration.This enables a direct comparison between the different configurations.

The simulations were initialised using pre-industrial conditions. The models were spun up by using the meteorological data for the period 1860-1879 repeatedly until the change in the carbon
stocks was less than 0.01 % decade$^{-1}$ globally. The soil C distribution in JULES-CN$_{layer}$ is particularly slow to reach equilibrium. Therefore the 'modified accelerated decomposition' technique (modified-AD) described by Koven et al. (2013) was used to spin the soil C in these versions up to an initial pre-industrial equilibrium distribution (Burke et al., 2017). Further spin up was then carried out for these layered models using repeated pre-industrial conditions until the change in soil C was
again less than 0.01 % decade$^{-1}$ globally. It should be noted that neither transient land-use change or fertiliser were included in any of these simulations.

## 5 Results

This paper mainly focuses on the differences in JULES output when including the N cycle in the model configuration. When available, we additionally use any observational based estimates to eval-
uate the quality of the simulations. First a broad-brush comparison between JULES-CN, JULES-C and JULES-CN$_{layer}$ is made. This is followed by a more complete discussion of the impact of the N cycle on the carbon stocks and fluxes and their changes over time. Then we show spatial distributions and time series of the N stocks and fluxes. Finally the extra processes modelled in JULES-CN$_{layer}$ are assessed. For completeness figures often include both JULES-CN and JULES-CN$_{layer}$

but JULES-CN$_{layer}$ is only discussed at the end of the results.

It should be noted that the addition of the N cycle in JULES is only one component of the recent developments. In the UKESM configuration of JULES Sellar et al. (2019) the N cycle was combined with a new competition scheme Harper et al. (2018) and additional PFTs both of which modify the

global vegetation distribution. Therefore we are most interested in the changes in the vegetation distribution between the different versions which will be caused by the N cycle. Figure 3 shows the total area covered by each type of vegetation. The Climate Change Initiative (CCI) land cover observations Hartley et al. (2017) are added for completeness. In general, the models all tend to over-estimate the shrubs and underestimate the grass. However, Sellar et al. (2019) shows that once the

additional PFTs and new competition scheme are included the model does a good job of representing the vegetation distribution.

As expected the configurations with the N cycle have more bare soil and less vegetation than JULES-C. This is mainly observed as a decrease in the shrub and grass regions in JULES-CN. As we shall see later (Figure 10) this is because the tropical forests dominate the tree region and their

growth is not limited by N in the model. JULES-CN$_{layer}$ has a reduction in trees compared to JULES-CN, which is focused in the boreal region where it is more likely to simulate grass or shrubs.

### 5.1 Summary of C and N stocks and fluxes

Figure 4 provides an overview of the stocks and fluxes of C and N in JULES-CN and JULES-

CN$_{layer}$ and compares them with JULES-C. As expected for a present-day simulation, the majority of C stocks and fluxes are very similar for JULES-C and JULES-CN. The main difference is the present-day NPP which is ~12% higher in JULES-C than in JULES-CN. There is also a small re-duction in the GPP of ~4% caused by some differences in the vegetation fractional cover distribution (Figure 3) and indirectly resulting from the N limitation.


Soil organic N and vegetation N are both consistent with the available observation-based estimates of stocks. The biological N fixation is tuned to be approximately 100 Tg N year$^{-1}$ in the present day and the N deposition is prescribed. The majority of N losses from the land surface occur via the gaseous pathway with total losses of 111 Tg N year$^{-1}$ for JULES-CN. Leaching is fairly low

at 7 Tg N year$^{-1}$ compared to estimates of leaching, which are as high as ~25 - 55% of N inputs in European forests (Dise et al., 2009) and range between 59 and 118 Tg N year$^{-1}$ in the available observations (Boyer et al., 2006; Galloway et al., 2004). There is no N fertilizer applied in the model which might partially explain why the leaching is so low. In reality there is ~200 Tg N applied annu-ally as either manure or fertilizer Potter et al. (2010), a proportion of this will be leached resulting in

an increase of global leaching. N uptake and net N mineralisation are relatively high and are fairly

comparable in magnitude implying a largely closed cycling of nutrients between vegetation and soil. These N stocks and fluxes are also consistent with results from other models such as: Xu-Ri and Prentice (2008), Smith et al. (2014), Zaehle (2013b) and von Bloh et al. (2018).

## 5.2 Comparing C stocks and fluxes

Carbon use efficiency (CUE) is defined as the ratio of net C gain to gross C assimilation during a given period (NPP/GPP). Plants with a higher CUE have a lower autotrophic respiration and allocate more C from photosynthesis to the terrestrial biomass and vice-versa. In JULES-CN there is less C available to be allocated because it is constrained by the amount of N present. This reduces the C use efficiency. Figure 5 shows the zonal total GPP and NPP for JULES-CN and JULES-C. As expected from Figure 4 the NPP and GPP have very similar latitudinal profiles for the two model configurations. Both JULES-C and JULES-CN have a higher GPP in the tropics than the observations but they are more comparable in the extra-tropical latitudes where the GPP tends to be smaller. The NPP in JULES-CN is less than JULES-C and generally closer to the MODIS observations particularly in the tropics. Figure 5 also shows the zonal mean CUE. JULES-CN has a lower CUE than JULES-C for all latitudes. On average it is 0.44 for JULES-CN and 0.49 for JULES-C. JULES-CN is consistently low compared to the Kim et al. (2018) observation-based data set with a bias of $\sim$0.09. This bias is relatively constant with latitude. However, considerable uncertainties remain in these estimates.

Figure 5 also shows the changes in these C fluxes for the period 1860-2007 with respect to the multi-annual mean period of 1860-1899. Changes over time are shown to enable the differences between the two different model configurations to be more easily compared. Apparently small differences between JULES-C and JULES-CN in the NPP and GPP become more noticeable in the CUE. The small differences between JULES-C and JULES-CN in GPP are mainly caused by small changes in the vegetation distribution and a slight increase in bare soil in JULES-CN. In the case of NPP - JULES-C increases quicker than JULES-CN because JULES-CN becomes progressively more N limited. The change in CUE shows the impact of the N cycle on the uptake of C by the vegetation in JULES-CN over the twentieth century. There is an increase in CUE in both configurations, mainly caused by $CO_2$ fertilisation, but this is limited by N in the JULES-CN configuration.

The zonal total soil and vegetation C stocks are shown in Figure 6. The vegetation C is very similar for both JULES-C and JULES-CN as expected from Figure 4 and is consistent with the available observations. There are some differences in the soil C in the northern high latitudes with JULES-CN having slightly less soil C than JULES-C. This is a consequence of the higher N limitation on JULES-CN leading to less litter fall and subsequently less soil C. The corresponding N limitation induced reduction in soil organic matter decomposition is not strong enough to offset the decrease

in C input leading to a smaller pool size.

### 5.2.1 Net ecosystem exchange

A key measure of a land C cycle model is how well it simulates the temporal variation of the land C
sink, which is the difference between Net Ecosystem Exchange (NEE) and the flux of C to the atmo-
sphere from land-use change. The interannual variability in the sink is dominated by the variability
of NEE, which is itself correlated with the magnitude of the temperature-carbon cycle feedback in
the tropics (Cox et al., 2013). As a result, simulation of NEE variability is highly relevant to climate-
carbon cycle projections (Wenzel et al., 2016).

Figure 7 compares global annual mean values of Net ecosystem exchange (NEE; defined as NPP
- heterotrophic respiration) for JULES-C and JULES-CN to observation-based estimates from the
Global Carbon Project. We specifically focus on the years from 1960 to 2009, which is the maximum
overlap period between the model simulations and the GCP annual budget data (Friedlingstein et al.,
2020). To avoid the circularity of using GCP estimates of NEE which are themselves derived from
land-surface models, we instead calculate the GCP estimates of NEE as the residual of the best
estimates of the total emissions from fossil fuel ($FF$) plus land-use change ($LU$), and the rate of
increase of the carbon content of the atmosphere ($F_a$) plus the ocean ($F_o$):

$$NEE_{gcp} = FF + LU - F_a - F_o \tag{62}$$

The observations and both of the models show an upward trend in NEE but with very significant
interannual variability (Figure 7). Due to N limitations on $CO_2$ fertilization, mean NEE in JULES-
CN (1.66 Pg C year$^{-1}$) is lower than in JULES-C (2.06 Pg C year$^{-1}$), and also lower than the
estimate from GCP (2.11 Pg C year$^{-1}$). This absolute value will be sensitive to the vegetation cover
which is much improved by including the height-based competition as has been done in UKESM1
Sellar et al. (2019). However, JULES-CN outperforms JULES-C on all of the other key metrics
of the NEE variation. JULES-CN produces a smaller but much more realistic trend in NEE, and a
smaller and more realistic interannual variability about that trend (see Table 5.2.1). The correlation
coefficient for NEE between the JULES-CN and GCP estimates (r=0.71) is also improved compared
to JULES-C (r=0.63). There remains a significant underestimate of NEE in the years following the
Pinatubo volcanic eruption in 1991, most likely due to the neglect of diffuse-radiation fertilization
in these versions of JULES (Mercado et al., 2009). However, it is especially notable that JULES-CN
significantly reduces the systematic overestimate of NEE seen in JULES-C during extended La Nina
periods, such as the years centred around 1974 and 2000 (Figure 7).

| | Mean (Pg C year$^{-1}$) | Trend (Pg C year$^{-1}$ year$^{-1}$) | IAV (Pg C year$^{-1}$) | r |
|---|---|---|---|---|
| JULES-CN | 1.66 | 0.025 | 0.86 | 0.71 |
| JULES-C | 2.06 | 0.034 | 1.31 | 0.63 |
| JULES-CN$_{layer}$ | 1.75 | 0.026 | 0.83 | 0.64 |
| GCP(residual) | 2.11 | 0.027 | 1.01 | |

**Table 3.** Statistics of NEE from JULES-CN, JULES-C, JULES-CN$_{layer}$, and the GCP observation-based estimates (Friedlingstein et al., 2020), over the period from 1960 to 2009 inclusive. Columns 2-4 show respectively the mean, linear trend, and the interannual variability (standard deviation) around that trend. Column 5 shows the correlation coefficient between each model NEE timeseries and the GCP timeseries.

### 5.2.2 Residence times

In general, carbon residence times of the soil and ecosystem are given by the stocks divided by the fluxes. These are emergent properties of the model and thus a valuable metric to evaluate. Figure 8 shows the ecosystem residence time and the soil C residence times for different biomes. Here, the land surface is split into biomes based on the 14 World Wildlife Fund terrestrial ecoregions (Olson et al., 2001) and characterised by Harper et al. (2018). The ecosystem residence time defined as the

total ecosystem C divided by the GPP is shown in Figure 8(a). These residence times have been estimated from a multi-annual mean on a grid cell by grid cell basis and then aggregated to biomes. The observational values were derived in a similar way using spatial data from Carvalhais et al. (2014). In general the ecosystem residence times are slightly reduced in JULES-CN compared with JULES-C, both of which are generally lower than in the observations. The largest difference between observed

and modelled ecosystem residence time occurs in the tundra and boreal regions and the grasslands where the observed residence times are much longer than either JULES-C or JULES-CN. The soil carbon residence time is shorter than the observational-based measure in the tundra and the boreal regions but longer in the grassland regions. Overall, this leads to the the global soil carbon residence time in the model being too short. When vertical discretisation, including additional permafrost pro-

cesses, is added in JULES-CN$_{layer}$ the residence times in the boreal and tundra increase notably (see Section 3.2.1 for further discussion).

### 5.3 Impact of N limitation

IN JULES-CN and JULES-CN$_{layer}$ the N limitation mainly acts through reducing the NPP. This

can be quantified using the response ratio which is defined as the ratio of the potential amount of C that can be allocated to growth and spreading of the vegetation ($NPP_{pot}$) compared with the actual amount achieved in the natural state (NPP). Both of these diagnostics are output from the JULES simulations. Figure 9 shows the spatial distribution of the model simulated response ratio.

Green areas are not very N limited with a response ratio close to 1.0 and yellow areas are more N limited with a larger response ratio. There are distinct regions of N limitation - in Australia and south Africa, the Sahel, western Europe and parts of Siberia. However much of the global land surface, particularly the forested regions has relatively weak N limitation. Figure 9(c) also shows the JULES-CN response ratio has obvious inter-annual variability superimposed on an increasing trend over the twentieth century, indicating increasing N limitation which will limit the increase in carbon use efficiency shown in Figure 5(f).

Figure 10 shows the biome-based response ratio of net primary productivity. All biomes have a response ratio of greater than 1 in both the model and observations which means that adding extra N to the system will enhance the NPP achieved. Globally the response ratio is lower than the observations but for the majority of the biomes including the tropical forests and the tundra the model response ratios fall within the range of uncertainties of the observations. However, LeBauer and Treseder (2008) suggests the tropical forest is somewhat N limited, whereas in JULES-CN tropical forest is not a N limited biome. Phosphorus has long been considered as the most limiting nutrient in tropical regions (Yang et al., 2014), therefore we expect JULES to simulate a larger response ratio in the future once a phosphorus cycle is added.

In the model the soil C decomposition can be limited when the N available in the soil is less than the N required by decomposition. This process does not play a major role in our simulations.

## 5.4 Nitrogen stocks and fluxes

The zonal profile of soil organic nitrogen (Figure 11) shows a similar distribution to the soil organic C (Figure 6) reflecting the relatively consistent C to N ratio of the soil within the model. $CN_{soil}$ - the C to N ratio of the HUM and BIO pools - is a spatially constant parameter set to 10 in these simulations. The observed soil N content is slightly higher at all latitudes than simulated by JULES-CN particularly in the northern tundra region. In contrast to the zonal distribution of soil organic nitrogen, the soil inorganic nitrogen in JULES-CN is larger in the tropics than in the northern high latitudes (Figure 11). Figure 12 shows the net soil N mineralisation fluxes are large in the tropics and smaller in the northern regions. This is reflected in the spatial distribution of the N uptake. As might be expected the spatial distribution of the N uptake as a fraction of N demand is similar to the N limitation shown in Figure 9. Biological N fixation and N gas losses are an order of magnitude smaller than the N uptake and net N mineralisation. However, again the spatial patterns are very comparable. N leaching is generally very small except in parts of south America and south-east Asia. Figure 13 shows a slight increase in the N demand and N uptake over the twentieth century associated with the increase in vegetation growth (Figure 5). Similarly there is an increase in the

BNF which is parameterised such that it is proportional to the NPP.

## 5.5 Impact of vertical discretisation of soil biochemistry

This section discusses the differences between JULES-CN and JULES-CN$_{layer}$. In general over the tropics and southern latitudes, JULES-CN$_{layer}$ is very comparable to JULES-CN. The majority of
the differences occur in the northern regions where there is soil freezing–either permafrost or seasonally frozen soils. The reduction in global mean tree covered area seen in Figure 3 is caused by a reduction in the boreal regions which have a larger proportion of shrubs and grasses in JULES-CN$_{layer}$. In the higher latitudes the soil in JULES-CN$_{layer}$ also has more organic C (Figure 6). This increase in soil organic C represents a store of permafrost carbon more comparable to the carbon
found by Batjes (2014) and Carvalhais et al. (2014). This build up of carbon in JULES-CN$_{layer}$ occurs because the decomposition deeper in the soil is reduced with the lower soil temperatures at depth - the soil C in JULES-CN only respond to the soil temperatures near the surface which are warmer. This also causes in increase in the residence time of the soil carbon shown in Figure 8(b). The modelled soil C residence time in JULES-CN$_{layer}$ is now much longer and more comparable to
that observed.

The spatial distributions of N fluxes in JULES-CN$_{layer}$ (not shown) are very similar to those of JULES-CN. In addition, the time series of changes in N fluxes over the twentieth century are also comparable (Figure 13). The main differences are in the N gas loss which is larger in JULES-
CN$_{layer}$ and the N leaching which is larger in JULES-CN. Figure 11 shows an increase in both organic and inorganic N in JULES-CN$_{layer}$ over that in JULES-CN in the northern high latitude similar to that seen in the organic C. As is the case for soil organic C, in the colder regions the soil N builds up within the frozen soil because of the limitation of the decomposition rates by cold temperatures, therefore larger pools deeper in the soil are maintained in an equilibrium climate. The
parameterisation of the vertically resolved soil biogeochemistry means that, once JULES-CN$_{layer}$ is spun-up there is inorganic N within the soil profile which cannot be taken up by the vegetation, either because the soil is frozen or because the roots cannot readily access it. This means that the extra inorganic N in JULES-CN$_{layer}$ (Figure 11) is mainly stored deeper in the soil profile and within the permafrost itself and is typically inavailable in the current climate. This improved representation
of the soil biogeochemistry will have implications for simulations of climate change feedbacks from the northern high latitudes.

## 6 Discussion

This study presents the first implementation of nutrient cycles into the UK land and earth system models. The scheme is parsimonious in that it captures the first order and large scale effects of interacting carbon and nitrogen on the land surface in the simplest way possible. One important assumption is that of fixed plant stoichiometry and that a plant strives to achieve stoichiometric homeostasis to maintain ecosystem structure, function and stability under change environments (Sterner and Elser, 2002). This assumption has some support in the literature (e.g Brix and Ebell (1969); Wang et al. (2012)) and is a common approach amongst complex DGVMs (Meyerholt and Zaehle, 2015). However, recent meta analyses of field observations show a distinct increase in foliar N to additional N availability (Mao et al., 2020) and a modelling study found that assuming fixed C : N ratios and/or scaling leaf N concentration changes to other tissues, as employed here, were not supported by available evaluation data (Meyerholt and Zaehle, 2015). Employing flexible stoichometry has the potential to significantly affect the modelled biogeochemical feedbacks. For instance, nutrient limitation tends to limit productivity and thus the production of litter, the input to soil organic matter, leading to a reduction in soil carbon that the nutrient limitation in soil turnover is too weak to oppose. Allowing for flexible stoichometry may lead to a lower litter quality but a comparable amount of litter. This reduction in litter quality will strengthen the soil turnover response possibly leading to an overall increase in soil organic matter. Plant stochiometric relationships are therefore a key uncertainty in assessing the carbon cycle feedbacks to climate change. Future versions of this model will explore the use of plant trait information (Harper et al., 2016) to parameterise leaf, root and wood C:N ratios for individual PFTs, and further developments to allow for flexible stoichiometry.

While the total BNF in JULES-CN is in the range of Davies-Barnard et al. (2020) and Vitousek et al. (2013), the spatial distribution of BNF more heavily favours the tropics than recent observations suggest (Sullivan et al., 2014; Davies-Barnard et al., 2020). The response of BNF to the multiple factors likely to occur in future varies between factor (e.g. warming, elevated atmospheric carbon dioxide, drought, N deposition, etc.), biome, and BNF type (nodulating, bryophyte, litter, etc.) (Zheng et al., 2020). Therefore how BNF will change is spatially variable and not controlled by a single factor. A move from an empirical to a process driven BNF function may provide better fit to present day BNF distribution and more robust future projections.

Further work is required to explore the impact of a spatially varying soil C to N ratio which can vary widely depending on the amount and decomposition of organic matter within the soil. For example, peat soils have relatively high C to N ratios up to 30-40 Hugelius et al. (2020). This type of soil is not yet included within JULES.In addition, N leaching is very low in the model, notwithstanding the lack of N fertiliser. One reason for this could be that too much mineral N is assumed to be

sorped within the soil. This requires further evaluation and potential modifications to the scheme.

In this paper we have not explicitly separated the impact of $CO_2$ fertilization from climate change or from the impact of N deposition. However, this was explored by Davies-Barnard et al. (2020) who put the response of JULES in context by comparing it with the responses from 4 additional land

surface models and a meta-analysis of site observations. Davies-Barnard et al. (2020) used a slightly different configuration of JULES (JULES-ES) which is the configuration used in UKESM1 with a bulk soil biogeochemistry (Sellar et al., 2019). They found that JULES-ES has a relatively small increase in NPP caused by the addition extra N in the form of deposition compared with both the meta-analyses and CLM / LPJ-GUESS. However, it is comparable to that found in JSBACH. This

small response is, in part, caused by the smaller initial N limitation in JULES-ES. However, JULES' increase in NPP in response to $CO_2$ fertilisation is aligned with the majority of the models and the meta-analyses.

## 7 Conclusions

In this paper we have documented a model to quantify the impact of coupling the nitrogen cycle with the carbon cycle in a fully dynamic vegetation model. In the model, N limitation affects NPP and how much C is allocated but it only indirectly affects the photosynthesis via leaf area development. This enables the carbon use efficiency (ratio of net carbon gain to gross carbon assimilation) to respond to changing N availability. Since the CUE affects the ability of the land surface to uptake

carbon in a changing climate, this will impact carbon budgets under future projections of climate change. This scheme (based on JULES-CN) is only one of the new components of JULES that has been included within UKESM1 (Sellar et al., 2019). Relevant additions to the JULES-ES configuration used in UKESM1 includes more plant functional types with improved plant physiology and vegetation dynamics (Harper et al., 2016) plus a new land use module (Robertson et al., in prep.).


Overall the N enabled configuration of JULES – JULES-CN – produces a more realistic trend in the net ecosystem exchange (NEE) and the interannual variability of NEE about that trend. It also produces an improved estimate of NPP in the northern high latitudes. For other regions and diagnostics the simulation of present-day state and behaviour is not substantially different between

JULES-C and the N-enabled configuration, JULES-C. This is largely because JULES-C has been tuned to replicate observed carbon stores and fluxes and therefore implicitly includes a level of N availability. What JULES-C lacks is a mechanism for this to change substantially in time – either under more limiting conditions as elevated $CO_2$ outpaces demand for nutrients (e.g. Zaehle (2013b)), or under conditions of increased N availability due to anthropogenic deposition or accelerated soil

decomposition caused by climate change leading to increased mineralisation rates (Meyerholt et al., 2020b; Zaehle and Dalmonech, 2011). The response of the N cycle in JULES under changes in climate and $CO_2$ conditions–which will be affected by nutrient limitations–will be quantified and assessed in subsequent work.

An extended version of the nitrogen-enabled model additionally includes the vertical discretisation of the soil biogeochemistry model. This configuration improves the ecosystem residence times in the tundra and boreal regions. This more detailed representation of permafrost biogeochemistry in the northern high latitudes will used to understand the impact of the coupled carbon and nitrogen cycle on the permafrost carbon feedback.


## Code Availability

The JULES code used in these simulations is available from the Met Office Science Repository Service (registration required) at https://code.metoffice.gov.uk/trac/jules. To access the code a freely available non-commercial research licence is required (https://jules-lsm.github.io/). The suites re-

quired for running JULES are available here: https://code.metoffice.gov.uk/trac/roses-u. JULES-CN is u-ah896, JULES-C is u-ah932 and JULES-CN$_{layer}$ is u-ai571.

## Competing Interests

The authors declare that they have no conflict of interest.

*Author contributions.* Andy Wiltshire designed the model in collaboration with the rest of the co-authors and
wrote the first draft of the text. Eleanor Burke and Sarah Chadburn added the layered soil biogeochemistry. Peter Cox performed the analysis of inter-annual variability. All authors reviewed the paper and proposed improvements.

*Acknowledgements.* This work has received funding from the European Union's Horizon 2020 research and innovation programme under grant agreement No 641816 Coordinated Research in Earth Systems and Climate:
Experiments, kNowledge, Dissemination and Outreach (CRESCENDO). Andy Wiltshire and Eleanor Burke were also supported by the Joint UK BEIS/Defra Met Office Hadley Centre Climate Programme (GA01101). Sarah Chadburn was supported by the Natural Environment Research Council (grant no. NE/M01990X/1 and NE/R015791/1). Peter Cox was supported by the European Research Council (ERC) ECCLES project, grant agreement number 742472. Sönke Zaehle was also supported by the European Research Council (ERC) under
the European Union's Horizon 2020 research and innovation programme (grant agreement No 647204).

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

**Figures**

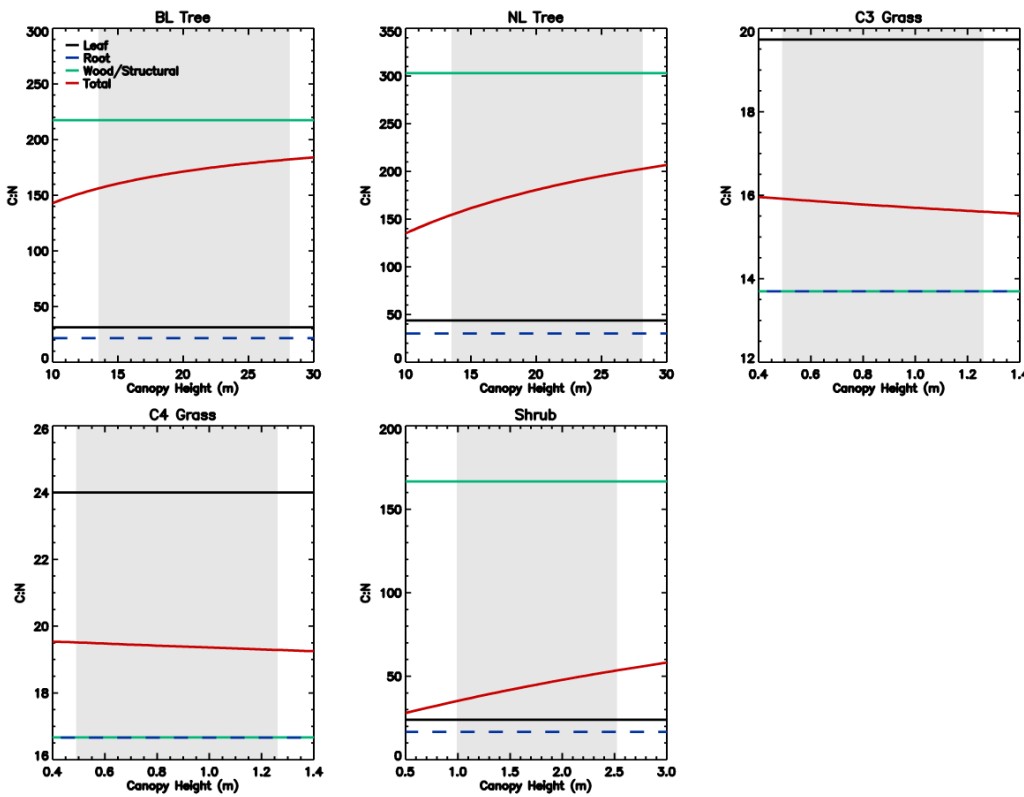

**Figure 1.** Stoichiometry of the vegetation nitrogen pools as a function of canopy height for individual PFTs at full leaf. Leaf N concentration are defined at the canopy level and are higher than those for the top leaf. The grey region shows the defined range of canopy height within the model. Note: both the x- and y-scales are very different for each PFT.

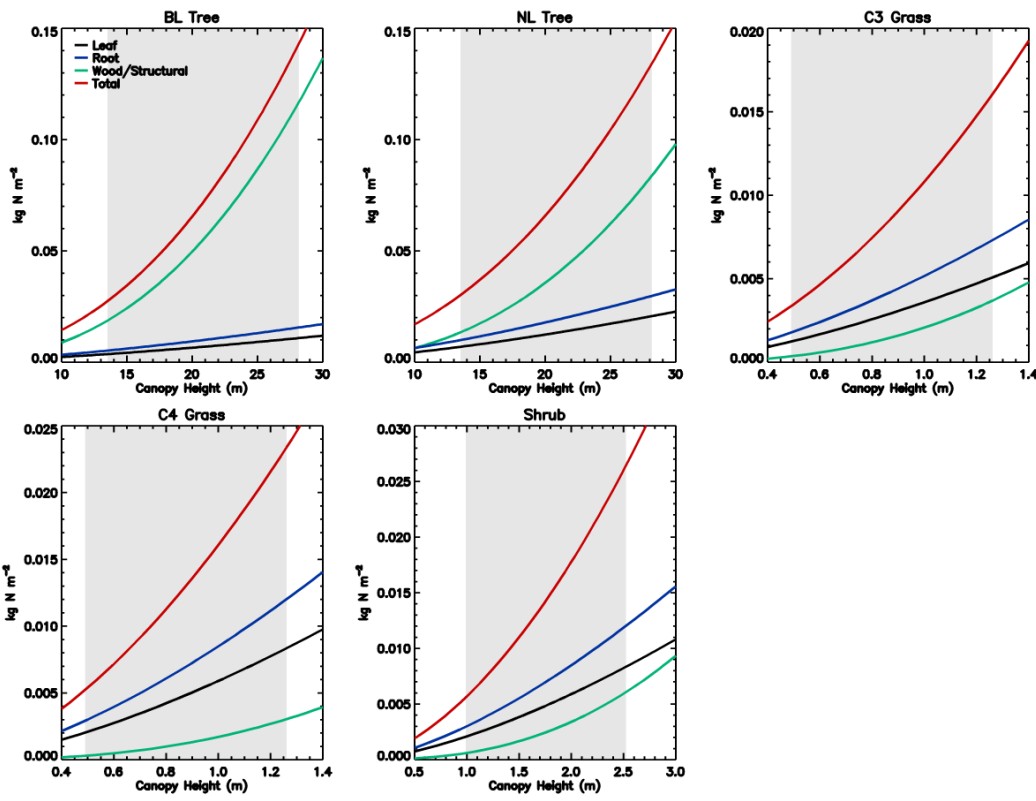

**Figure 2.** Total vegetation N along with N pools of leaf, root and wood as a function of canopy height for individual PFTs at full leaf. The grey region shows the defined range of canopy height within the model. Note: both the x- and y-scales are very different for each PFT.

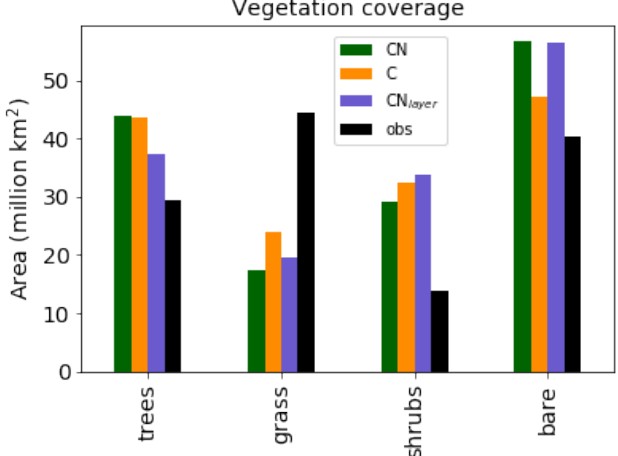

**Figure 3.** Total area covered by each vegetation type for the three different JULES configurations. The observations are derived from the European Space Agency (ESA) Climate Change Initiative (CCI) Land Cover data for 2010 Poulter et al. (2015) converted to JULES PFTs by Hartley et al. (2017).

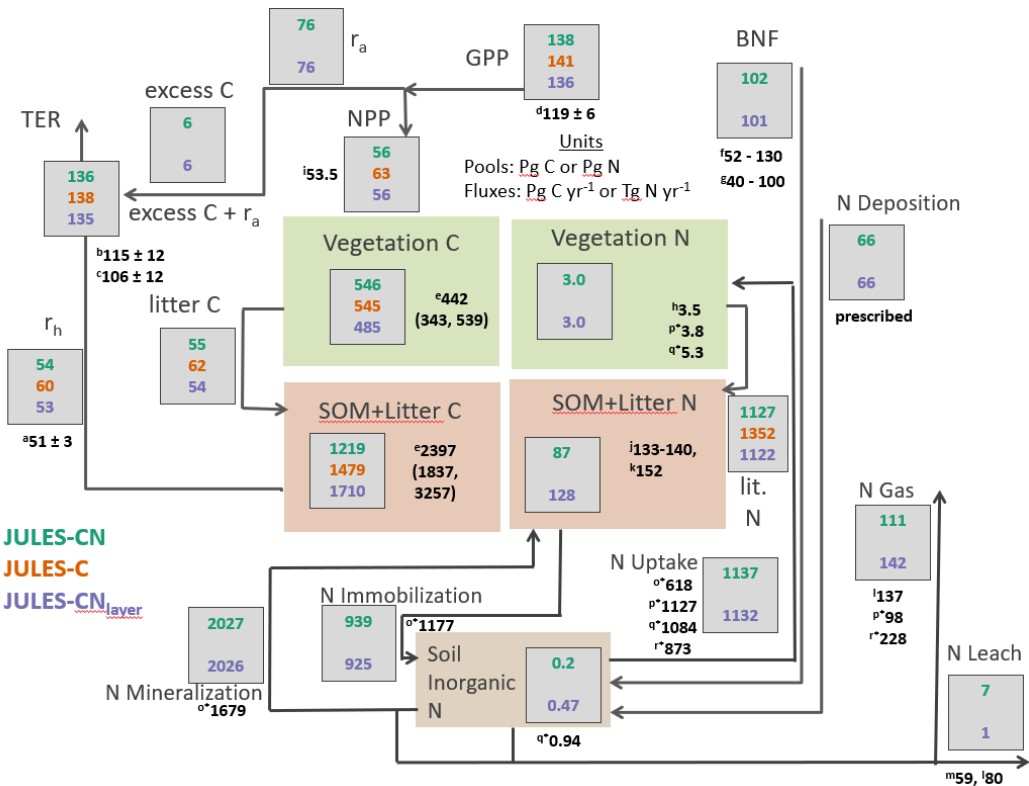

**Figure 4.** Carbon and nitrogen stocks and fluxes for JULES-CN, JULES-C, and JULES-CN$_{layer}$ for the period 1996-2005 (after Davies-Barnard et al. (2020)). C = Carbon; N = Nitrogen; r$_h$ = Heterotrophic respiration; r$_a$ = Autotrophic respiration; TER = Total ecosystem respiration; GPP = Gross primary productivity; NPP= Net primary productivity; SOM = Soil organic matter; $BNF$ = Biological N fixation; N gas is the sum of $N_{gas}$ and $N_{gasI}$ with $N_{gasI}$ representing approximately 90 % of the total gas loss. The black numbers are the observational-constrained values from the literature, where observational-based values are not available JULES is compared with other global models. (a) Heterotrophic respiration: Hashimoto et al. (2015); (b) TER: Li et al. (2018); (c) TER: Ballantyne et al. (2017); (d) GPP: Jung et al. (2011); (e) Vegetation carbon and SOM+litter carbon: Carvalhais et al. (2014); (f) BNF Davies-Barnard and Friedlingstein (2020); (g) BNF Vitousek et al. (2013); (h) Vegetation nitrogen: Schlesinger (1997); (i) NPP: Zhao and Running (2010); (j) soil organic nitrogen: Batjes (2014); (k) soil organic nitrogen: Group (2000); (l) nitrogen losses including nitrogen leaching: Gruber and Galloway (2008); (m) nitrogen leaching: Boyer et al. (2006); (n) nitrogen leaching: Galloway et al. (2004); (o*) organic nitrogen immobilisation and mineralisation and plant uptake von Bloh et al. (2018); (p*) nitrogen uptake, vegetation nitrogen and nitrogen emissions Zaehle et al. (2010); (q*) nitrogen uptake and inorganic nitrogen content Xu-Ri and Prentice (2008); and (r*) nitrogen uptake and total nitrogen emissions Wania et al. (2012). (o*), (p*), (q*) and (r*) are model derived estimates.

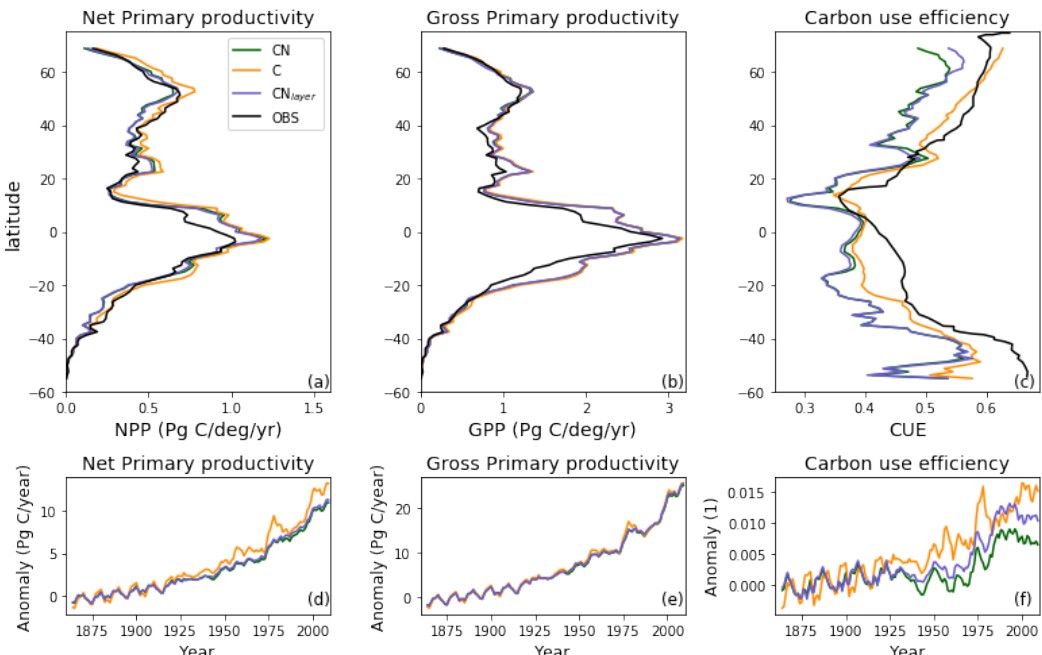

**Figure 5.** Zonal total values of (a) net primary productivity (NPP) and (b) gross primary productivity (GPP) for JULES-C, JULES-CN and JULES-CN$_{layer}$ simulations for the period 1996-2005 in Pg C / degree latitude / year. The observational-constraint for NPP is from MODIS (Zhao and Running, 2010) and that for GPP is from Jung et al. (2011). The zonal mean carbon use efficiency (CUE = NPP/GPP) is shown in (c). The CUE observational constraint was digitised from Kim et al. (2018). Also shown are changes in (d) NPP, (e) GPP and (f) CUE over the historical period with respect to the multi-annual mean period of 1860-1899.

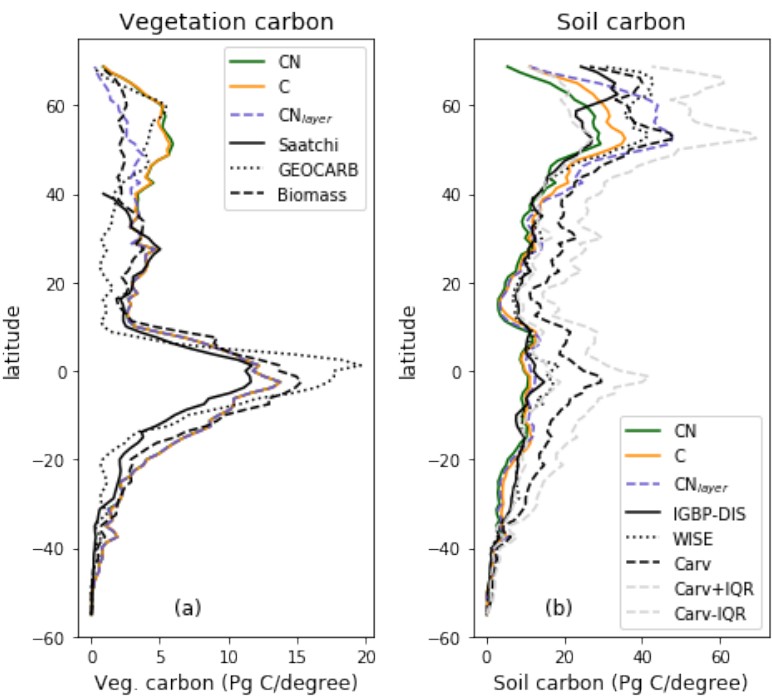

**Figure 6.** Zonal total values of (a) vegetation and (b) soil C for JULES-C, JULES-CN and JULES-CN$_{layer}$ simulations for the period 1996-2005 in Pg C / degree latitude. For the vegetation C the observational-based constrains are Saatchi: Saatchi et al. (2011); GEOCARB: Avitabile et al. (2016); and Biomass: Ruesch and Gibb. The observational-based constraints for the soil carbon are IGBP-DIS: Group (2000); WISE: Batjes (2016); and Carvahlais: Carvalhais et al. (2014).

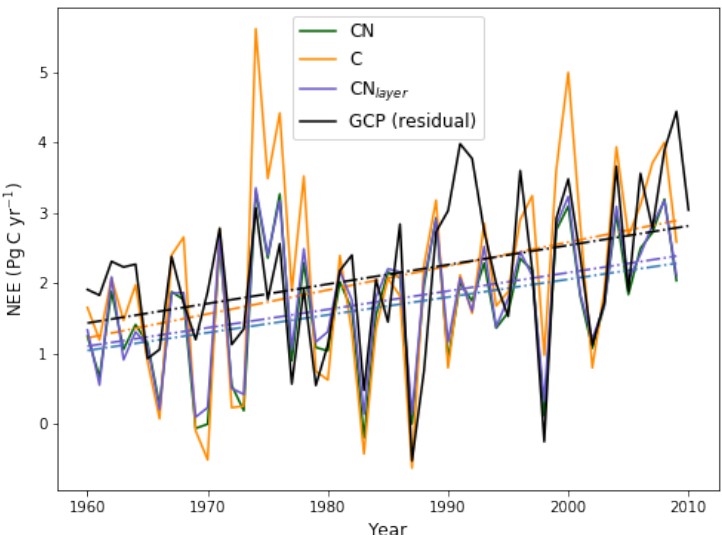

**Figure 7.** Evaluation of global annual mean NEE from JULES-CN, JULES-C and JULES-CN$_{layer}$ compared with observations based on estimates from GCP (Friedlingstein et al., 2020) over the period from 1960 to 2009 inclusive. Positive values represent the land surface as a net sink of carbon. The solid lines are the data and the dashed-dotted lines represent a linear fit of the data against time.

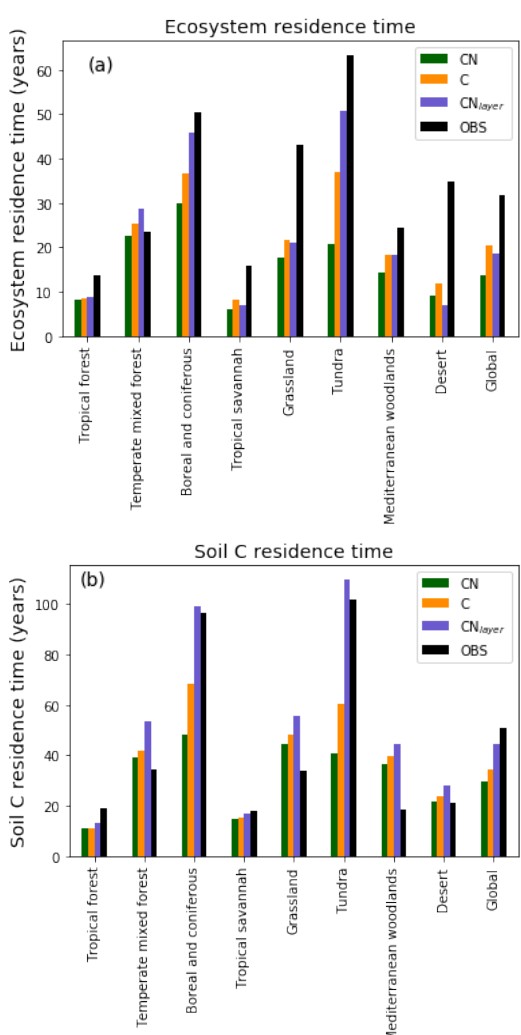

**Figure 8.** Biome-based (a) ecosystem turnover times and (b) soil carbon turnover times calculated on a grid-cell by grid-cell basis then aggregated temporally to biome level. JULES-C, JULES-CN and JULES-CN$_{layer}$ are shown for the period 1996-2005. The land surface is split into biomes based on the 14 World Wildlife Fund terrestrial ecoregions (Olson et al., 2001) and characterised by Harper et al. (2018). The observed ecosystem residence times are derived from the Carvalhais et al. (2014) global data set and the observed soil residence times are from the WISE: Batjes (2016) soil carbon combined with the Hashimoto et al. (2015) soil respiration.

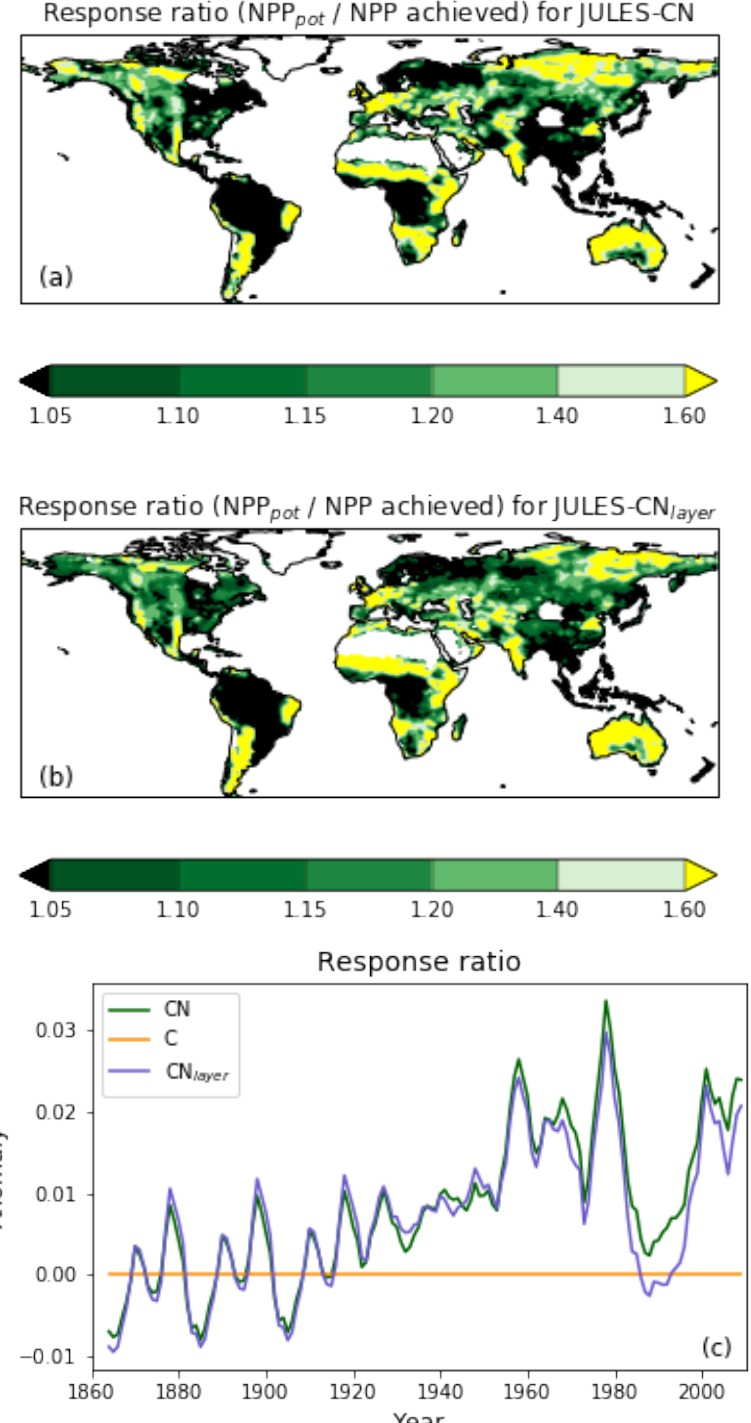

**Figure 9.** The spatial distribution of the response ratio defined as the potential amount of carbon that can be allocated to growth and spreading of the vegetation ($NPP_{pot}$) as a fraction of the NPP achieved in the natural state for (a) JULES-CN, and (b) JULES-CN$_{layer}$. A value greater than one means that the addition of nitrogen will enhance NPP. Any grid cells with an annual NPP of less than 0.016 g [C] m$^{-2}$ are set to missing. This is the spatial distribution of the metric shown in Figure 10. (c) shows the change in the response ratio over the historical period with respect to the multi-annual mean from the period of 1860-1899.

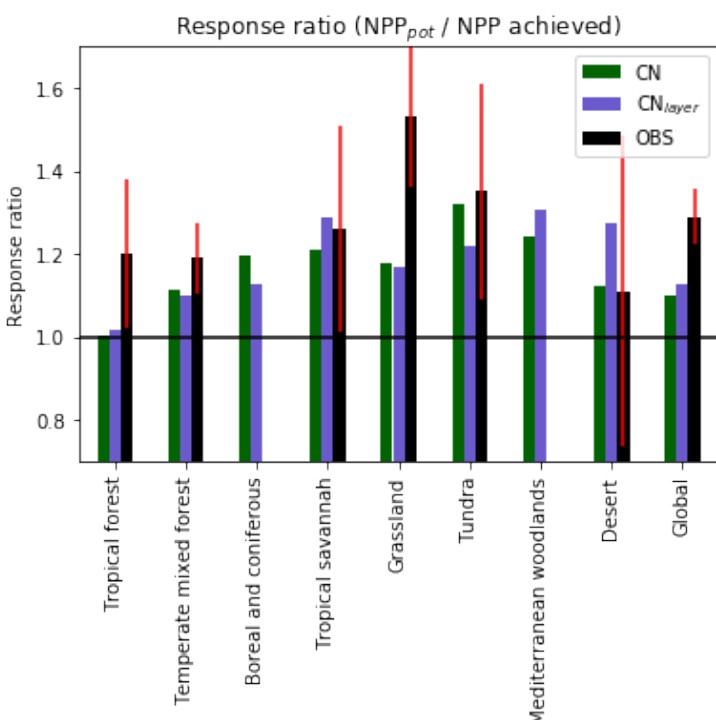

**Figure 10.** The response ratio is the ratio of the potential amount of carbon that can be allocated to growth and spreading of the vegetation ($NPP_{pot}$) compared with the actual amount achieved in the natural state (NPP). As in Figure 9, any grid cells with an annual NPP of less than 0.016 g [C] m$^{-2}$ are set to missing. The median of JULES-CN and JULES-CN$_{layer}$ are shown for each biome for the period 1996-2005. The biomes are discussed in more detail in Figure 8. The observational constraint is taken from Table 1 in LeBauer and Treseder (2008) which summarises a meta analysis of nitrogen addition experiments. The black bars show the mean of the observations and the red lines the uncertainty.

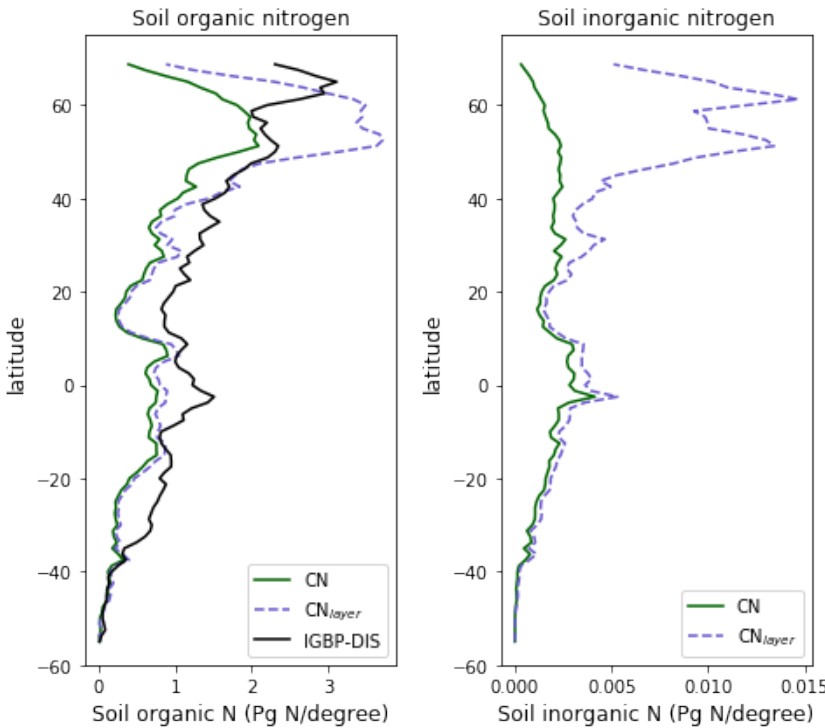

**Figure 11.** The zonal total soil organic and inorganic nitrogen stocks in Pg N / degree of latitude. JULES-CN$_{layer}$ shows the stocks for the top 1 m of soil. The observations of nitrogen stocks are from Group (2000).

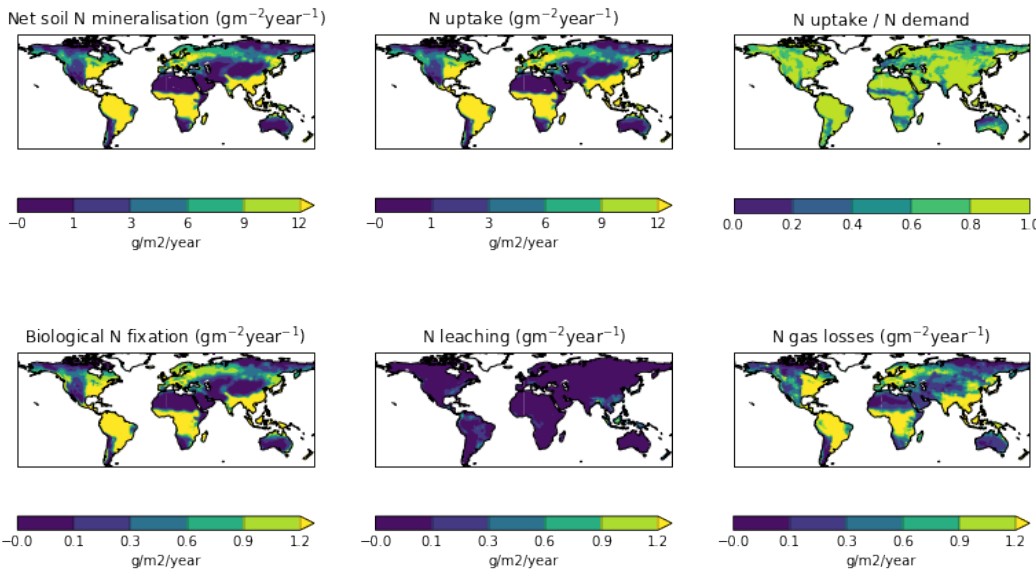

**Figure 12.** Spatial distribution of N fluxes for JULES-CN for the period 1996-2005. JULES-CN$_{layer}$ is not shown because the spatial patterns are very similar to those for JULES-CN.

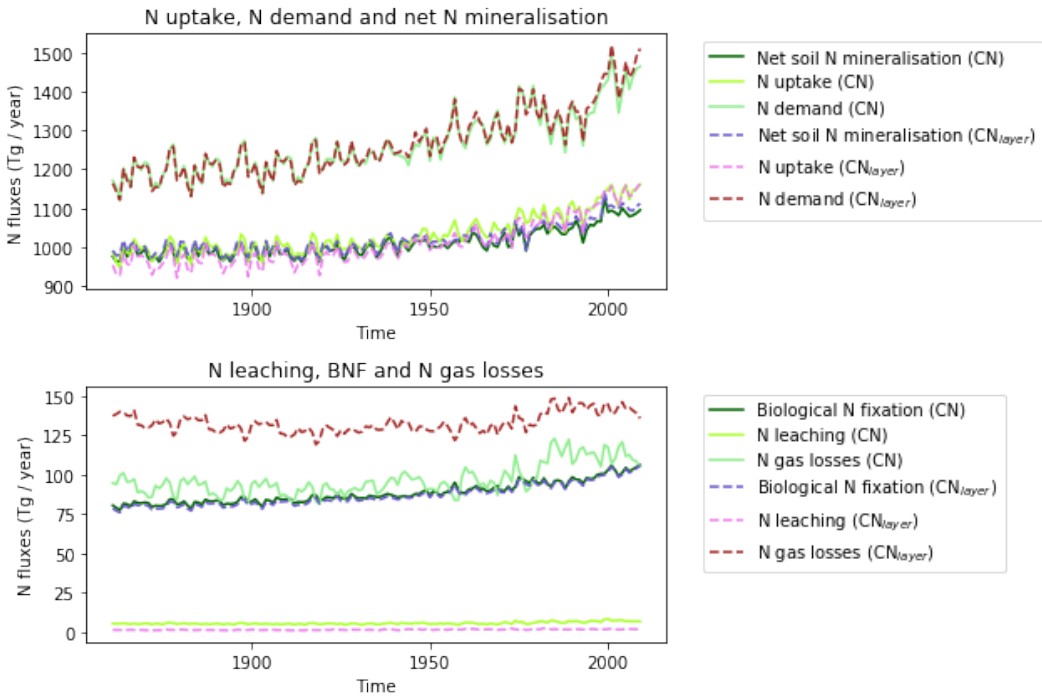

**Figure 13.** N fluxes for JULES-CN and JULES-CN$_{layer}$ over the historical period.