# Peer review of "JULES-CN: a coupled terrestrial Carbon-Nitrogen Scheme (JULES vn5.1)"

_Geoscientific Model Development, 2020_

## Referee Comment (RC1) · David Wårlind (Referee) · 17 Aug 2020

General comments

Introducing a prognostic nutrient cycle, here the nitrogen cycle, into a land surface model (LSM) is a challenging task. As the importance of nutrient limitation on productivity has been clear for a while and we have gone from one LSM with a prognostic N cycle in CMIP5 to several in CMIP6 this is a step all LSM are taking. So for undertaking this task and finishing an LSM that have included all the major N related processes I congratulate the authors. Some processes have been left quite simplistic (e.g. Ngas with its additional turnover) but this is a natural step in the process of developing a modelling framework. The paper goes through the steps they have taken to incorporate the

key terrestrial N cycle processes and show how different model setups behaves over historical simulations. These simulations have then been analysed on a global and biome scale and have shown that the model simulates the carbon and nitrogen pools and fluxes comparable to the limited available observations.

The main reason to include a prognostic nutrient cycle is to represent a limitation on plant productivity. The authors have shown that their N limitation is within observation on the biome level, but the global spatial distribution still puzzles me (see general comments). It would also be interesting to see how N limitation affect PFT distributions or at least some mention of it even if N limitation doesn't have any direct influence. In general, it would have been nice to see some perturbation experiments to see how the N cycle would react. Especially BNF and N limitation on productivity. But as this is covered in another paper (Davies-Barnard et al. 2020) it could have been good to refer to those results more than in just a short note in the introduction.

I think this is an excellent model description paper. All the relevant equations and model structures are well documented and described. I would like to congratulate the author to a job well done! Hope my comments will be to some help.

Specific comments

Section 3.1.1 – Biological Nitrogen Fixation feels misplaced in Section 3.1 Vegetation Carbon and Nitrogen. Would fit better in section 3.2 Soil Biogeochemistry together with other N sources and losses that are described here.

Section 3.1.3 – With eqn 12 and that z is the fraction of canopy above current layer, the canopy will always have the same C:N ratio and it will not depend on LAI as it was in Mercado et al. (2007). In Davies-Barnard et al. (2020) it is stated that leaves have flexible C:N ratio. How have I misunderstood this? Yes, leaves have flexible C:N ratios, but the canopy as a whole have a fixed C:N ratio. If the canopy C:N ratio is fixed then there will be a mismatch between canopy N and irradiance compared to Mercado et al. (2007) as irradiance will decrease exponentially through the canopy depending on LAI

but leaf N will not. Will this affect the photosynthesis?

L245-248: "If not enough inorganic nitrogen is available, the system is nitrogen limited and an additional term is required in the carbon balance representing excess carbon which cannot be assimilated into the plant due to lack of available nitrogen ($\Psi c$). A positive $\Psi c$ results in a reduction of carbon use efficiency." – N limitation only affects NPP and not GPP with an additional respiration term decreasing the CUE. As GPP isn't affected by N limitation then the water demand will stay the same. So the water "cost" for NPP will by higher in JULES compare to models that let N limitation directly affect GPP. Is this something that has been considered during the development?

L271: "The nitrogen available for growth is the total available nitrogen multiplied through by $(1 - \lambda)$." – I assume that the "nitrogen available for growth" is Navail and is used in L283. Navail isn't defined until L378. Please clarify this in the text.

Section 3.2.1 – Does litter and diffused SOM enter frozen soil layers? Could be the reason we see a higher soil C for CNlayer at higher latitudes (Figure 7).

L430-436: – The additional turnover of inorganic nitrogen is a great solution to a well-known issue when soil N starts building up uncontrollable due to N deposition or BNF.

Section 3.2 and 3.3 – A table with constants from sections 3.2 and 3.3 similar to Table 1 for section 3.1 would be a nice addition to the manuscript.

L532-534: – N leach is very small. Any idea why it is so small? Have you considered some adjustments to get the number to increase? Change the value of $\beta$?

L538-539 and Figure 4. – Net N mineralisation and N uptake seem to be very small. Are the units for them really Tg N yr-1?

L564-565: "This is a consequence of the higher nitrogen limitation on JULES-CN leading to less litter fall and subsequently less soil carbon." – I guess N limitation on SOM decomposition isn't strong enough to make the SOM pools increase in size? Could it be that the fixed plant C:N ratios prevent feedback of poorer litter quality under higher

N limitation that would result in a slowdown of SOM decomposition?

Figure 1. – Fixation seems to enter the vegetation in the figure, but section 3.1.1 says it enters inorganic N pool. Update figure.

Figure 6. – Is the increased soil C at high latitudes for CNlayer mainly due to the additional decay rate modifier per depth or is it due to N limitation on decomposition? Because with a lot less vegetation C the input of litter must also be less. So something else needs to dictate the build-up of soil C as this is opposite to what is stated in L564-565.

Figure 6, 7 and 9. – Figure 6 is the result we are after when introducing an N cycle, N limitation on productivity. The N limitation spatial distribution puzzles me to some extent. That you haven't investigated the reason for the strong N limitation in tropical savannah (L550-551 "Further work is required to understand why tropical savannah is so limited.") is something I think should have been done. And also that Northern Europe doesn't see any N limitation, but Western Europe does is also strange. I would have liked to have maps for figure 7 and 9 to try and understand this better, now a lot of information is hidden within the latitudinal bands. Also, a figure with annual net mineralisation would be of interest to understand what is happening.

Figure 6, 7 and 9. – How can it be that CNlayer has stronger N limitation at higher latitudes than CN (less Veg C in figure 7 and more yellow in figure 6) when there is more inorganic N in the soil (figure 9)? This needs to be explained better. Is it due to the root profile and that all N isn't available?

Technical corrections

L9: "Biological fixation and nitrogen deposition are external inputs..." – From section 3.1.1 it is clear that BNF isn't an external input. Please revise this sentence

L204-205: "We therefore a new parameterisation of retranslocation and labile nitrogen that is dependent on the phenological state" – please revise this sentence

L278: ". . . is is . . ." – remove one is.

L474: ". . . Equation51 . . ." – change to ". . . Equation 51 . . ."

L646: ". . . residence tome of carbon . . ." – change tome to time.

L675: " . . . model model . . ." – remove one model.

Figure 4. ". . . period 19960-2005 . . ." – correct to 1960.

References

Davies-Barnard, T., Meyerholt, J., Zaehle, S., Friedlingstein, P., Brovkin, V., Fan, Y., Fisher, R. A., Jones, C. D., Lee, H., Peano, D., Smith, B., Wårlind, D., and Wiltshire, A.: Nitrogen Cycling in CMIP6 Land Surface Models: Progress and Limitations, Biogeosciences Discuss., https://doi.org/10.5194/bg-2019-513, in review, 2020.

Mercado, L. M., Huntingford, C., Gash, J. H., Cox, P. M., and Jogireddy, V.: Improving the representation of radiation interception and photosynthesis for climate model applications, Tellus B, 59, 553–565, 2007, https://doi.org/10.1111/j.1600-0889.2007.00256.x.

---

## Referee Comment (RC2) · William Wieder (Referee) · 19 Aug 2020

Wiltshire and co-authors nicely document their additions of a nitrogen cycle and vertically resolved soil biogeochemical model to the JULES model for use in UKESM1. The offline simulations include documentation of simulated vegetation and soil carbon and N pools and fluxes and their change over the historical period. A comparison with some observations is provided for model evaluation

Major concerns

My major concerns aren't that substantial, but stem from contradictions in what's expected from the paper and what's actually delivered.

The paper sets off comparing the C only, CN and CN_Layered implementation of the

model, but a number of display items omit results from the CN_Layered configuration. Specifically, Fig. 10-12 & Table 2 do not show results from the layered model, why? Should these effects of vertical soils also be discussed in 5.3? Because these results are not presented, I think major revisions are warranted.

Are there meaningful differences in plant distributions simulated with the new N enabled or CN_layered models?

The multi-layered canopy model is introduced in section 3, but never really discussed in section 5. Should it be? Are there any interesting insights enabled by this new feature of the model?

Minor and technical concerns: These are more numerous, but intended to clarify and improve the paper.

I like the high-level overview of the main findings summarized in the abstract, but I wondered if more quantitative results should also be provided (pending length requirements for the journal)?

Paragraph starting on line 70. I appreciate how clearly model assumptions are laid out. For example, the approach here looks at the "large-scale role of nitrogen limitation on carbon use efficiency", but I wonder if there's evidence to support this common assumption made in models in real ecosystems? What is the assumed impact of N limitation on NEP? The net results it that is dampens

Can paragraphs around lines 60 & 90 effectively be combined? Both paragraphs seem to have a common purpose of documenting the model connections and history. It's also not really clear how JULES fits into UKESM (also called UKESM1) vs. HadGEM2

Is section 2 subheading really warranted? Maybe just combine subheadings for 2 & 3 into one longer section?

There are some redundancies in the text (section 3) where sentences are repeated at different points.

[Figure]

Line 162. I'm confused why "These stoichiometric functions already exist in the model" for MR fluxes. This suggests the new work here is just to explicitly represent the N pools that were being implicitly assumed in the carbon only model? Separately, is it worth documenting the source for vegetation stoichiometry (presumably used in Cox et al. 2011)?

Fig 1: The assumption that 'roots' in the model have a lower (or equal) C:N than leaves seems surprising to me, buth this but seems contradicted by 'Ratio of root to top leaf nitrogen' (Table 1), please clarify. Roots have wide variation in C:N (Iversen et al. 2017), but if anything I'd assume they should have a higher C:N ratio than leaves (Kattge et al. 2011).

Table 1: "Top leaf nitrogen concentration": listed twice

Line 175, this statement doesn't seem to be true for grasses, which have declining C:N with height (Fig 2).

Section 3.1.1, oh no, why define N fixation (which should limit NPP) as a function of NPP in the model?! This isn't the first modeling group to make this assumption, but a brief discussion and literature review seems warranted (see Vitousek et al. 2013; Thomas et al. 2105; Wieder et al. 2015; Meyerholt et al 2016)

Section 3.1.1- I think inputs from N fix lead off these details of the CN model because that's where the N cycle 'starts', which seems logical, but putting it under a "Vegetation carbon and nitrogen" (subheading 3.1) seems odd, especially since Nfix contributes to the soil N pool (not plants). Maybe different names for the higher level subheadings (3.1 and 3.2) would be warranted? Alternatively, use Fig 1 to group these fluxes together.

Line 182 What is potential NPP? (eq. 9). How does this different than the 'actual' NPP? If not discussed here, please reference where this is described (3.1.4).

Line 225 where is the multi-layer canopy approach included in these simulations? I'm assuming with with CNlayered, but this isn't clear in section 4 (line 495)

What is 'spreading' in the model (section 3.1.4)? Is this prescribed by some land use time series dataset or prognostic (more like a DGVM)? Text on page 11 makes me think it's the later.

The assumptions made in the phenology and allocation section are thoroughly defined, but it's hard to understand for readers not familiar with TRIFFID how N limitation is implemented in the model. It seems like it's an instantaneous down regulation of NPP, with extra carbon respired by plants that are N limited? With that N limitation calculated by the tissue and pft specific stoichiometry defined in the model?

Eq. 25-28. I don't really understand how the soil model is wired based on these equations. If R_DPM and R_RPM are the respiration terms from litter pools, how do some of these fluxes go back into the BIO and HUM pools, which themselves are respired (and also simultaneously included as inputs to BIO and HUM)? It seems that soil respiration fluxes to the atmosphere are actually R_tot*B_R, if so, the R_* fluxes should be some kind of soil turnover term (not respiration).

It seems like B_R is a critical number here, as it controls the soil carbon use efficiency and the amount of N required during litter decomposition (eq. 35). Is this parameter value defined somewhere?

Eq. 29-32 do the N fluxes need to include I_DPM + I_RPM?

Line 355, as above can this be called potential turnover, not "potential respiration"?

Eq. 33. I'm trying to wrap my head around the vegetation controls over decay rates and how that may feedback to a CN model that has vegetation with very different stoichiometry and N demand (woody vs. grass pfts; Fig 2) but that allows for plant competition (on a single soil column). I assumed the maps of nutrient limitation (Fig. 6) reflect differences in vegetation N demand (per unit of C), but are decay rates also slower for grasses (increasing the N limitation in these ecosystems)?

Eq. 36, Is this still a potential decomposition rate, as it's 'limited' by N availability?

Line 385, what are 'these two pools'? I think it should be DMP and RMP, but it's not clear in the text?

What happens to wood in the soil CN model? How is it allocated to the pools described?

Eq. 39, is there anything that prevents this flux from being negative? Are there times when immobilization > mineralization?

Eq. 39, Should the N loss description go into 3.3 (inorganic N) instead of the soils section (3.2)?

Where does N_turnover flux (eq. 46) go in the model, the atmosphere? How large is this tuning flux relative to other loss terms?

Eq. 46, where does N_gas (eq. 39) fit into the N budget summarized here?

Section 4, How does the model handle agricultural fractions of grid cells?

Section 5.1 I'm used to fluxes and pools being roughly proportional in models like this. If NPP is 11% lower in the CN model, why are the vegetation stocks roughly equal in the C and CN model? Similarly, if the vertically resolved model has a similar NPP to the CN model why are vegetation C pools so different?

Fig 4, 8 and others, Since the text is organized with C, CN, and CN_layered should the display items be similarly organized?

Fig 4 what is the 'N-loss' flux supposed to represent? As drawn, I think this is a gaseous N loss, but as labeled it's not clear how this connects with N_gas and N_turnover fluxes (see above).

Fig 4, how deep are the soils being represented, this is especially important to consider in the vertically resolved model and should likely be described in methods (3.2.1)

Line 535, doesn't this just mean the model is at equilibrium as it should be given your

spinup procedure?

Section 5.2 seems out of place, as the extent of N limitation should be preceded by a more thorough comparision of the model states and fluxes. One suggested organization could be comparing the 1) Spatial distribution of present day stocks / fluxes and residence times (e.g. Figs 4, 10, 7, 9, & 8 in that order) and 2) Temporal evolution of relevant stocks / fluxes over the historical period (e.g. Fig 11 & 12) 3) N limitation (Fig 6, 5) as diagnosed by NPP_Potential / NPP and its evolution over time (Fig 11b).

The title for Fig 5 (and associated text) implies that you conducted a N fertilization experiment (see Wieder et al. 2019), but I don't think this is accurate. Instead you're calculating a N limitation diagnostic (NPP_pot/NPP) and comparing that to results from an observational synthesis.

I'd suggest flipping the order of Figs 5 & 6, as they both show the same information, but Fig 6 is less processed model output, with 5 serving to summarize biome-specific information and related it to observations.

Fig 6, Line 553. It seems like the model is more strongly limited in grasslands, which have much higher N requirements / unit of C (Fig 2). This doesn't really show up in results for 'tundra' or 'grasslands' (only for Savannah). I wonder why?

Should multi-paneled figures be labeled ('a', 'b', 'c') and accordingly described in the figure caption?

Fig 7, can legends be smaller or moved into the figures (as in Fig. 9) so the data are easier to read?

Fig 9, should the bottom panel be labeled soil C residence time and also include data from the C-only model?

Fig. 11b what is the time series of 'response ratio anomaly'? Is this the change in NPP_Pot / NPP_act that used to diagnose N limitation shown in Figs 5 & 6? If so, is this what you're calling 'progressive N limitation' (line 599), in which case this should

be clarified on and expanded in the text.

Fig 12 & section 5.2.3 The low bias in NEE (∼0.5 Pg / y, roughly 25%). This would lead to an underestimation of the land carbon sink of about 25 Pg over the period from 1960-2010 (or about 12 ppm CO2 in the atmosphere). Thus, while the IAV of NEE looks better here, that overall magnitude of the land sink may be too low with the CN version of the model. This isn't a deal breaker for the paper but time implications of the low bias with the CN (and CN_layer) model should be discussed in the text, especially since JULES_CN is included in UKESM1.

Section 5.3. Is it just frozen soils that are causing this? it seems the differences in Veg C pools extend down to ∼ 40 degrees north (Fig 7). Is this somehow connected to assumptions about the fraction of N that plants have access to in the vertically resolved model (e.q. 51)?

Line 611, as noted in Fig 12, the low biases in land C uptake seems notable if you're trying to capture changes in the atmospheric CO2 growth rate.

Line 671 what is "climate-induced mineralization" I'm assuming this has something to do with accelerated decomposition from climate change increasing N mineralization rates?

Is there a data availability statement required for the journal? References: Iversen, C. M., McCormack, M. L., Powell, A. S., Blackwood, C. B., Freschet, G. T., Kattge, J., . . . Violle, C. (2017). A global Fine-Root Ecology Database to address below-ground challenges in plant ecology. New Phytologist, 215(1), 15-26. doi:10.1111/nph.14486

Meyerholt, J., Zaehle, S., & Smith, M. J. (2016). Variability of projected terrestrial biosphere responses to elevated levels of atmospheric CO2 due to uncertainty in biological nitrogen fixation. Biogeosciences, 13(5), 1491-1518. doi:10.5194/bg-13-1491-2016

Thomas, R. Q., Brookshire, E. N., & Gerber, S. (2015). Nitrogen limitation on land:

how can it occur in Earth system models? Glob Chang Biol, 21(5), 1777-1793. doi:10.1111/gcb.12813

Vitousek, P. M., Menge, D. N. L., Reed, S. C., & Cleveland, C. C. (2013). Biological nitrogen fixation: rates, patterns and ecological controls in terrestrial ecosystems. Philosophical Transactions of the Royal Society B: Biological Sciences, 368(1621). doi:10.1098/rstb.2013.0119

Wieder, W. R., Cleveland, C. C., Lawrence, D. M., & Bonan, G. B. (2015). Effects of model structural uncertainty on carbon cycle projections: biological nitrogen fixation as a case study. Environmental Research Letters, 10(4), 044016. doi:10.1088/1748-9326/10/4/044016

Wieder, W. R., Lawrence, D. M., Fisher, R. A., Bonan, G. B., Cheng, S. J., Goodale, C. L., . . . Thomas, R. Q. (2019). Beyond Static Benchmarking: Using Experimental Manipulations to Evaluate Land Model Assumptions. Global Biogeochem Cycles, 33(10), 1289-1309. doi:10.1029/2018GB006141
* * *

---

## Referee Comment (RC3) · Anonymous Referee #3 · 20 Aug 2020

This manuscript explains the N cycle in the JULES-CN model which forms the land component of the UKESM. Simulations from the UKESM have contributed to the CMIP6 effort. The N cycle component of JULES, as explained, here is very simple compared to existing models out there. This is completely acceptable as long as it is clarified that the model parameterizations are simple, their limitations acknowledged, and the implications discussed. I am afraid, however, that the manuscript doesn't appear to do so and in my mind requires substantial work to address this and other concerns I raise below.

[Figure]

**1 Major comments**

I have several major concerns.

1. It is well known that leaf N content is related to its photosynthesis capacity (Field and Mooney, 1986). When CO2 increases, photosynthesis increases but this rate of increase is slowed if enough N is not available. This process is referred to as photosynthesis downregulation (McGuire et al., 1995). So, it is clear then, that N limitation acts on photosynthesis and thus on the gross primary production (GPP) flux. However, the approach used in the manuscript, in contrast, reduces the NPP (without adjusting the GPP) which is equivalent to reducing carbon use efficiency (CUE = NPP/GPP). Since there is no biological justification for this provided, I am struggling to understand the reasoning behind this. Also, if that framework is still used, TRIFFID models Vcmax as a function of leaf N content (eqn. 51 in Cox 2001) so it makes sense to adjust Vcmax.

   Related to this concern, is the fact, that I am not able to find in the manuscript in detail how this reduction in NPP is implemented or how it results and because of the interaction of which processes. Unless I missed it, the only reference to this important process is made on line 78 as "... and then reducing plant net carbon gain to match available nutrients".

   It is well known that current observation-based CUE is around 0.5. This is also seen in Figure 10. The CUE for the JULES-CN model is lower than that for the JULES-C model because that's how JULES-CN is designed - to lower NPP and hence CUE as CO2 increases and N supply can't keep up. I am wondering what happens in a future simulation for RCP 8.5 scenario. Will your CUE drop down to something like 0.25 by year 2100 which seems totally unrealistic? This will be one implication of your model design since you have chosen to reduce NPP and not GPP.

2. The second big assumption in the model is that of fixed C:N ratios of plant tissues. The implications of this assumption are not discussed. Since C:N ratio of plants varies in space (as indicated by different values of $n_{l0}$ in Table 1) this indicates their ability to adapt to different environmental conditions in space. Assuming, plants can do the same in time as CO2 increases doesn't this imply that the assumption of fixed C:N is too strong and your model will limit NPP perhaps more excessively than it in the real world (with the caveat that in the real world GPP is constrained).

3. In context of model evaluation, it would have been extremely helpful to include a simulation in which N deposition is turned off. This simulation would have allowed to see if the effect of N deposition is indeed to increase NPP as would be intuitively expected.

   In addition, the TRENDY model simulation S2 doesn't take into account land use change and the fertilization of crops. Crop fertilization is a major source of leaching and gaseous emissions of $N_2O$ and $NO_x$. I am wondering if this is the possible reason that the simulated leaching in Figure 4 is so low compared to other estimates.

   Also, does the model simulate the realistic sign of response when driven with climate forcing only.

   Typically, a model's response to various forcings allows to see at least if the sign of the response is consistent with expectations.

4. I realize that there are very few observation-based estimates available for N related pools and fluxes. However, still there are plenty of quasi-observation and model based estimates against which model results could have been compared. For example, in Figure 4 there are no quasi-observed or model estimates for several quantities. Model estimates are, however, available for immobilization and mineralization (von Bloh et al. 2018), plant N uptake (Zaehle et al. 2010; Xu-Ri

and Prentice, 2008; Wania et al., 2012), and inorganic N mass (von Bloh et al. 2018; Xu-Ri and Prentice, 2008; Wania et al., 2012). These estimates will allow to put your model results in some context.

5. Model parameterizations are not compared to other models, and the conceptual basis of parameterizations and their implications, are not discussed (as mentioned above for the choice to reduce NPP and use fixed C:N ratios) .

   For example, biological nitrogen fixation (BNF) is modelled as a straight-forward function of NPP. This is okay but the manuscript doesn't note that meta-analysis studies have found that BNF increases with increasing CO2 (Liang et al., 2016) but decreases with increasing N deposition and fertilizer application (Ochoa-Hueso et al., 2013) both of which apparently result in increase in NPP. In addition, BNF is typically higher over agricultural areas.

   Similarly, all gaseous losses are expressed using $N_{turnover}$ but in nature there are several pathways using which gaseous losses occur. $N_2O$ and $NO_x$ losses occur during nitrification (via nitrifer denitrification) and $N_2$, $N_2O$, and $NO_x$ losses occur during denitrification.

   It would be scientifically beneficial for the manuscript, and for a reader, if simplifications made are clearly highlighted and their limitations discussed, because then it is possible to interpret the model results in light of these limitations.

6. The majority of the results shown in the manuscript focus on the ability of the new model to reproduce all the aspects of the C cycle as the previous model did. As a result, the N cycle module is not evaluated rigorously. The manuscript doesn't report N demand, how it changes over time, what part of the N demand is not met, what part of N demand due to increasing CO2 is met by N deposition, time series of mineral N pool, time series of plant N uptake, time series of C:N ratio of whole plant and other plant components, and geographical distribution of simulated C:N ratios (even though I realize they are specified). Since this is the

first time JULES' N cycle component is being published it is reasonable to expect that such a manuscript will rigorously assess the new N cycle module.

7. There is no mention of phosphorus cycle at all. It is well know that in the tropics phosphorus limits photosynthesis and not nitrogen. How is this accounted for? My guess is this is somehow built into the Vcmax rates which are function of leaf N content (eqn. 51 in Cox 2001). If the model can reproduce correct zonal distribution of GPP it must take phosphorus limitation in the tropics somehow into account.

8. Finally, the lack of units, the lack of rate change equations for several pools, and unclear statements make it difficult to understand the model parameterizations as noted below in minor comments. In its current form, there is no way a reader can fully understand and reproduce the parameterizations reported here in some other model.

**2  Minor comments**

9. Abstract, line 8, "It represents all the key terrestrial nitrogen processes in an efficient way.". The word "efficient" here is misleading.

10. Abstract, line 9, I find it extremely confusing that BNF is mentioned as an external input. BNF is how N enters the coupled vegetation and soil system. Consider the case, if we were to refer GPP as an external input since that's how C enters the coupled vegetation and soil system. N deposition and fertilizer, on the other hand, can be called external because they are not natural just like fossil fuel emissions.

On page 2, in addition to BNF, leaching is also referred to as an external (loss). This also seems strange since on the carbon cycle side we don't refer to heterotrophic respiration or dissolved inorganic C in runoff as external losses.

11. Page 2, line 34, "Internally organic N is lost ...". Here "internally", perhaps is much better described as "cycling of N within the coupled vegetation and soil system".

12. Page 2, line 36, "Both inorganic and organic nitrogen may become available for plant uptake". Since organic N uptake is very small and therefore not even modelled (including in your model) perhaps it would be better if this is clarified.

13. Page 2, line 39. "In a changing climate, rising atmospheric CO2 drives **an increase in the terrestrial carbon cycle** and Gross Primary Productivity (GPP)." This is a vague sentence. What does "an increase in the terrestrial carbon cycle" means?

14. Page 2, line 56, " ... are between a reduction of 39 % and a slight increase of 1 % ...". Please consider rewording this sentence/phrase. It is somewhat hard to follow.

15. Page 3, line 65. " ... and a new managed land module ...". Please consider rewording to "and a new module for land management ...".

16. Page 3, lines 72-74. "This is achieved by extending the implicit representation of nitrogen in the existing dynamic vegetation and plant physiology modules TO ENABLE A MORE COMPREHENSIVE NITROGEN CYCLE WITHIN THE LAND SURFACE". Please consider deleting the text in capitals given N cycle framework used here is extremely simplified.

17. Page 3, Lines 74-75. "Nutrient limitation operates through two mechanisms; the available carbon for growth and spreading is reduced and the decomposition of litter carbon into the soil carbon is slowed". The word spreading at this point in the manuscript is unclear. Only after reading the rest of the manuscript it is clear that "spreading" means changes in the spatial extent of vegetation. Please consider using another phrase/word to replace "spreading".

Please also consider not using the phrase "decomposition of litter carbon into the soil carbon" here and elsewhere. Technically litter doesn't decomposes into soil carbon. As litter decomposes it releases CO2 and the dead organic matter is broken into smaller more recalcitrant materials, which the models consider as soil carbon. In reality, of course, there is a continuum.

18. Page 4, lines 114-115. "As standard, JULES-C includes an implicit representation of nitrogen which has been extended to be fully interactive.". A sentence or two about how nutrient constraints on photosynthesis are implicitly modelled in JULES-C will be helpful.

19. Page 4, line 120. "The vegetation nitrogen component captures the nitrogen limitation **on the C stock**, and ...". As described here the N limitation acts on NPP which is a C flux and not on the C stock.

20. Page 4, last sentence, line 126. " ... it slows the rate of litter decomposition INTO SOIL ORGANIC MATTER." Please consider removing the phrase in capitals.

21. Page 5, lines 129-135. I felt, it is little too early to introduce the seven JULES-CN parameters given that at this point in the manuscript, the parameterization themselves haven't been introduced.

    Also on line 130, Does " ... the effective solubility of nitrogen", refers to solubility in water.

22. Section 3.1. It seems the model's roots are in fact fine roots (since $R_c = L_c$ in eqn(3)), and coarse roots and stem are included in the $W_c$ term. Please make this clear.

23. For eqn (1) please specify the units of all terms. I suspect these are KgC m$^{-2}$.

24. For eqn (2) what are the units of $\sigma_l$ and L$_b$.

25. What are units of the individual terms in eqn (5) through (9) and the remaining equations.

26. Page 6, lines 160-178. This entire section is based on Figures 2 and 3 which form the backbone of specified C:N ratios and their variation with canopy height. It would be extremely helpful to know the basis of these relationships.

27. Page 6, lines 180-181. "Biological nitrogen fixation (BNF) is ASSUMED TO BE THE largest **natural** supplier of nitrogen to the terrestrial ecosystem". Consider removing the words in capitals and including the word in bold. Fertilizer application is the largest anthropogenic N flux and BNF is largest natural flux.

28. Page 6, line 181. "Following Cleveland et al. (1999), the nitrogen fixation is determined as a proportion of the net primary production before nitrogen limitation ($NPP_{pot}$)". This is incorrect. Cleveland et al. (1999) parameterized BNF as a function of actual evapotranspiration (AET) not NPP.

    Also, $NPP_{pot}$ is not defined anywhere close to this equation where it is introduced the first time. The first definition of $NPP_{pot}$ occurs on page 9, line 242, as "$NPP_{pot}$ supplied to TRIFFID represents the potential amount of carbon that can be allocated to growth". Then a somewhat different definition occurs on page 19 which defines $NPP_{pot}$ as the NPP when nitrogen is unlimited. Isn't $NPP_{pot}$ just the NPP from the original framework without any reduction. I don't think, you do a calculation with unlimited N applied, per se.

    In context of BNF, and eqn (9), the parameter $\zeta$ is not listed in Table 1.

29. Page 7, Table 1. It would be extremely help if $n_{l0}$ is inverted and written as $\frac{1}{n_{l0}}$ in units of Kg C/Kg N so that the values are easily comparable to C:N ratios reported in literature.

    Also, $n_{l0}$ is listed twice in Table 1 and please consider rewording "Top leaf N concentration" to "N concentration at the canopy top".

30. Page 7, lines 188-189. "However, in JULES-CN$_{layered}$ the vertical distribution of the **fixed** nitrogen in the soil **depends on the root distribution** ...". What does "fixed" refers to in this context?

    Also, at this stage in the manuscript it is not clear what does "depends on root distribution" means?

31. Page 7, lines 201-203. "This distinction is inconsequential in the carbon only mode but is more critical when considering nitrogen interactions as the implication is that at all times the plant has enough nitrogen in reserve to maintain full leaf". From here on it becomes difficult to follow the logic used in the model. I am not able to understand what does "the plant has enough nitrogen in reserve to maintain full leaf" means?

32. Page 8, eqn. (10). I am confused here. $L_b$ is introduced as a variable called **balanced leaf area index** but not explained what actually it means. In eqn (2), leaf C, $L_C$ is a function of $L_b$. In equation (9), leaf area index (LAI) (L) is also related to $L_b$ through $p$. Somewhere here, there is the split of $L_C$ into labile and non-labile (the one which determines the actual LAI). Did I get this correct? How are $L$ in eqn. (9) and $L_C$ in eqn. (2) related? Are they related through specific leaf area (SLA)?

33. All through up to this point in the manuscript, the rate change equations for the vegetation N pool are not presented. At this point in the manuscript, I am still unclear what "retranslocation" means. Is this the transfer of resorbed N from leaves before they are shed. If yes, to which plant components?

34. page 8, lines 251-216. "During leaf-off the labile component is the equivalent of the retranslocated leaf nitrogen **plus an additional store of nitrogen** in preparation for the following bud burst". This sentence introduces yet another pool. It would be really helpful if all the pools and their rate change equations are properly introduced.

35. Page 8, line 29. "The mean canopy nitrogen content is described by ...". Please reword this to "The vertical distribution of leaf N content in the canopy is described by ...".

36. Page 9, line 235. "Canopy Leaf C:N ratios are resultingly 44% higher than top leaf ratios". I am unable to understand this. Does "canopy leaf C:N ratios" refers to mean canopy leaf C:N ratio or the vertical profile of C:N ratios along the canopy depth starting from the top.

    If leaf N content in the leaves at the top of the canopy is higher and decreases exponentially, and if C content is uniform than it implies that C:N ratio of leaves is lower in the leaves at the top of the canopy and higher at the bottom. Integrating eqn (12) over LAI yields

    $$\int_0^L n_{l0}\, exp(-k_n z)\, dz = n_{l0} \frac{1}{k_n}\left(1 - exp(-k_n L)\right)$$

    which implies that the mean C:N will depend on the LAI, L. So I am unclear where does the number 44% comes from.

37. Page 9. Section 3.1.4. The term $\Lambda_{lc}$ in eqn. (13) is not defined and only when the reader reaches eqn. (21) it is clear what this term is. Similarly for $\Lambda_{ln}$.

38. Page 10. Line 263. $\Lambda_{ln}$ is defined as the retranslocation of nitrogen from leaves and roots into the plant labile pool. I am not sure how does it relate to $p$ in equation 10 which is also related to retranslocation.

39. Where is $\Psi_c$ from equation (13) defined? Is this what $\Psi$ is in eqn. (17)?

40. Page 10. Line 271. "The nitrogen available for growth is the total available nitrogen multiplied ...". Please reword this as "The nitrogen uptake used for plant growth is the total nitrogen uptake multiplied ...". I think, that's what is meant

here. Available N sounds like the N available in the soil inorganic pool that can be potentially taken up by plants.

41. Page 10. Line 272. "Equations 13 and 15 are then solved by bisection such that the nitrogen uptake for growth ($\Phi_g$) is less than or equal to the available nitrogen ...". Do you mean the bisection method to find root of an equation? This and remaining part of this paragraph is difficult to understand since there is no $\Phi_g$ term in either equation (13) or (15).

   In addition, since units of the various terms are not provided it is difficult to follow the equations on page 10.

42. Page 10, line 282. "... and Nv/Cv defines the whole plant C:N ratio ...". You mean Cv/Nv?

43. In the absence of the competition module of the TRIFFID model properly described it is difficult for a reader to know what does "density-dependent litter production" and "density-dependent componennt for intra-PFT competition for space" means in Section 3.1.5. Please consider introducing this in a sentence or two.

44. Page 11, please define $\Lambda_c$ and $\Lambda_n$ in words explicitly where the are first introduced. $\Lambda_c$ was introduced in equation (22) but not defined until next page near eqn. (28).

45. Page 11, lines 310-311. "The effect of nitrogen limitation on the litter carbon flux is captured in the excess carbon term $\Psi_i$". Throughout the manuscript there is no expression for $\Psi_i$ so it's difficult to understand it. I do understand based on what is written in the manuscript that it the excess C that cannot be used. So it must be related to N uptake, allocation fractions for C, and specified C:N ratio of the three C pools.
[Figure]

46. Page 12, line 339. "$\beta_R$ depends on soil texture". I don't think, this dependence can be too strong. Can you please mention the typical value of $\beta_R$.

47. The rate change equations for litter and soil C pools are helpful. Similar equations for vegetation C and N pools would be so helpful.

48. Page 13, line 349-350. "Input into the BIO and HUM nitrogen pools comes from the total immobilisation of inorganic nitrogen into organic nitrogen where $I_{tot} = I_{DPM} + I_{RPM} + I_{BIO} + I_{HUM}$".

    $I_{tot}$ is divided into BIO and HUM pools. Since BIO is the microbial pool shouldn't all immobilization end up there.

49. Page 13, eqn (33). Does the subscript $i$ still refers to PFTs?

50. Page 13, line 365. " ... the respired fraction ($\beta_R$) and the C to N ratio of the destination pool ...". This is confusing since on line 339 (1-$\beta_R$) was referred to as "the fraction of soil respiration that is emitted to the atmosphere".

51. Page 14, line 371. " ... where $i$ is one of RPM or DPM." Please use a different subscript here since you have used $i$ previously to represent PFTs.

52. Pages 13 and 14. The $F_N$ terms in eqn (36) limits the respiration of the DPM abd RPM litter pools. So it is unclear to me why $F_N$ would depend on $I_{BIO}$ and $I_{HUM}$ in eqn. (37). In this same equation, I am also unclear what is $N_{avail}$ at this point in the manuscript. As with several other terms, the terms are introduced but their expressions are mentioned or the terms clarified much later which makes it very difficult to follow the logic. It is only further down in eqn. (51) that $N_{avail}$ is clarified.

    Also, in eqn. (37) what happens if $D_{DPM}$ or $D_{RPM}$ are negative? Is this possible, since minrealization can be more than immobilization?

53. Page 14. Similarly in eqn. (39) can $I_{tot}$ be more than $M_{tot}$ making $N_{gas}$ negative.

54. Page 15. Eqn.(41). Is $f_{dpm}$ used here different from $f_{DPM}$ used in eqn. (25).

55. Page 15. Lines 416-417."The litter inputs are distributed so that they decline exponentially with depth, with an e-folding depth of 0.2 m".

    With this parameterization can litter enter a soil layer even if there are no roots in that layer.

56. Page 15. Line 423. Please consider using "bulk" or "single layer" instead of "non-layered".

57. Please consider using another term for gaseous losses rather than turnover.

58. Page 16, Lines 433-434."Without this additional turnover available N may increase excessively, potentially due to excessive biological fixation **in regions that are generally unlimited**". What does "regions that are generally unlimited" means?

59. Page 16. Line 434-435. "In the current model configuration this parameter is set to 1.0 (360 day-1) such that the whole pool turns over once every model year".

    Do you mean 1.0 year$^{-1}$ which would translate to (1/360) day$^{-1}$ and not 360 day$^{-1}$? Also, since the time step of the biogeochemistry is the same as for TRIFFID (i.e. 10 days) there has to be $\Delta T$ somewhere. And, I suspect, 360 is used and not 365 since the calendar year in the UKESM model is 360 days. Correct?

60. Page 16, line 436. "This results in an effective saturation limit of 0.002 KgN m$^{-2}$ ...". Not clear - saturation limit of what?

61. What are the units of $\beta$ in eqn. (47). Just above eqn. (47) $\beta$ is said to be assigned "a value of 0.1 based on sorption buffer coefficient of Ammonia although here it

represents the sorption of all inorganic nitrogen species". Note here that typically only NO3- leaches into the runoff and not NH4+ so please consider modifying this sentence.

62. Page 16. Eqn. (48). Isn't $f_1$ simply the fraction of roots in each soil layer. And again, $f_2$ is not defined or described here but further down in eqn. (53).

63. Page 16. Line 453. "where $f_{root}(z)$ is the volumetric root fraction at a given depth". You mean "for a given soil layer" as opposed to "at a given depth". And, an $i$ subscript seems to be missing here. Although, I wouldn't suggest using $i$ which has been used for PFTs, DPM or RPM, and now soil layers. Very confusing!

64. Page 17, eqn. (50). Is the parameter $\tau_{resp}$ tuned so that $N_{turnover}$ is similar in the "bulk" and "layered" versions.

65. Page 17. Eqn (51). Assuming, the subscript $i$ represents the PFT shouldn't there be (z) term here to indicate the nitrogen availability in each layer.

66. Page 17. Eqn (52). I am unable to follow eqn. (51). Looking at eqn. (51) the term in parantheses in eqn. (52) should be zero since $f_{root,i}\ N_{in} = N_{avail,i}$ from eqn. (51).

    The value/units of $\gamma_{diff}$ is also confusing. I am not sure what 100 [360 day]$^{-1}$ means.

67. Page 17, lines 471-474. "Any fixation goes directly into the available pool, and other fluxes are simply added according to the ratio of the available to total inorganic N pools at equilibrium (thus the available pool would always follow Equation 51 were it not for the fixation and uptake by plants)". I am sorry but I am unable to follow this sentence.

68. Page 17. Eqn. (54). In the absence of its units, I am not sure if the term $dz_n$ is a single variable or do you mean $\Delta z_n$. And, I have no clue, what $z_n$ is at this point in the manuscript.

69. Page 17. Line 483. "... is then the sum of all nitrogen that leaves the soil by lateral runoff ...". Does the lateral runoff from each layer mean that JULES is capable of producing runoff based on slope of the ground? Please clarify what exactly lateral runoff means.

70. Page 18. Lines 501-502. "They were spun up by repeating the time period 1860-1870 ...". This is confusing. Please consider rewording as "The models were spun up by using the meteorological data for the period 1860-1870 repeatedly ..."

71. Page 19. Lines 522-524. "The main difference is the present-day NPP which is 12% higher in JULES-C than in JULES-CN. **This is a direct consequence of nitrogen limitation which restricts the ability of the plants to utilise all of the carbon**". No this is the direct consequence of JULES-CN reducing NPP. I don't think, it is necessary to spin this in a more biological way.

72. In Figure 4, it would be really useful to see seperate estimates for mineralization and immobilization. In it current form, only net mineralization is reported.

73. Page 20. Lines 580-582. "This [CUE] represents the capacity of the plants to allocate carbon from photosynthesis to the terrestrial biomass". I don't think this sentence is entirely correct. Since CUE is the fraction of GPP converted to NPP, it is a measure of autotrophic respiration.

74. Page 20, line 582-583. "In the model nitrogen limitation restricts the ability of plants to allocate carbon and reduces the carbon use efficiency". Here again, the "restriction of ability of plants to allocate carbon" appears as if carbon is there but

some how plants can not allocate it. In contrast, as JULES-CN is designed, there is simply less carbon to be allocated. I don't think, JULES' allocation module has been changed in JULES-CN to limit how much C flows to different components.

75. Page 21. Line 596. " ... by structural changes in the vegetation in particular ...". Please clarify if structural changes refer to changes in vegetation height, LAI, and rooting depth.

76. Page 22. Lines 626-628. "There remains a significant underestimate of NEE in the years following the Pinatubo volcanic eruption ...". Please make it explicit in which year Pinatubo erupted since it's not marked in Figure 12.

77. Page 22. Line 646. Please change "tome" to "time".

78. Page 23. Line 656. "In this model, nitrogen limitation affects NPP and how the carbon is allocated ...". As mentioned above, I think, it's more appropriate to say **how much C is allocated** since the underlying C allocation module has not changed between JULES-C and JULES-CN.

**3  References**

1. Cox, P. (2001). Description of the "TRIFFID" Dynamic Global Vegetation Model. Met Office, Hadley Centre Technical 24.

2. Field, C. and Mooney, H.: The Photosynthesis-Nitrogen Relationship in Wild Plants, Biol. Int., 13, 25–56, 1986.

3. Liang, J., Qi, X., Souza, L. and Luo, Y.: Processes regulating progressive nitrogen limitation under elevated carbon dioxide: a meta-analysis, Biogeosciences, 13(9), 2689–2699, doi:10.5194/bg-13-2689-2016, 2016.

4. McGuire, A. D., Melillo, J. M. and Joyce, L. A.: The role of nitrogen in the response of forests net primary production to elevated atmospheric carbon dioxide, Annual Reviews Ecol. Syst., 26(1), 473–503, doi:10.1146/annurev.es.26.110195.002353, 1995.

5. Ochoa-Hueso, R., Maestre, F. T., Ríos, A. [de los, Valea, S., Theobald, M. R., Vivanco, M. G., Manrique, E. and Bowker, M. A.: Nitrogen deposition alters nitrogen cycling and reduces soil carbon content in low-productivity semiarid Mediterranean ecosystems, Environ. Pollut., 179, 185–193, doi:https://doi.org/10.1016/j.envpol.2013.03.060, 2013.

---

## Author Comment (AC1) · 3 Nov 2020

David Wårlind (Referee)

david.warlind@nateko.lu.se

General comments

Introducing a prognostic nutrient cycle, here the nitrogen cycle, into a land surface model (LSM) is a challenging task. As the importance of nutrient limitation on produc- tivity has been clear for a while and we have gone from one LSM with a prognostic N cycle in CMIP5 to several in CMIP6 this is a step all LSM are taking. So for undertaking this task and finishing an LSM that have included all the major N related processes I congratulate the authors. Some processes have been left quite simplistic (e.g. Ngas with its additional turnover) but this is a natural step in the process of developing a modelling framework. The paper goes through the steps they have taken to incorporate the key terrestrial N cycle processes and show how different model setups behaves over historical simulations. These simulations have then been analysed on a global and biome scale and have shown that the model simulates the carbon and nitrogen pools and fluxes comparable to the limited available observations.

The main reason to include a prognostic nutrient cycle is to represent a limitation on plant productivity. The authors have shown that their N limitation is within observation on the biome level, but the global spatial distribution still puzzles me (see general comments). It would also be interesting to see how N limitation affect PFT distributions or at least some mention of it even if N limitation doesn't have any direct influence. In general, it would have been nice to see some perturbation experiments to see how the N cycle would react. Especially BNF and N limitation on productivity. But as this is covered in another paper (Davies-Barnard et al. 2020) it could have been good to refer to those results more than in just a short note in the introduction.

We have added a figure showing the fractional distribution of the vegetation and how it changes with the different configurations. We have also extended the discussion section to include next steps and a description of the results in the Davies-Barnard paper.

I think this is an excellent model description paper. All the relevant equations and model structures are well documented and described. I would like to congratulate the author to a job well done! Hope my comments will be to some help.

Thank you for your helpful review comments. As you note we have endeavoured to develop a parsimonious scheme for application in the UK Earth System Model. This is a first step in enabling further representation of the role of nutrients including fully coupling with gas phase chemistry.

In revision we will include reference to the Davies-Barnard paper and other relevant results from CMIP experiments.

Specific comments

Section 3.1.1 – Biological Nitrogen Fixation feels misplaced in Section 3.1 Vegetation Carbon and Nitrogen. Would fit better in section 3.2 Soil Biogeochemistry together with other N sources and losses that are described here.

This has been moved to the soil inorganic nitrogen section and sign posted earlier on in the text.

Section 3.1.3 – With eqn 12 and that z is the fraction of canopy above current layer, the canopy will always have the same C:N ratio and it will not depend on LAI as it was in Mercado et al. (2007). In Davies-Barnard et al. (2020) it is stated that leaves have flexible C:N ratio. How have I misunderstood this? Yes, leaves have flexible C:N ratios, but the canopy as a whole have a fixed C:N ratio. If the canopy C:N ratio is fixed then there will be a mismatch between canopy N and irradiance compared to Mercado et al. (2007) as irradiance will decrease exponentially through the canopy depending on LAI but leaf N will not. Will this affect the photosynthesis?

Thank you pointing out the issue. In Davies-Barnard, there is an error, in that leaves have a variable C:N ratio with canopy depth, which is not the same as flexible stoichiometry. We will endeavour to correct this in the Davies-Barnard paper through a correction.

Agreed, there is a mismatch between canopy N and irradiance in the current formulation. This is being investigated and will be documented separately and addressed in subsequent configuration updates.

L245-248: "If not enough inorganic nitrogen is available, the system is nitrogen limited and an additional term is required in the carbon balance representing excess carbon which cannot be assimilated into the plant due to lack of available nitrogen ($\Psi c$). A positive $\Psi c$ results in a reduction of carbon use efficiency." – N limitation only affects NPP and not GPP with an additional respiration term decreasing the CUE. As GPP isn't affected by N limitation then the water demand will stay the same. So the water "cost" for NPP will by higher in JULES compare to models that let N limitation directly affect GPP. Is this something that has been considered during the development?

You are correct that N limitation doesn't directly impact water demand. However, there is an in-direct affect via the coupling between N limitation and LAI. This is something we are aware of and will be taking into account in analysis of CMIP experiments and future model developments. .

L271: "The nitrogen available for growth is the total available nitrogen multiplied through by (1 λ)." – I assume that the "nitrogen available for growth" is Navail and is used in L283. Navail isn't defined until L378. Please clarify this in the text.

Corrected

Section 3.2.1 – Does litter and diffused SOM enter frozen soil layers? Could be the reason we see a higher soil C for CNlayer at higher latitudes (Figure 7).

This has been added to the model description: $D(z)$ is the diffusivity in m$^2$ s$^{-1}$ and varies both spatially and with depth \citep{burke2016gmd}:

\begin{equation} \label{diff}

D(z) = \begin{Bmatrix}

D_o & ; & z \leq 1 m \\

\frac{D_o}{2}(3 -z) & ; & 1 m < z < 3 m \\

0.0 & ; & z \geq 3 m

\end{Bmatrix}

\end{equation}

Without permafrost, $D_o$ (m$^2$ s$^{-1}$) is given by a bioturbation mixing rate equivalent to 1 cm$^2$ year$^{-1}$. When permafrost is present, the mixing represents cryoturbation and $D_o$ increases to a value equivalent to 5 cm$^2$ year$^{-1}$. This parameterisation of $D(z)$ means that the soil organic pools can transfer between permafrost and non-permafrost soils albeit at a relatively slow rate.

We have expanded the discussion around Figure 7 and the vertically resolved soil biogeochemistry to include the "The soil in JULES-CN$_{layer}$ has more organic carbon (Figure \ref{fig:zonal_stocks}), organic and inorganic nitrogen (Figure \ref{fig:fluxes_stocks}). The parameterisation of the vertically resolved soil biogeochemistry means that once JULES-CN$_{layer}$ is spun-up the soil carbon and nitrogen within the frozen soil is relatively stable because of the low temperatures."

L430-436: – The additional turnover of inorganic nitrogen is a great solution to a well- known issue when soil N starts building up uncontrollable due to N deposition or BNF.

Agreed. It is something we plan to investigate in greater depth in the future.

Section 3.2 and 3.3 – A table with constants from sections 3.2 and 3.3 similar to Table 1 for section 3.1 would be a nice addition to the manuscript.

This has been added as Table 2.

L532-534: – N leach is very small. Any idea why it is so small? Have you considered some adjustments to get the number to increase? Change the value of β?

We have changed the value of the effective solubility of nitrogen in water and can get an increase in the leaching by doing this. However, it is still fairly small compared with the estimates in Figure 4. One of these reasons is that, in reality, some component of the leaching is from the fertilizer which is not yet included in JULES-CN. We have added a comment to this effect in the document.

L538-539 and Figure 4. – Net N mineralisation and N uptake seem to be very small. Are the units for them really Tg N yr-1?

These were in the wrong units and have now been updated

L564-565: "This is a consequence of the higher nitrogen limitation on JULES-CN lead- ing to less litter fall and subsequently less soil carbon." – I guess N limitation on SOM decomposition isn't strong enough to make the SOM pools increase in size? Could it be that the fixed plant C:N ratios prevent feedback of poorer litter quality under higher N limitation that would result in a slowdown of SOM decomposition?

Yes, it is feasible a shift to a lower C:N plant ratio would decrease little quality in turn slowing decomposition. The impact will be dependent on the balance of processes and any change in total litterfall.

Figure 1. – Fixation seems to enter the vegetation in the figure, but section 3.1.1 says it enters inorganic N pool. Update figure.

Figure 1 has been eliminated because it is very similar to Figure 4 and supplies no additional information over Figure 4.

Figure 6. – Is the increased soil C at high latitudes for CNlayer mainly due to the additional decay rate modifier per depth or is it due to N limitation on decomposition? Because with a lot less vegetation

C the input of litter must also be less. So something else needs to dictate the build-up of soil C as this is opposite to what is stated in L564- 565.

In the Nhlat when JULES-C is compared with JULES-Clayers there is a large increase in organic carbon (see Figure 6 in https://gmd.copernicus.org/articles/10/959/2017/gmd-10-959-2017.pdf). In both JULES-CN and JULES-CNlayered the N limitation on decomposition is relatively small. The vertical profile of soil temperature has a big impact on the decomposition in the layered models and allows soil carbon to build up in the deeper soils. The layered model is expanded upon further in the text.

Figure 6, 7 and 9. – Figure 6 is the result we are after when introducing an N cycle, N limitation on productivity. The N limitation spatial distribution puzzles me to some extent. That you haven't investigated the reason for the strong N limitation in tropical savannah (L550-551 "Further work is required to understand why tropical savannah is so limited.") is something I think should have been done. And also that Northern Europe doesn't see any N limitation, but Western Europe does is also strange. I would have liked to have maps for figure 7 and 9 to try and understand this better, now a lot of information is hidden within the latitudinal bands. Also, a figure with annual net mineralisation would be of interest to understand what is happening.

Interestingly, I have changed how to extract the biome specific information out of the model results (medians instead of means) and now we get the savannah and tundra forest being OK limitation-wise but the tropical forests not being limited enough. (it's a bit scary how different the use of a slightly different metric can make the results appear!). We do think, however, that the new Figure 5 and 6 are a more appropriate reflection of each other. This means that we are now interested in why tropical forests aren't limited enough - Phosphorus?. This has been added to the discussion.

We have also added an additional figure which includes of the more relevant N stocks and fluxes and a discussion about this impact of this figure.

Figure 6, 7 and 9. – How can it be that CNlayer has stronger N limitation at higher latitudes than CN (less Veg C in figure 7 and more yellow in figure 6) when there is more inorganic N in the soil (figure 9)? This needs to be explained better. Is it due to the root profile and that all N isn't available?

Indeed, there are two inorganic nitrogen pools in the layered model - the total pool and the inorganic N that is available to the plants. This depends on the root distribution and on whether the soil is frozen. There may well be less available inorganic nitrogen in JULES-CNlayered than total inorganic nitrogen in JULES-CN meaning that the plants could be more nitrogen limited in some regions. This discussion is expanded in the discussion about JULES-CNlayered.

Technical corrections

L9: "Biological fixation and nitrogen deposition are external inputs. . ." – From section 3.1.1 it is clear that BNF isn't an external input. Please revise this sentence

Corrected

L204-205: "We therefore a new parameterisation of retranslocation and labile nitrogen that is dependent on the phenological state" – please revise this sentence

Done

L278: ". . . is is . . ." – remove one is.

Done

L474: ". . . Equation51 . . ." – change to ". . . Equation 51 . . ."

Done

L646: ". . . residence tome of carbon . . ." – change tome to time.

Done

L675: " . . . model model . . ." – remove one model.

Done

Figure 4. ". . . period 19960-2005 . . ." – correct to 1960.

Done

References

Davies-Barnard, T., Meyerholt, J., Zaehle, S., Friedlingstein, P., Brovkin, V., Fan, Y., Fisher, R. A., Jones, C. D., Lee, H., Peano, D., Smith, B., Wårlind, D., and Wiltshire, A.: Nitrogen Cycling in CMIP6 Land Surface Models: Progress and Limitations, Biogeo- sciences Discuss., https://doi.org/10.5194/bg-2019-513, in review, 2020.

Mercado, L. M., Huntingford, C., Gash, J. H., Cox, P. M., and Jogireddy,  V.:  Improving the representation of radiation interception and photosynthesis for cli- mate model applications, Tellus B, 59, 553–565, 2007, https://doi.org/10.1111/j.1600- 0889.2007.00256.x.

---

## Author Comment (AC2) · 3 Nov 2020

William Wieder (Referee)

wwieder@ucar.edu

Wiltshire and co-authors nicely document their additions of a nitrogen cycle and vertically resolved soil biogeochemical model to the JULES model for use in UKESM1. The offline simulations include documentation of simulated vegetation and soil carbon and N pools and fluxes and their change over the historical period. A comparison with some observations is provided for model evaluation

Major concerns

My major concerns aren't that substantial, but stem from contradictions in what's expected from the paper and what's actually delivered.

The paper sets off comparing the C only, CN and CN_Layered implementation of the model, but a number of display items omit results from the CN_Layered configuration. Specifically, Fig. 10-12 & Table 2 do not show results from the layered model, why? Should these effects of vertical soils also be discussed in 5.3? Because these results are not presented, I think major revisions are warranted.

These were not all included so as to simplify the story. However, we will re-examine and add the CN_layered simulations where it is most appropriate in the revised version. This has involved a significant re-write of the Results section which is more comprehensive.

Are there meaningful differences in plant distributions simulated with the new N enabled or CN_layered models?

We have added a Figure showing the pft distribution of the different types of vegetation. This configuration of the model has not yet been brought together with the new height competition which is included in UKESM1 so the exact PFT distributions will change with extra vegetation types and a height-based competition. Therefore, the results are just an indication of the effects of changing the model configuration.

The multi-layered canopy model is introduced in section 3, but never really discussed in section 5. Should it be? Are there any interesting insights enabled by this new feature of the model?

The section has now been extended. The idea behind this section was to document the link between leaf level photosynthesis and respiration and the interactive N scheme. The section has been restructured and updated.

Minor and technical concerns: These are more numerous but intended to clarify and improve the paper.

I like the high-level overview of the main findings summarized in the abstract, but I wondered if more quantitative results should also be provided (pending length requirements for the journal)?
We have added a couple of sentences discussing the values of the nitrogen limitation and the carbon use efficiecy to the abstract.

Paragraph starting on line 70. I appreciate how clearly model assumptions are laid out. For example, the approach here looks at the "large-scale role of nitrogen limitation on carbon use efficiency", but I wonder if there's evidence to support this common assumption made in models in real ecosystems? What is the assumed impact of N limitation on NEP? The net results it that is dampens

The introduction has been changed significantly to include these additional bits of information, plus the additional text suggested by the other reviewers.

Can paragraphs around lines 60 & 90 effectively be combined? Both paragraphs seem to have a common purpose of documenting the model connections and history. It's also not really clear how JULES fits into UKESM (also called UKESM1) vs. HadGEM2

We agree this is unclear. We have reworded this so as to make it clearer. We have combined line 90 on into the beginning of the model description section when more details are required.

Is section 2 subheading really warranted? Maybe just combine subheadings for 2 & 3 into one longer section?

Done

There are some redundancies in the text (section 3) where sentences are repeated at different points.

Section 3 has been altered so it is now a general introduction to JULES and the model description below.

Line 162. I'm confused why "These stoichiometric functions already exist in the model" for MR fluxes. This suggests the new work here is just to explicitly represent the Npools that were being implicitly assumed in the carbon only model? Separately, is it worth documenting the source for vegetation stoichiometry (presumably used in Cox et al. 2011)?

This section has been revised to make clarify what is existing and what has had to be extended to have a fully interactive N scheme. The vegetation stoichiometry is also referenced - Enquist, B. J., Brown, J. H., and West, G. B.: Allometric scaling of plant energetics and population density,Nature, 395, 163–165, 1998

Fig 1: The assumption that 'roots' in the model have a lower (or equal) C:N than leaves seems surprising to me, but this but seems contradicted by 'Ratio of root to top leaf nitrogen' (Table 1), please clarify. Roots have wide variation in C:N (Iversen et al. 2017), but if anything I'd assume they should have a higher C:N ratio than leaves (Kattge et al. 2011).

Roots have the same C:N ratio as the top leaf, but as N concentration decreases through the canopy the current formulation means that the C:N ratio is lower. Future work will explore parameterising root C:N ratios directly. We note this in the discussion.

Table 1: "Top leaf nitrogen concentration": listed twice

Removed

Line 175, this statement doesn't seem to be true for grasses, which have declining C:N with height (Fig 2).

Corrected.

Section 3.1.1, oh no, why define N fixation (which should limit NPP) as a function of NPP in the model?! This isn't the first modeling group to make this assumption, but a brief discussion and literature review seems warranted (see Vitousek et al. 2013; Thomas et al. 2105; Wieder et al. 2015; Meyerholt et al 2016)

We have inserted further discussion, including the references suggested.

Section 3.1.1- I think inputs from N fix lead off these details of the CN model because that's where the N cycle 'starts', which seems logical, but putting it under a "Vegetation carbon and nitrogen" (subheading 3.1) seems odd, especially since Nfix contributes to the soil N pool (not plants). Maybe different names for the higher level subheadings (3.1 and 3.2) would be warranted? Alternatively, use Fig 1 to group these fluxes together.

We have tried to sign post the different components of the nitrogen cycle better. The fixation is now included in the inorganic nitrogen section.

Line 182 What is potential NPP? (eq. 9). How does this different than the 'actual' NPP? If not discussed here, please reference where this is described (3.1.4).

This section been moved to the Inorganic nitrogen section. NPP_pot is defined very clearly in the vegetation growth and allocation section..

Line 225 where is the multi-layer canopy approach included in these simulations? I'm assuming with with CNlayered, but this isn't clear in section 4 (line 495)

This has been changed to - "JULES-CN$_{layer}$ is a version of JULES-CN which has identical above ground processes to JULES-CN but additionally includes vertically discretised soil biochemistry."

What is 'spreading' in the model (section 3.1.4)? Is this prescribed by some land use time series dataset or prognostic (more like a DGVM)? Text on page 11 makes me think it's the later.

This has been added: "Biomass can also increase by spreading through an increase in covered area" where the term spreading has been itnroduced.

The assumptions made in the phenology and allocation section are thoroughly defined, but it's hard to understand for readers not familiar with TRIFFID how N limitation is implemented in the model. It seems like it's an instantaneous down regulation of NPP, with extra carbon respired by plants that are N limited? With that N limitation calculated by the tissue and pft specific stoichiometry defined in the model?

Yes, this is correct. The model description has been updated to make it clearer.

Eq. 25-28. I don't really understand how the soil model is wired based on these equations. If R_DPM and R_RPM are the respiration terms from litter pools, how do some of these fluxes go back into the BIO and HUM pools, which themselves are respired (and also simultaneously included as inputs to BIO and HUM)? It seems that soil respiration fluxes to the atmosphere are actually R_tot*B_R, if so, the R_* fluxes should be some kind of soil turnover term (not respiration).

This has been changed to make the respiration/turnover clearer. New text -", $R_{tot} = R_{DPM}+R_{RPM}+R_{BIO}+R_{HUM}$ where $R_{tot}$ is the total turnover in kg\,[C]\,m$^{-2}$ s$^{-1}$. $(1-\beta_R)$ is the fraction of the total turnover that is respired to the atmosphere. $\beta_R$ depends on soil texture and ranges from 0.75 for a clay soil to 0.85 for a soil with no clay content. From this the respiration to the atmosphere can be defined as $(1-\beta_R)$ $R_{tot}$.\\"

It seems like B_R is a critical number here, as it controls the soil carbon use efficiency and the amount of N required during litter decomposition (eq. 35). Is this parameter value defined somewhere?

Beta R is now defined in a new equation: \beta_R = \frac{1}{4.09+2.67e^{(-0.079clay)}}

Eq. 29-32 do the N fluxes need to include I_DPM + I_RPM?

No - immobilisation is a microbial process in which inorganic nitrogen is made into new organic matter. Microbes don't make new plant litter (plants make that!), they only produce BIO/HUM. The I_DPM and I_RPM terms are there in I_tot. They're somewhat confusingly named. I_DPM is the immobilised nitrogen that *originated* from DPM.

Line 355, as above can this be called potential turnover, not "potential respiration"?

This has been changed

Eq. 33. I'm trying to wrap my head around the vegetation controls over decay rates and how that may feedback to a CN model that has vegetation with very different stoichiom- etry and N demand (woody vs. grass pfts; Fig 2) but that allows for plant competition (on a single soil column). I assumed the maps of nutrient limitation (Fig. 6) reflect differences in vegetation N demand (per unit of C), but are decay rates also slower for grasses (increasing the N limitation in these ecosystems)?

The interactions and feedbacks are potentially highly complicated given the ability for the PFTs to compete. The grasses produce a higher fraction of decomposable plant material relative to the tree PFTs (0.67 to 0.25, now in Table 1). In turn, decomposable plant material decays approximately 300 faster than resistant material. Grasses therefore have a faster turnover of nutrients. Our interpretation of Figure 6 (now Figure 4) is that it reflects the vegetation N demand. However, more work is required to understand the savannah grass response.

Eq. 36, Is this still a potential decomposition rate, as it's 'limited' by N availability?

This has been changed

Line 385, what are 'these two pools'? I think it should be DMP and RMP, but it's not clear in the text?

This has been clarified – indeed there are the two litter pools

What happens to wood in the soil CN model? How is it allocated to the pools de- scribed?

The ratio of dpm to rpm is a PFT dependent parameter so implicitly takes into account the proportion of wood in a PFT. It is lower for a woody pft and higher for a grass pft.  This is discussed at the top of Section 2.2.

Eq. 39, is there anything that prevents this flux from being negative? Are there times when immobilization > mineralization?

Fluxes will have been limited by Fn to make sure this isn't negative. If it hits the minimum pool size, it calculates a correction term (neg_n) and that correction term is then included as a negative gas flux. But that is applied just as an 'extra' gas flux and not applied to minl and immob. So Eq 39 is never negative, but gas flux can be, if that makes sense! This has been added: ". f$\_N$ limits the nitrogen fluxes so that (M$\_{tot}$ - I$\_{tot}$) is always positive. However, if pool sizes become too small N$\_{gas}$ could become negative to ensure nitrogen is conserved."

Eq. 39, Should the N loss description go into 3.3 (inorganic N) instead of the soils section (3.2)?

I think it is clearer to have this first component of gas loss here because it is defined using the mineralisation/immobilisation which is discussed here. I agree that it is on the boundary between inorganic and organic nitrogen.

Where does N_turnover flux (eq. 46) go in the model, the atmosphere? How large is this tuning flux relative to other loss terms?

N-turnover flux it has been renamed N_gasl and goes to the atmosphere. This is now discussed in Figure 4 and we state the proportion of loss via this process is about 90% of the total gas loss.

Eq. 46, where does N_gas (eq. 39) fit into the N budget summarized here? Section 4, How does the model handle agricultural fractions of grid cells?

This has been added: The total gas loss is the sum of $N_{gasI}$ and $N_{gas}$ from Equation \ref{eq:ngas}. There are no agricultural fractions represented by this model. Ive stated that there are two gas loss terms.

Section 5.1 I'm used to fluxes and pools being roughly proportional in models like this. If NPP is 11% lower in the CN model, why are the vegetation stocks roughly equal in the C and CN model? Similarly, if the vertically resolved model has a similar NPP to the CN model why are vegetation C pools so different?

This is because the turnover times change – the vegetation and soil turnover times are now plotted separately

Fig 4, 8 and others, Since the text is organized with C, CN, and CN_layered should the display items be similarly organized?

We put JULES-CN first because that is the configuration we are describing as the main focus of the paper. JULES-C is only included in the paper for comparison purposes and JULES-CNlayered is included last because is an extension of JULES-CN. We will check through the text and make sure it is that way in the text. This was particularly relevant when discussing the "historical simulations".

Fig 4 what is the 'N-loss' flux supposed to represent? As drawn, I think this is a gaseous N loss, but as labelled it's not clear how this connects with N_gas and N_turnover fluxes (see above).

The N loss term has been changed to a N Gas term and it is the sum of the gas losses from the inorganic N pool and the organic N pool. This has been added to the captionN- gas is the sum of $N_{gas}$ and $N_{gasI}$ with $N_{gasI}$ approximately 90 \% of the total gas loss

Fig 4, how deep are the soils being represented, this is especially important to consider in the vertically resolved model and should likely be described in methods (3.2.1)

This has been added to section 3: These configurations of JULES adopt the standard 4 layer soils with a maximum depth of 3 m. However it should be noted that \cite{burke2016gmd,chadburn2015gmd} adopt a configuration which increases both the maximum soil depth and number of soil layers.

Line 535, doesn't this just mean the model is at equilibrium as it should be given your spinup procedure?

However, the model could still be in equilibrium with a slower recycling rate. The recycling rate through the system is a characteristic of the model.

Section 5.2 seems out of place, as the extent of N limitation should be preceded by a more thorough comparision of the model states and fluxes. One suggested organiza- tion could be comparing the 1) Spatial distribution of present day stocks / fluxes and residence times (e.g. Figs 4, 10, 7, 9, & 8 in that order) and 2) Temporal evolution of relevant stocks / fluxes over the historical period (e.g. Fig 11 & 12) 3) N limitation (Fig 6, 5) as diagnosed by NPP_Potential / NPP and its evolution over time (Fig 11b).

The results section has been re-worked to make a clearer flow through the figures. Stocks and fluxes followed by N limitation.

The title for Fig 5 (and associated text) implies that you conducted a N fertilization experiment (see Wieder et al. 2019), but I don't think this is accurate. Instead you're calculating a N limitation diagnostic (NPP_pot/NPP) and comparing that to results from an observational synthesis.

Title changed to "Response ratio ($NPP_{pot}$ / NPP achieved)" as in Figure 6. This is defined as "the response ratio, is the ratio of the potential amount of carbon that can be allocated to growth and spreading of the vegetation ($NPP_{pot}$) compared with the actual amount achieved in the natural state (NPP)". Text about the observations is also changed to include: "which summarises a meta analysis of nitrogen addition experiments. The black bars showing the mean of the observations and the red lines the uncertainty."

I'd suggest flipping the order of Figs 5 & 6, as they both show the same information, but Fig 6 is less processed model output, with 5 serving to summarize biome-specific information and related it to observations.

We have switched the order of Figures 5 and 6 as suggested and changed the associated text.

Fig 6, Line 553. It seems like the model is more strongly limited in grasslands, which have much higher N requirements / unit of C (Fig 2). This doesn't really show up in results for 'tundra' or 'grasslands' (only for Savannah). I wonder why?

We have looked at this again. We have decided to use the median of each biome rather than the mean of each biome to calculate the results shown in Figure 6. This is because JULES does not necessarily simulate the correct vegetation for the whole of each biome and outliers will influence the mean. These will not influence the median in the same manner. Looking at Figure 6 the results are more comparable with what we might expect – JULES is not limited enough in the forest biomes but it is now more appropriately limited in the tundra and the savanna. We have also changed the scale of they-axis in the  figure to make the results clearer to see.

Should multi-paneled figures be labeled ('a', 'b', 'c') and accordingly described in the figure caption?

This has been done

Fig 7, can legends be smaller or moved into the figures (as in Fig. 9) so the data are easier to read?

This figure has been improved as suggested.

Fig 9, should the bottom panel be labeled soil C residence time and also include data from the C-only model?

This has been deleted and the soil residence times plotted instead in Figure 8.

Fig. 11b what is the time series of 'response ratio anomaly'? Is this the change in NPP_Pot / NPP_act that used to diagnose N limitation shown in Figs 5 & 6? If so, is this what you're calling 'progressive N limitation' (line 599), in which case this should be clarified on and expanded in the text.

We have moved this sub figure to sit alongside the other discussion of Nitrogen limitation and expanded upon it in the text.

Fig 12 & section 5.2.3 The low bias in NEE ( 0.5 Pg / y, roughly 25%). This would lead to an underestimation of the land carbon sink of about 25 Pg over the period from 1960-2010 (or about 12 ppm $CO_2$ in the atmosphere). Thus, while the IAV of NEE looks better here, that overall magnitude of the land sink may be too low with the CN version of the model. This isn't a deal breaker for the paper but time implications of the low bias with the CN (and CN_layer) model should be discussed in

the text, especially since JULES_CN is included in UKESM1.

A discussion has been added in the text. The relationship between JULES-CN and UKESM1 has been made clearer – they are related, but the configuration of JULES in UKESm1 (JULES-ES) has a whole bunch of other components which are not included here and will affect both the vegetation distribution and the NEE.

Section 5.3. Is it just frozen soils that are causing this? it seems the differences in Veg C pools extend down to 40 degrees north (Fig 7). Is this somehow connected to assumptions about the fraction of N that plants have access to in the vertically resolved model (e.q. 51)?

Indeed, the plants with the shallower roosts preferentially take nitrogen from the shallower soils so this process will also contribute to the nitrogen limitation in JULES-CNlayer. This has been added: "This additional limitation of nitrogen uptake caused by frozen soils and the dependence of plant N uptake on root distribution"

Line 611, as noted in Fig 12, the low biases in land C uptake seems notable if you're trying to capture changes in the atmospheric CO2 growth rate.

This is stated here slightly further down: ". Due to nitrogen limitations on CO2 fertilization, mean NEE in JULES-CN (1.66 Pg C/yr) is lower than in JULES-C (2.06 Pg C/yr), and also lower than the estimate from GCP (2.11 Pg C/yr)"

Line 671 what is "climate-induced mineralization" I'm assuming this has something to do with accelerated decomposition from climate change increasing N mineralization rates?

Changed to "accelerated soil decomposition caused by climate change leading to increased mineralisation rates"

Is there a data availability statement required for the journal?

As a model description paper we provide access to the code as documented here and the rest of the JULES model subject to a freely available non-commercial licence agreement. In addition to further encourage and support the use and application of standard 'configurations' we provide access to the 'suites' used here.

References: Iversen, C. M., McCormack, M. L., Powell, A. S., Blackwood, C. B., Freschet, G. T., Kattge, J., . Violle, C. (2017). A global Fine-Root Ecology Database to address below-ground challenges in plant ecology. New Phytologist, 215(1), 15-26. doi:10.1111/nph.14486

Meyerholt, J., Zaehle, S., & Smith, M. J. (2016). Variability of projected terrestrial biosphere responses to elevated levels of atmospheric CO2 due to uncertainty in bio- logical nitrogen fixation. Biogeosciences, 13(5), 1491-1518. doi:10.5194/bg-13-1491- 2016

Thomas, R. Q., Brookshire, E. N., & Gerber, S. (2015). Nitrogen limitation on land: how can it occur in Earth system models? Glob Chang Biol, 21(5), 1777-1793. doi:10.1111/gcb.12813

Vitousek, P. M., Menge, D. N. L., Reed, S. C., & Cleveland, C. C. (2013). Biologi- cal nitrogen fixation: rates, patterns and ecological controls in terrestrial ecosystems. Philosophical Transactions of the Royal Society B: Biological Sciences, 368(1621). doi:10.1098/rstb.2013.0119

Wieder, W. R., Cleveland, C. C., Lawrence, D. M., & Bonan, G. B. (2015). Effects of model structural uncertainty on carbon cycle projections: biological nitrogen fixation as a case study. Environmental Research Letters, 10(4), 044016. doi:10.1088/1748- 9326/10/4/044016

Wieder, W. R., Lawrence, D. M., Fisher, R. A., Bonan, G. B., Cheng, S. J., Goodale, C. L., . . . Thomas, R. Q. (2019). Beyond Static Benchmarking: Using Experimental Ma- nipulations to Evaluate Land Model Assumptions. Global Biogeochem Cycles, 33(10), 1289-1309. doi:10.1029/2018GB006141.

---

## Author Comment (AC3) · 3 Nov 2020

This manuscript explains the N cycle in the JULES-CN model which forms the land component of the UKESM. Simulations from the UKESM have contributed to the CMIP6 effort. The N cycle component of JULES, as explained, here is very simple compared to existing models out there. This is completely acceptable as long as it is clarified that the model parameterizations are simple, their limitations acknowledged, and the implications discussed. I am afraid, however, that the manuscript doesn't appear to do so and in my mind requires substantial work to address this and other concerns I raise below.

1       Major comments

I have several major concerns.

It is well known that leaf N content is related to its photosynthesis capacity (Field and Mooney, 1986). When $CO_2$ increases, photosynthesis increases but this rate of increase is slowed if enough N is not available. This process is referred to as photosynthesis downregulation (McGuire et al., 1995). So, it is clear then, that N limitation acts on photosynthesis and thus on the gross primary production (GPP) flux. However, the approach used in the manuscript, in contrast, reduces the NPP (without adjusting the GPP) which is equivalent to reducing carbon use efficiency (CUE = NPP/GPP). Since there is no biological justification for this provided, I am struggling to understand the reasoning behind this. Also, if that framework is still used, TRIFFID models Vcmax as a function of leaf N content (eqn. 51 in Cox 2001) so it makes sense to adjust Vcmax.

Related to this concern, is the fact, that I am not able to find in the manuscript in detail how this reduction in NPP is implemented or how it results and because of the interaction of which processes. Unless I missed it, the only reference to this important process is made on line 78 as "... and then reducing plant net carbon gain to match available nutrients".

It is well known that current observation-based CUE is around 0.5. This is also seen in Figure 10. The CUE for the JULES-CN model is lower than that for the JULES-C model because that's how JULES-CN is designed - to lower NPP and hence CUE as $CO_2$ increases and N supply can't keep up. I am wondering what happens in a future simulation for RCP 8.5 scenario. Will your CUE drop down to something like 0.25 by year 2100 which seems totally unrealistic? This will be one implication of your model design since you have chosen to reduce NPP and not GPP.

Agreed, it is well established that tissue level N concentrations correlate with photosynthetic capacity and metabolism. It is also established the first order effect of N fertilisation is enhanced growth. However, it is less clear on the mechanisms, for instance field experiments of enhanced N fertilisation have found increases in growth but no change in photosynthetic rate (e.g. Brix et al., 1969, Wang et al., 2012). Other analysis, looking at climatological gradients in N deposition found no

dependency between foliar N and N deposition. It is, however, fair to say other analyses (e.g. Mao et al., 2020) do establish this link. In general, models to date make differing assumptions about these coupling mechanism between C and N cycles leading to substantial uncertainty in their projections (Zaehle and Dalmonech, 2011).

Our approach here is to capture the established first order emergent response of N addition on growth which translates into leaf area and biomass without the complex and uncertain impacts on leaf physiology. As we use a fully dynamic vegetation scheme Nitrogen availability can drive changes in plant level C:N ratios through competition. This is the first implementation of a coupled C-N scheme in the UK model and we fully expect to develop this aspect further including assessing flexible stoichiometry.

Further to the points raised above, not of all the CMIP6 models only UKESM and MPI-ESM 1.2 (Mauritsen et al., 2019) include a coupled fully dynamic vegetation model with nitrogen scheme. In both cases, the schemes assume fixed plant stoichiometry. In which case, if CUE dropped very low there would be a dynamic vegetation response leading to a dieback. In our CMIP6 experiments we do not see a strong reduction in CUE (over the course of ssp585 reduction from 0.53 to 0.48). Analysis of CMIP6 runs as part of C4MIP demonstrates we have a strong and robust representation of carbon feedbacks (Arora et al., 2020).

The second big assumption in the model is that of fixed C:N ratios of plant tissues. The implications of this assumption are not discussed. Since C:N ratio of plants varies in space (as indicated by different values of nl0 in Table 1) this indicates their ability to adapt to different environmental conditions in space. Assuming, plants can do the same in time as CO2 increases doesn't this imply that the assumption of fixed C:N is too strong and your model will limit NPP perhaps more excessively than it in the real world (with the caveat that in the real world GPP is constrained).

It is common for DGVMs to parameterise a top-leaf nitrogen content per PFT as part of the process of capturing diversity and functional traits. This is the case whether a full nitrogen cycle is included or not. This is common with the point raised in 1. It is likely that foliar N varies in space independent of nutrients as was found in the Aber study. This is not captured in the typical approach to dynamic vegetation modelling. It is entirely plausible that with increased nutrient limitation plants limit their foliar N and therefore GPP and NPP. In our approach, with fixed stochiometry we may excessively constrain the model but through the dynamic vegetation response we might see a shift towards a plant with a lower C:N.

In context of model evaluation, it would have been extremely helpful to include a simulation in which N deposition is turned off. This simulation would have allowed to see if the effect of N deposition is indeed to increase NPP as would be intuitively expected.

This paper is just an initial description of the JULES-CN model which alongside other additional land surface processes has been implemented in UKESM. The JULES-C run implicitly gives an signal of the effect of N deposition.  It has also been used within Davies Barnard et al. (2020, https://bg.copernicus.org/articles/17/5129/2020/bg-17-5129-2020.pdf) who explored the response of NPP to N and CO2 fertilization from perturbation experiments. More detail of results from the Davies Barnard paper have been added to a new discussion section.

In addition, the TRENDY model simulation S2 doesn't take into account land use change and the fertilization of crops. Crop fertilization is a major source of leaching and gaseous emissions of N2O

and NOx. I am wondering if this is the possible reason that the simulated leaching in Figure 4 is so low compared to other estimates.

Yes, it is quite likely although we haven't explicitly quantified that. We have added a discussion about N fertilization and leaching into the leaching section.

Also, does the model simulate the realistic sign of response when driven with climate forcing only. Typically, a model's response to various forcings allows to see at least if the sign of the response is consistent with expectations.

These biogeochemistry only/ radiative only/no N deposition runs are available in TRENDY and C4MIP and will be assessed as part of our future work. However, we think it is beyond the scope of this paper to include a comprehensive assessment of these results. We will refer to these studies as part of the new discussion.

4.      I realize that there are very few observation-based estimates available for N re- lated pools and fluxes. However, still there are plenty of quasi-observation and model based estimates against which model results could have been compared. For example, in Figure 4 there are no quasi-observed or model estimates for sev- eral quantities. Model estimates are, however, available for immobilization and mineralization (von Bloh et al. 2018), plant N uptake (Zaehle et al. 2010; Xu-Ri and Prentice, 2008; Wania et al., 2012), and inorganic N mass (von Bloh et al. 2018; Xu-Ri and Prentice, 2008; Wania et al., 2012). These estimates will allow to put your model results in some context.

These numbers have been added to figure 4 and it has been noted in the text that some of these comparisons are from other models rather than available observations.

5.      Model parameterizations are not compared to other models, and the conceptual basis of parameterizations and their implications, are not discussed (as men- tioned above for the choice to reduce NPP and use fixed C:N ratios) .

We have significantly revised the model description section. We think our new description will address these concerns in sufficient detail.

For example, biological nitrogen fixation (BNF) is modelled as a straight-forward function of NPP. This is okay but the manuscript doesn't note that meta-analysis studies have found that BNF increases with increasing CO2 (Liang et al., 2016) but decreases with increasing N deposition and fertilizer application (Ochoa- Hueso et al., 2013) both of which apparently result in increase in NPP.

This has been added to the BNF section.

 In addition, BNF is typically higher over agricultural areas.

BNF is, in effect, very high over agricultural areas in JULES, as nitrogen is not limiting for cropland areas and the source (fertilisation or BNF) does not affect the model outcome.

Similarly, all gaseous losses are expressed using Nturnover but in nature there are several pathways using which gaseous losses occur. N2O and NOx losses occur during nitrification (via nitrifer denitrification) and N2, N2O, and NOx losses occur during denitrification.

Not interested in losses to atmoshere but removing the appropriate amount of N

It would be scientifically beneficial for the manuscript, and for a reader, if sim- plifications made are clearly highlighted and their limitations discussed, because then it is possible to interpret the model results in light of these limitations.

We have significantly revised the model description section. We think our new description will address these concerns in sufficient detail.

6.      The majority of the results shown in the manuscript focus on the ability of the new model to reproduce all the aspects of the C cycle as the previous model did. As a result, the N cycle module is not evaluated rigorously. The manuscript doesn't report N demand, how it changes over time, what part of the N demand is not met, what part of N demand due to increasing CO2 is met by N deposition, time series of mineral N pool, time series of plant N uptake, time series of C:N ratio of whole plant and other plant components, and geographical distribution of simulated C:N ratios (even though I realize they are specified). Since this is the first time JULES' N cycle component is being published it is reasonable to expect that such a manuscript will rigorously assess the new N cycle module.

We have added a figure with time series of N demand for growth and spreading, N uptake for growth and spreading alongside the unmet N. Net N mineralisation, C to N ratio of litter,

7.      There is no mention of phosphorus cycle at all. It is well know that in the tropics phosphorus limits photosynthesis and not nitrogen. How is this accounted for? My guess is this is somehow built into the Vcmax rates which are function of leaf N content (eqn. 51 in Cox 2001). If the model can reproduce correct zonal distribution of GPP it must take phosphorus limitation in the tropics somehow into account.

The effect of Phosphorus has been discussed as part of a future direction and understanding paragraph.

8.      Finally, the lack of units, the lack of rate change equations for several pools, and unclear statements make it difficult to understand the model parameterizations as noted below in minor comments. In its current form, there is no way a reader can fully understand and reproduce the parameterizations reported here in some other model.

Units have been added to the relevant variables, additional rate change equations have been added. This has involved quite a few changes which are apparent in the document but hard readily document here. I think we have significantly improved the readability of the document. We have also added a nomenclature section as an appendix.

2      Minor comments

9.      Abstract, line 8, "It represents all the key terrestrial nitrogen processes in an efficient way.". The word "efficient" here is misleading.

Change to parsimonious

10.     Abstract, line 9, I find it extremely confusing that BNF is mentioned as an external input. BNF is how N enters the coupled vegetation and soil system. Consider the case, if we were to refer GPP as an external input since that's how C enters the coupled vegetation and soil system. N deposition and fertilizer, on the other hand, can be called external because they are not natural just like fossil fuel emissions.

Changed to "Biological fixation is dependent on productivity, with nitrogen deposition as an external input"

On page 2, in addition to BNF, leaching is also referred to as an external (loss). This also seems strange since on the carbon cycle side we don't refer to heterotrophic respiration or dissolved inorganic C in runoff as external losses.

Changed to "Nitrogen leaves the vegetation and soil system via leaching and a bulk gas loss parameterisation"

11.      Page 2, line 34, "Internally organic N is lost ...". Here "internally", perhaps is much better described as "cycling of N within the coupled vegetation and soil system".

Changed to: "Within the system organic nitrogen is transferred from the vegetation to the soil through the production of litter and disturbance"

12.      Page 2, line 36, "Both inorganic and organic nitrogen may become available for plant uptake". Since organic N uptake is very small and therefore not even mod- elled (including in your model) perhaps it would be better if this is clarified.

Changed to : ", although the amount of inorganic N uptake by plants is  small and typically not included in models"

13.      Page 2, line 39. "In a changing climate, rising atmospheric CO2 drives an in- crease in the terrestrial carbon cycle and Gross Primary Productivity (GPP)." This is a vague sentence. What does "an increase in the terrestrial carbon cycle" means?

Changed to: "rising atmospheric CO2 drives an increase in the land carbon uptake and hence an increase in the gross primary productivity. This results in an extra demand for nitrogen which could potentially limit the increase in future carbon stocks"

14.      Page 2, line 56, " ... are between a reduction of 39 % and a slight increase of 1% ...". Please consider rewording this sentence/phrase. It is somewhat hard to follow.

Changed to: "For example, \cite{doi:10.1111/gcb.15114} used a perturbed model ensemble to show that N limitation reduces both the projected future increase in land carbon store due to CO$_{2}$ fertilisation and the projected loss in land carbon due to climate change"

15.      Page 3, line 65. " ... and a new managed land module ...". Please consider rewording to "and a new module for land management ...".

Done

16.      Page 3, lines 72-74. "This is achieved by extending the implicit representation of nitrogen in the existing dynamic vegetation and plant physiology modules TO EN- ABLE A MORE COMPREHENSIVE NITROGEN CYCLE WITHIN THE LAND SURFACE". Please consider deleting the text in capitals given N cycle framework used here is extremely simplified.

Done

17.      Page 3, Lines 74-75. "Nutrient limitation operates through two mechanisms; the available carbon for growth and spreading is reduced and the decomposition of litter carbon into the soil carbon is slowed". The word spreading at this point in the manuscript is unclear. Only after reading

the rest of the manuscript it is clear that "spreading" means changes in the spatial extent of vegetation. Please consider using another phrase/word to replace "spreading".

changed to "the available carbon for vegetation uptake is reduced"

Please also consider not using the phrase "decomposition of litter carbon into the soil carbon" here and elsewhere. Technically litter doesn't decomposes into soil carbon.  As litter decomposes it releases CO2 and the dead organic matter        is broken into smaller more recalcitrant materials, which the models consider as soil carbon. In reality, of course, there is a continuum.

changed to "the decomposition of litter carbon is slowed"

18.      Page 4, lines 114-115. "As standard, JULES-C includes an implicit representation of nitrogen which has been extended to be fully interactive.". A sentence or two about how nutrient constraints on photosynthesis are implicitly modelled in JULES-C will be helpful.

Additional text to explain the implicit scheme added - "The philosophy behind the developments described here is to produce a parsimonious model to capture the established first order emergent response of N addition on growth which translates into leaf area and biomass without the complex and uncertain impacts on leaf physiology. Our approach is therefore to simulate the large-scale role of N limitation on vegetation carbon use efficiency (CUE - ratio of net to gross primary productivity) and soil carbon turnover. This is achieved by extending the implicit representation of N in the existing dynamic vegetation and plant physiology modules to be fully interactive. At the core of surface exchange in JULES is a coupled stomatal conductance photosynthesis scheme parameterised in terms of the maximum rate of Rubisco carboxylation, $V_{cmax}$ (mol CO$_2$m$^{-2}$s$^{-1}$). $V_{cmax}$ has a dependency on the leaf N concentration. Similarly, plant maintenance respiration has a dependency on leaf, root and stem N concentration \citep{cox1998canopy,cox1999impact,cox2001,clarketal2011}. Implicit within JULES, even in simulations excluding the carbon cycle is the parameterisation of plant tissue level N concentrations and associated allometry \citep{gmd-13-483-2020}. Simulations with an interactive carbon cycle therefore assume that enough N is available to meet vegetation growth and turnover. Here, we simply limit growth if not enough N is available. To do this requires a full representation of the N cycle in the land surface including a coupled soil carbon-nitrogen and inorganic N scheme."

19.      Page 4, line 120. "The vegetation nitrogen component captures the nitrogen limitation on the C stock, and ...". As described here the N limitation acts on NPP which is a C flux and not on the C stock.

changed to "nitrogen limitation on the net primary productivity, and includes retranslocation"

20.      Page 4, last sentence, line 126. " ... it slows the rate of litter decomposition INTO SOIL ORGANIC MATTER." Please consider removing the phrase in capitals.

Done

21.      Page 5, lines 129-135. I felt, it is little too early to introduce the seven JULES- CN parameters given that at this point in the manuscript, the parameterization themselves haven't been introduced.

Moved to the end of the model description section to summarise.

Also on line 130, Does " ... the effective solubility of nitrogen", refers to solubility in water.

Changed to "the effective solubility of nitrogen in water"

22.      Section 3.1. It seems the model's roots are in fact fine roots (since Rc = Lc in eqn(3)), and coarse roots and stem are included in the Wc term. Please make this clear.

Done

23.      For eqn (1) please specify the units of all terms. I suspect these are KgC m−2.

Added

24.      For eqn (2) what are the units of σl and Lb.

Done

25.      What are units of the individual terms in eqn (5) through (9) and the remaining equations.

Added

26.      Page 6, lines 160-178. This entire section is based on Figures 2 and 3 which form the backbone of specified C:N ratios and their variation with canopy height. It would be extremely helpful to know the basis of these relationships.

Section 3.1 has been updated to make clear these are the existing relationships that are implicit in the JULES model. The basis for these is now given in the text. "TRIFFID employs fixed allometry such that the split between leaf, root and stem carbon are defined by a single state prognostic variable that defines the total biomass. Biomass density increases via growth and is reduced by litter production and competition. Biomass can also increase by spreading through an increase in covered area. Nitrogen is implemented to limit growth and spreading such that the change in vegetation nitrogen cannot exceed that available. This section documents the vegetation model starting with the vegetation carbon and nitrogen structure (\ref{sec:struc}) including the additional complexity of labile nitrogen (\ref{phen}). The following section describes how growth and spreading is limited by nutrient availability (\ref{sec:allocup}. The final section describes how vegetation carbon and nitrogen is turned over by disturbance and competition (\ref{sec:litter})."

27.      Page 6, lines 180-181. "Biological nitrogen fixation (BNF) is ASSUMED TO BE THE largest natural supplier of nitrogen to the terrestrial ecosystem". Consider re- moving the words in capitals and including the word in bold. Fertilizer application is the largest anthropogenic N flux and BNF is largest natural flux.

Done

28.      Page 6, line 181. "Following Cleveland et al. (1999), the nitrogen fixation is determined as a proportion of the net primary production before nitrogen limitation (NPPpot)". This is incorrect. Cleveland et al. (1999) parameterized BNF as a function of actual evapotranspiration (AET) not NPP.

While we concur that ET is the primary parameterization described by Cleveland et al. (1999), we refer the reviewer to page 637 of Cleveland et al. (1999): "NPP could also relate to N fixation; NPP may be a proxy for carbon potentially available to fixers. The relationships between N fixation and modeled NPP are depicted in Figure 2…". It would be remiss to not cite Cleveland as this parameterisation is directly related to that work. We have clarified in the text that NPP is the secondary model from Cleveland et al. (1999).

Also, NPPpot is not defined anywhere close to this equation where it is intro- duced the first time. The first definition of NPPpot occurs on page 9, line 242, as "NPPpot supplied to TRIFFID represents

the potential amount of carbon that can be allocated to growth". Then a somewhat different definition occurs on page 19 which defines NPPpot as the NPP when nitrogen is unlimited. Isn't NPPpot just the NPP from the original framework without any reduction. I don't think, you do a calculation with unlimited N applied, per se.

Changed-We have revised the definition of NPP_pot to "potential amount of carbon that can be allocated to growth and spreading of the vegetation" and the response ratio to "the ratio of the potential amount of carbon that can be allocated to growth and spreading of the vegetation ($NPP_{pot}$) compared with the actual amount achieved in the natural state (NPP)" We have also added a sentence saying "the NPP_pot is defined in the same way as the net primary productivity in JULES before the explicit nitrogen cycle was included"

In context of BNF, and eqn (9), the parameter ζ is not listed in Table 1.

ζ is not dependent on pft so I don't think it is necessary to add it to table 1.

29.     Page 7, Table 1. It would be extremely help if nl0 is inverted and written as 1 in units of Kg C/Kg N so that the values are easily comparable to C:N ratios reported in literature.

Changed – this statement has been added to the caption: "$n_{l0}$ is the N concentration at the top of the canopy but is shown here as 1/$n_{l0}$ so that it is comparable to expected C:N ratios from the literature."

Also, nl0 is listed twice in Table 1 and please consider rewording "Top leaf N concentration" to "N concentration at the canopy top".

Changed

30.     Page 7, lines 188-189. "However, in JULES-CNlayered the vertical distribution of the fixed nitrogen in the soil depends on the root distribution ...". What does "fixed" refers to in this context. Also, at this stage in the manuscript it is not clear what does "depends on root distribution" means?

Changed to "However, in JULES-CN$_{layered}$ the $BNF$ is distributed vertically in the soil depending on the fraction of roots in each layer. If a soil layer is frozen there is no $BNF$ into that layer."

31.     Page 7, lines 201-203. "This distinction is inconsequential in the carbon only mode but is more critical when considering nitrogen interactions as the implication is that at all times the plant has enough nitrogen in reserve to maintain full leaf". From here on it becomes difficult to follow the logic used in the model. I am not able to understand what does "the plant has enough nitrogen in reserve to maintain full leaf" means?

This section has been restructured and clarified.

32.     Page 8, eqn. (10). I am confused here. Lb is introduced as a variable called balanced leaf area index but not explained what actually it means. In eqn (2), leaf C, LC is a function of Lb. In equation (9), leaf area index (LAI) (L) is also related to Lb through p. Somewhere here, there is the split of LC into labile and non-labile (the one which determines the actual LAI). Did I get this correct? How are L in eqn. (9) and LC in eqn. (2) related? Are they related through specific leaf area (SLA)?

This section has been updated. Now Lb is clearly defined, and it has been made clear this variable is the main mechanism that changing vegetation structure affects surface exchange. Units are now explicit.

33.     All through up to this point in the manuscript, the rate change equations for the vegetation N pool are not presented. At this point in the manuscript, I am still unclear what "retranslocation" means. Is this the transfer of resorbed N from leaves before they are shed. If yes, to which plant components?

The whole section has now been updated and clarified. Retranslocation is nitrogen being moved from leaves to the labile pool prior to leaf fall.

34.     page 8, lines 251-216. "During leaf-off the labile component is the equivalent of the retranslocated leaf nitrogen plus an additional store of nitrogen in preparation for the following bud burst". This sentence introduces yet another pool. It would be really helpful if all the pools and their rate change equations are properly introduced.

The rate change equations are included. The structure has been updated to clarify the pools and the implementation of rate changes.

35.     Page 8, line 29. "The mean canopy nitrogen content is described by ...". Please reword this to "The vertical distribution of leaf N content in the canopy is described by ...".

done

36.     Page 9, line 235. "Canopy Leaf C:N ratios are resultingly 44% higher than top leaf ratios". I am unable to understand this. Does "canopy leaf C:N ratios" refers to mean canopy leaf C:N ratio or the vertical profile of C:N ratios along the canopy depth starting from the top.

The mean canopy C:N ratio. Text clarified.

If leaf N content in the leaves at the top of the canopy is higher and decreases exponentially, and if C content is uniform than it implies that C:N ratio of leaves is lower in the leaves at the top of the canopy and higher at the bottom. Integrating eqn (12) over LAI yields $\int L \, nl0 \exp(-knz) \, dz = nl0 \, 1 \, (1 - \exp(-knL))$ which implies that the mean C:N will depend on the LAI, L. So I am unclear where does the number 44% comes from.

Agreed. However, in the Mercado implementation there is no dependence on total LAI. $z$ is the fraction of the canopy above a point in the canopy and is therefore independent of LAI. The implication of this is being explored elsewhere.

37.     Page 9. Section 3.1.4. The term $\Lambda lc$ in eqn. (13) is not defined and only when the reader reaches eqn. (21) it is clear what this term is. Similarly for $\Lambda ln$.

Updated and clarified.

38.      Page 10. Line 263. Λln is defined as the retranslocation of nitrogen from leaves and roots into the plant labile pool. I am not sure how does it relate to p in equation 10 which is also related to retranslocation.

Clarified in the text. Here, retranslocation is used to define the flux of carbon. In eq 10, the retranslocation coefficient is used to parameterise the labile store. Under the assumption that higher retranslocation corresponds to a greater store.

39.      Where is $\Psi_c$ from equation (13) defined? Is this what $\Psi$ is in eqn. (17)?

Apologies, this was a typo and should be $\Psi_g$.

40.      Page 10. Line 271. "The nitrogen available for growth is the total available nitrogen multiplied ...". Please reword this as "The nitrogen uptake used for plant growth is the total nitrogen uptake multiplied ...". I think, that's what is meant here. Available N sounds like the N available in the soil inorganic pool that can be potentially taken up by plants.

done

41.      Page 10. Line 272. "Equations 13 and 15 are then solved by bisection such that the nitrogen uptake for growth ($\Phi_g$) is less than or equal to the available nitrogen...". Do you mean the bisection method to find root of an equation? This and remaining part of this paragraph is difficult to understand since there is no $\Phi_g$ term in either equation (13) or (15).

This section has been updated to explain more clearly the solution to the equations presented.

In addition, since units of the various terms are not provided it is difficult to follow the equations on page 10.

42.      Page 10, line 282. "... and $N_v/C_v$ defines the whole plant C:N ratio ...". You mean $C_v/N_v$?

Now given as the inverse of the whole plant C:N ratio

43.      In the absence of the competition module of the TRIFFID model properly de- scribed it is difficult for a reader to know what does "density-dependent litter production" and "density-dependent componennt for intra-PFT competition for space" means in Section 3.1.5. Please consider introducing this in a sentence or two.

Done

44.      Page 11, please define $\Lambda_c$ and $\Lambda_n$ in words explicitly where the are first intro- duced. $\Lambda_c$ was introduced in equation (22) but not defined until next page near eqn. (28).

Done

45.      Page 11, lines 310-311. "The effect of nitrogen limitation on the litter carbon flux is captured in the excess carbon term $\Psi_i$". Throughout the manuscript there is no expression for $\Psi_i$ so it's difficult to understand it. I do understand based on what is written in the manuscript that it the excess C that cannot be used. So it must be related to N uptake, allocation fractions for C, and specified C:N ratio of the three C pools.

Made it clearer in the text that the subscript, i, is used to indicate PFT levels and is defined in previous equations.

46.     Page 12, line 339. "βR depends on soil texture". I don't think, this dependence can be too strong. Can you please mention the typical value of βR.

This line has been added: $(1-\beta_R)$ is the fraction of soil respiration that is emitted to the atmosphere - this depends on soil texture and ranges from 0.75 for a clay soil to 0.85 for a soil with no clay content

47.     The rate change equations for litter and soil C pools are helpful. Similar equations for vegetation C and N pools would be so helpful.

The rate changes are in Eq 12 and 14. This section should be a lot clearer now.

48.     Page 13, line 349-350. "Input into the BIO and HUM nitrogen pools comes from the total immobilisation of inorganic nitrogen into organic nitrogen where Itot = IDPM + IRPM + IBIO + IHUM ". Itot is divided into BIO and HUM pools. Since BIO is the microbial pool shouldn't all immobilization end up there.

In reality, carbon (and therefore also nitrogen) should go from litter pools -> microbe pool -> Humified pool (since HUM is made of microbial necromass). But in RothC, carbon can go straight from litter to HUM. Therefore the nitrogen fluxes must follow this as well. This sentence now says: "Following the framework of the RothC model, input into both the $BIO$ and $HUM$ nitrogen pools is from the total immobilisation of inorganic nitrogen into organic nitrogen where $I_{tot} = I_{DPM}+I_{RPM}+I_{BIO}+I_{HUM}$ (in kg\,[N]\,m$^{-2}$ s$^{-1}$)"

49.     Page 13, eqn (33). Does the subscript i still refers to PFTs?

Changed i to p so soil carbon pools are represented by subscript p and vegetation pfts are always represented by subscript i.

50.     Page 13, line 365. " ... the respired fraction (βR) and the C to N ratio of the destination pool ...". This is confusing since on line 339 (1-βR) was referred to as "the fraction of soil respiration that is emitted to the atmosphere".

This was a mistake in line 365 - the fraction respired to the atmosphere is (1-beta_r)

51.     Page 14, line 371. " ... where i is one of RPM or DPM." Please use a different subscript here since you have used i previously to represent PFTs.

Changed i to p so soil carbon pools are represented by subscript p and vegetation pfts are always represented by subscript i.

52.     Pages 13 and 14. The FN terms in eqn (36) limits the respiration of the DPM abd RPM litter pools. So it is unclear to me why FN would depend on IBIO and IHUM in eqn. (37).

Respiration is carried out by microbes so they won't decompose as much of the DPM/RPM pools if they haven't got enough nitrogen to convert that carbon into BIO/HUM. The total amount of nitrogen they have available depends on I_BIO and I_HUM because M_BIO - I_BIO (and similarly M_HUM-I_HUM) is the net mineralised nitrogen from the turnover of BIO and HUM. This ahs been added: "Respiration is carried out by microbes who require sufficient nitrogen to convert the $RPM$

and $DPM$ pools into $BIO$ and $HUM$ pools. This nitrogen is available from the net mineralised nitrogen from the turnover of $BIO$ and $HUM$ pools."

In this same equation, I am also unclear what is Navail at this point in the manuscript. As with several other terms, the terms are introduced but their expressions are mentioned or the terms clarified much later which makes it very difficult to follow the logic. It is only further down in eqn. (51) that Navail is clarified.

A pointer to Navail which has been redefined as Ninorg for the bulk case has been added to help the document flow better. Fn has also been added to the vertically resolved case because it has a slightly different definition.

Also, in eqn. (37) what happens if DDP M or DRP M are negative? Is this possible, since minrealization can be more than immobilization?

They are always positive because the values for CN_soil are << CN_dpm/rpm. If they were negative, Fn should just be 1 because there would be more mineralisation than immobilisation from **all** pools. A sentence to address this has been added to the paper: "The demand is always positive because the C to N ratio of soil is very much less than the C to N ratio of the $DPM$ and $RPM$ pools"

53.     Page 14. Similarly in eqn. (39) can Itot be more than Mtot making Ngas negative.

Fluxes will have been limited by Fn to make sure this isn't negative. If it hits the minimum pool size, it calculates a correction term (neg_n) and that correction term is then included as a negative gas flux. But that is applied just as an 'extra' gas flux and not applied to minl and immob. So Eq 39 is never negative, but gas flux can be, if that makes sense! This has been added: ". f$_N$ limits the nitrogen fluxes so that (M$_{tot}$ - I$_{tot}$) is always positive. However, if pool sizes become too small N$_{gas}$ could become negative to ensure nitrogen is conserved."

54.     Page 15. Eqn.(41). Is fdpm used here different from fDPM used in eqn. (25).

Changed – they are the same.

55.     Page 15. Lines 416-417."The litter inputs are distributed so that they decline exponentially with depth, with an e-folding depth of 0.2 m". With this parameterization can litter enter a soil layer even if there are no roots in that layer.

This is correct and added "This means that litter can enter a soil layer even if there are no roots in that layer":

56.     Page 15. Line 423. Please consider using "bulk" or "single layer" instead of "non-layered".

Changed

57.     Please consider using another term for gaseous losses rather than turnover.

Changed "additional inorganic gas loss term"

58.     Page 16, Lines 433-434."Without this additional turnover available N may in- crease excessively, potentially due to excessive biological fixation in regions that are generally unlimited". What does "regions that are generally unlimited" means?

Changed "Without this additional gas loss term available N may increase excessively, potentially due to excessive biological fixation in regions where the $NPP$ is very close or equal to the $NPP_{pot}$"

59.    Page 16. Line 434-435. "In the current model configuration this parameter is set to 1.0 (360 day-1) such that the whole pool turns over once every model year". Do you mean 1.0 year−1 which would translate to (1/360) day−1 and not 360 day−1? Also, since the time step of the biogeochemistry is the same as for TRIF- FID (i.e. 10 days) there has to be ΔT somewhere. And, I suspect, 360 is used and not 365 since the calendar year in the UKESM model is 360 days. Correct?

Changed to "1/360 (day$^{-1}$)". Indeed 360 days represents a year in UKESM.

60.    Page 16, line 436. "This results in an effective saturation limit of 0.002 KgN m−2...". Not clear - saturation limit of what?

Turnover is limited by typical fluxes in and out of pool. In practise it never gets bigger. This line has been deleted.

61.    What are the units of β in eqn. (47). Just above eqn. (47) β is said to be assigned "a value of 0.1 based on sorption buffer coefficient of Ammonia although here it represents the sorption of all inorganic nitrogen species". Note here that typically only NO3- leaches into the runoff and not NH4+ so please consider modifying this sentence.

Beta is dimensionless (added) and the sentence is changed to: '$\beta$ is assumed to have a value of 0.1 and in JULES-CN represents the combined sorption of all inorganic nitrogen species \citep{wania2012carbon}.'

62.    Page 16. Eqn. (48). Isn't f1 simply the fraction of roots in each soil layer. And again, f2 is not defined or described here but further down in eqn. (53).

We have reformatted this section in an attempt to make things clearer. It now includes the following straight after the equation to act as a better signpost. Each of the modified components of Equation \ref{eq:ninorg} are discussed in detail below. The additional parameters required are $f_{R,i}(z)$ - the fraction of roots in each layer (Equation \ref{eq:norm_root}); $f_{I,i}(z)$ - the fraction of available inorganic nitrogen in each layer (Equation \ref{eq:frac_avail}) and $N_{flux}$ - the transport of inorganic nitrogen from the layer by the soil water fluxes (Equation \ref{eq:layer_leach}).\\

63.    Page 16. Line 453. "where froot(z) is the volumetric root fraction at a given depth". You mean "for a given soil layer" as opposed to "at a given depth". And, an i subscript seems to be missing here. Although, I wouldn't suggest using i which has been used for PFTs, DPM or RPM, and now soil layers. Very confusing!

Definitely confusing! i is actually for pft here. We have clarified and change the soil carbon pools to a p! Also added the I where required on the froot term.

64.    Page 17, eqn. (50). Is the parameter τresp tuned so that Nturnover is similar in the "bulk" and "layered" versions.

This has been added: Here $\tau_{resp}$ was tuned to give a realistic estimate of soil carbon in a vertically resolved version of JULES-C as in \cite{burke2016gmd}

65.    Page 17. Eqn (51). Assuming, the subscript i represents the PFT shouldn't there be (z) term here to indicate the nitrogen availability in each layer.

Added the z to this equation. Ive tried to clarify the i represents PFT.

66. Page 17. Eqn (52). I am unable to follow eqn. (51). Looking at eqn. (51) the term in parantheses in eqn. (52) should be zero since froot,i Nin = Navail,i from eqn. (51).

Indeed this is correct – Equation 51 is for an equilibrium state whereas equation 52 is for a transient state.

The value/units of γdiff is also confusing. I am not sure what 100 [360 day]−1 means.

Units of this parameter have been changed so it is per day instead of per 360 days.

67. Page 17, lines 471-474. "Any fixation goes directly into the available pool, and other fluxes are simply added according to the ratio of the available to total inor- ganic N pools at equilibrium (thus the available pool would always follow Equation 51 were it not for the fixation and uptake by plants)". I am sorry but I am unable to follow this sentence.

This has been rephrased to make it clearer – particularly focusing on the definition of equilibrium.

68. Page 17. Eqn. (54). In the absence of its units, I am not sure if the term dzn is a single variable or do you mean Δzn. And, I have no clue, what zn is at this point in the manuscript.

We have rewritten this equation and text: Leaching is now done in a process-based manner, where the inorganic N is transported through the soil profile by the soil water fluxes. For any given soil layer $n$ of thickness $\delta z_n$, the inorganic N flux (N$_{flux,n}$) of layer $n$ is given by:

\begin{equation}\label{eq:layer_leach}

  N_{flux,n} = \beta \delta z_{n} \frac{d}{dz} \left( W_{flux,n} \frac{N_{in,n}}{\theta_n} \right)

\end{equation}

where $\theta_n$(z) is the soil water content of layer $n$ in kg m$^{-2}$ and $W_{flux,n}$ is the flow rate of the water through soil layer $n$ in kg m$^{-2}$ s$^{-1}$. Multiplying by $\delta z_n$ gives the change in N content for each layer, $n$. The total leaching is then the sum of all nitrogen that leaves the soil by lateral runoff or out of the bottom soil layer.\\

69. Page 17. Line 483. "... is then the sum of all nitrogen that leaves the soil by lateral runoff ...". Does the lateral runoff from each layer mean that JULES is capable of producing runoff based on slope of the ground? Please clarify what exactly lateral runoff means.

We have removed the more specific details of how the water leaves the soil as I think it complicates further an already complicated paper. However, JULES has a version of TOPMODEL which can be switched on an generate lateral flows (https://agupubs.onlinelibrary.wiley.com/doi/full/10.1029/2004GL020919)

70. Page 18. Lines 501-502. "They were spun up by repeating the time period 1860- 1870 ...". This is confusing. Please consider rewording as "The models were spun up by using the meteorological data for the period 1860-1870 repeatedly..."

Changed

71. Page 19. Lines 522-524. "The main difference is the present-day NPP which is 12% higher in JULES-C than in JULES-CN. This is a direct consequence of nitrogen limitation which restricts the ability of the plants to utilise all of the carbon". No this is the direct consequence of JULES-CN reducing NPP. I don't think, it is necessary to spin this in a more biological way.

Changed

72.    In Figure 4, it would be really useful to see separate estimates for mineralization and immobilization. In its current form, only net mineralization is reported.

This figure has been updated and now includes both immobilisation and mineralisation.

73.    Page 20. Lines 580-582. "This [CUE] represents the capacity of the plants to allocate carbon from photosynthesis to the terrestrial biomass". I don't think this sentence is entirely correct. Since CUE is the fraction of GPP converted to NPP, it is a measure of autotrophic respiration.

Changed to: Plants with a higher CUE have a lower autotrophic respiration and allocate more carbon from photosynthesis to the terrestrial biomass and vice-versa.

74.    Page 20, line 582-583. "In the model nitrogen limitation restricts the ability of plants to allocate carbon and reduces the carbon use efficiency". Here again, the "restriction of ability of plants to allocate carbon" appears as if carbon is there but some how plants can not allocate it. In contrast, as JULES-CN is designed, there is simply less carbon to be allocated. I don't think, JULES' allocation module has been changed in JULES-CN to limit how much C flows to different components.

Changed to: In JULES-CN there is less carbon available to be allocated because it is constrained by the amount of N present. This reduces the carbon use efficiency.

75.    Page 21. Line 596. " ... by structural changes in the vegetation in particular ...". Please clarify if structural changes refer to changes in vegetation height, LAI, and rooting depth.

 This is mainly the vegetation distribution – this been made clearer in the text.

76.    Page 22. Lines 626-628. "There remains a significant underestimate of NEE in the years following the Pinatubo volcanic eruption ...". Please make it explicit in which year Pinatubo erupted since it's not marked in Figure 12.

Pinatubo erupted in 1991 – this has been added.

77.    Page 22. Line 646. Please change "tome" to "time".

 this has been changed.

78.    Page 23. Line 656. "In this model, nitrogen limitation affects NPP and how the carbon is allocated ...". As mentioned above, I think, it's more appropriate to  say how much C is allocated since the underlying C allocation module has not changed between JULES-C and JULES-CN.

this has been changed.

3    References

1.    Cox, P. (2001). Description of the "TRIFFID" Dynamic Global Vegetation Model. Met Office, Hadley Centre Technical 24.

2.    Field, C. and Mooney, H.: The Photosynthesis-Nitrogen Relationship in Wild Plants, Biol. Int., 13, 25–56, 1986.

3.    Liang, J., Qi, X., Souza, L. and Luo, Y.: Processes regulating progressive nitrogen limitation under elevated carbon dioxide: a meta-analysis, Biogeosciences, 13(9), 2689–2699, doi:10.5194/bg-13-2689-2016, 2016.

4.      McGuire, A. D.,  Melillo, J. M.  and  Joyce, L. A.:    The role of nitro- gen in the response of forests net primary production to elevated atmo-

GMDD

spheric carbon dioxide, Annual Reviews Ecol. Syst., 26(1), 473–503, doi:10.1146/annurev.es.26.110195.002353, 1995.

5.      Ochoa-Hueso, R., Maestre, F. T., Ríos, A. [de los, Valea, S., Theobald,

M. R., Vivanco, M. G., Manrique, E. and Bowker, M. A.: Nitrogen de- position alters nitrogen cycling and reduces soil carbon content in low- productivity semiarid Mediterranean ecosystems, Environ. Pollut., 179, 185–193, doi:https://doi.org/10.1016/j.envpol.2013.03.060, 2013.

---

## Referee Report (RR1)

Re-Review of JULES-CN: a coupled terrestrial Carbon-Nitrogen Scheme (JULES vn5.1) by Wilshire et al.

This is my second review of this manuscript. This manuscript is in much better form than it was before. It is easier to read, easier to follow, and the additional plots and figures that are included provide a much more coherent story than before. I am sure that the authors themselves would have found this exercise useful. I appreciate the effort put by the authors in revising their manuscript.

I went through the manuscript again in its entirety. However, I am sorry that there are still some issues that I feel need to be addressed. One of these issues appears serious and makes me wonder if the implementation of the model itself is correct. I have concerns with some equations as well. As last time, I am summarizing my major comments here but I am also attaching a scanned copy of an annotated version of your manuscript that you may refer to for several other comments. I have tried to keep my handwriting clean.

**Major comments**

**1. How does N demand vs uptake works?**

I am still confused about how nitrogen limitation works including through the use of the term $F_N$ in equation 39. On line 322, page 11, it reads

> N demand for growth. If the N demand is less than that available ($\Phi_{g,i} < (1-\lambda_i)\, N_{avail,i}$) growth is unlimited and the fluxes updated accordingly. Where N is limiting, growth N uptake is set equal

but I am confused about the role of $N_{avail}$. In the above sentence note that $\Phi_{g,i}$ has to be a flux not a pool/reservoir/store. This is the N uptake rate in equation 14. Below equation 14, you have its units written as if it were a pool. This is not correct. Note the $dt$ term in denominator on left hand side of equation 14. $\Phi_{g,i}$ cannot have units of a pool. In contrast, $N_{avail}$ appears to be a pool as indicated below.

In JULES-CN, on a PFT basis, the N available for plant uptake ($N_{avail,i}$) is the inorganic soil N pool ($N_{in}$) split equitably between the PFTs assuming there is no differential ability between PFTs
330 to acquire N. The available N in JULES-CN$_{layer}$ is more complicated taking into account the soil profile and is discussed in Section 3.3.2.

How can a flux be compared to a pool? Every time step as the model photosynthesizes and the C flux comes in from the atmosphere a corresponding N flux is required from soil. This is the nitrogen demand. In my mind, and I think in reality, whether this N demand is met or not depends on the rate of nitrogen uptake which in turn depends on transpiration (for the passive N uptake) and the ability of fine roots to uptake additional N if passive uptake is not sufficient (active N uptake). The pool size of inorganic N is a big number. It is almost always bigger than the demand at any given time step. The fact that the inorganic N pool size is bigger than the flux does not imply that all that N is available to be taken up to meet the instantaneous demand. In my mind, this logic cannot be used to determine if N is limiting or not. Unless, there is something the manuscript does not convey and I am misinterpreting the whole thing.

Similarly in equation 40, $N_{in}$ is the inorganic N pool size while all the other terms are fluxes. It does not make any sense to me how the fluxes and pool terms can be mixed together. It makes wonder if the model itself is implemented correctly or not.

**2. Errors in equations**

a. I have already mentioned that units in equation (14) are inconsistent.

b. Please see below your equation 50 that finds fraction of litter in each soil layer z.

$$f_{lit}(z) = \frac{\exp(-\tau_{lit}z)}{\int_0^{z_{max}} \exp(-\tau_{lit}z)dz}$$

Shouldn't this be written something like

$$f_{lit}^n = \frac{\int_{z_n}^{z_{n+1}} \exp(-\tau_{lit}z)dz}{\int_0^{z_{max}} \exp(-\tau_{lit}z)dz}$$

In equation (50) and elsewhere authors have used $z$ both as layer index and a continuous variable on which integration is performed. This is very confusing. See my example above where, in my mind, it makes more sense to use $n$ as a layer index and $z$ as the continuous variable (soil depth from top in this case). $z_n$ in my attempt thus represents the depth to the top of the $n^{th}$ layer.

c. In equation (55)

$$\frac{dN_{in}(z)}{dt} = N_{dep} + \sum_i v_i BNF_i f_{R,i}(z) - \sum_i v_i \Phi_i f_{I,i}(z) + M_{net}(z) - N_{flux}(z) - N_{gasI}(z)$$

(55)

$N_{dep}$ should only be applicable to the first layer. Correct? If yes, this equation needs to be written slightly differently to reflect this. Note that $z$ is used here as layer index.

[revised manuscript text omitted]

*I think, if you were to show Fig 2 for C (kg c/m²) it will become obvious. OR just add a sentence here.*

235 ### 3.1.2 Labile C and N: Phenology and Mobilisation

The total leaf C pool per PFT ($L_{c,i}$, Equation 2) varies allometrically with the vegetation C state on both short (seasonal) and long (centennial) timescales but not with changes in phenological state. Implicit within TRIFFID is a labile leaf C pool that acts as a reserve of C during spring and a store during fall. $L_{c,i}$ therefore includes a labile pool from which C can be mobilised during leaf out 240 plus an allocated pool representing the actual LAI. The labile pool is zero at full leaf out and at the allometrically defined maximum during the no leaf period. As part of the N coupling we introduce the ability for plants to retranslocate some of the allocated N to the labile N pool according to the phenology. The new parameterisation of retranslocation and labile N is therefore dependent on the

leaf phenological state as well as the fixed stoichometry. In JULES, leaf phenology is controlled by
a second state variable ($p_i$) which relates the LAI ($\mathcal{L}_i$) at any moment in time to the balanced leaf
area index ($\mathcal{L}_{b,i}$).

$$\mathcal{L}_i = p\mathcal{L}_{b,i} \qquad \qquad \qquad (9)$$

*[handwritten: → phenological]*

where $p_i$ is a scalar between 0 and 1 that describes the phenological state of the system (Clark
et al., 2011). For evergreen plants $p_i$ is a constant of 1. The two state variables $\mathcal{L}_{b,i}$ and $p_i$ combine
to define the vegetation state for each PFT $i$. Using the phenological state we extend the equivalent
approach to leaf C such that the leaf N pool ($L_{n,i}$) has fixed allometry dependent on the phenological
state and the magnitude of leaf retranslocation. We introduce this simple parameterisation under the
assumption that higher leaf retranslocation during autumn implies a higher labile N store. The leaf
N pool therefore becomes:

$$L_{n,i} = p_i n_{lc,i} L_{c,i} + (1 - p_i)(\frac{1+\lambda_{l,i}}{2})n_{lc,i}L_{c,i} \qquad \qquad (10)$$

where $\lambda_{l,i}$ is the dimensionless leaf N retranslocation coefficient and $n_{lc,i}$ is the mean canopy N
content (Equation 11). Here $\lambda_{l,i}$ is set to 0.5 for all PFTs (Zaehle and Friend, 2010). The formulation
of the labile pool, in this configuration, means that around half of the N required for full leaf-out is
taken from retranslocation with a further quarter acquired during the dormant phase while the rest is
acquired during the active period.

*[handwritten: unclear]* *[handwritten: → from where to where? leaf → root/stem?]*

*[handwritten right margin: units ?? Actually $n_{lc}$ looks like concentration like $n_{l0}$ with units of $gN/gC$]*

JULES assumes a process-based scaling-up of leaf level photosynthesis to the the canopy level. In
both the JULES-CN and JULES-CN$_{layer}$ configurations, to be consistent with the JULES-C model,
we assume a multi-level canopy with leaf N decreasing exponentially through the canopy (*CanRad-
Mod 5*). The plant physiology routines uses this assumed distribution to calculate penetration through
the canopy and photosynthesis on individual layers before scaling back to the canopy (Clark et al.,
2011). In the application here, we use this distribution to be fully consistent with the physiology.
The vertical distribution of leaf N content in the canopy is described by (Mercado et al., 2007):

$$n_{lc,i}(d) = n_{l0,i}\exp(-k_{n,i}d) \qquad \qquad (11)$$

*[handwritten: ←]*

where $k_{n,i}$ is a constant representing the profile of N and $d$ represents the fraction of canopy above
the layer. Based on observed N profiles in the Amazon basin (Carswell et al., 2000), a value of 0.78
for $k_{n,i}$ was found (Mercado et al., 2007). Equation 11 is independent of leaf area and therefore
equates to a constant of proportionality relating PFT-specific top leaf N to the mean canopy N con-
centration. The mean canopy leaf C:N ratio is consequently ∼44% higher than the top leaf ratio.

*[handwritten right margin: I got confused since $n_{l0}$ is $gN/gC$ and C:N is $gC/gN$. I get it now but consider rewording or explaining]*

*[handwritten bottom notes:*
*$d=0$, $d=1$ graph of $e^{-k_n \cdot d}$*
*means concentration decreases with canopy depth.*
*At $d=0$, $n_{lc} = n_{l0}$*
*$d=1$, $n_{lc} = n_{l0} \cdot e^{-k_n}$ ]*

**3.1.3 Vegetation Growth and Allocation**

[revised manuscript text omitted]

ing to a smaller pool size. *organic matter* OK

*Please flip Fig 5 sideways.*

*First show NPP & GPP, and then veg and soil C zonal distributions.*

715

Carbon use efficiency (CUE) is defined as the ratio of net C gain to gross C assimilation during a
given period (NPP/GPP). Plants with a higher CUE have a lower autotrophic respiration and allocate
more C from photosynthesis to the terrestrial biomass and vice-versa. In JULES-CN there is less C
available to be allocated because it is constrained by the amount of N present. This reduces the C use
720 efficiency. Figure 6 shows the zonal total GPP and NPP for JULES-CN and JULES-C. As expected
from Figure 4 the NPP and GPP have very similar latitudinal profiles for the two model configura-
tions. Both JULES-C and JULES-CN have a higher GPP in the tropics than the observations but they
are more comparable in the extra-tropical latitudes where the GPP tends to be smaller. The NPP in
JULES-CN is less than JULES-C and generally closer to the MODIS observations particularly in the
725 tropics. Figure 6 also shows the zonal mean CUE. JULES-CN has a lower CUE than JULES-C for

*Is this an indication that reducing NPP and not GPP  may not be a good strategy in response to N limitation.*

[revised manuscript text omitted]

*The color bar colours in Fig 9 are very hard to follow. Please consider using different colours scale. In addition, NPP achieved / NPP pot seems more intuitive*

*Fig 9c implies no N limitation in 1860 - 1900. Can you comment if this is realistic?*

Figure 10 shows the biome-based response ratio of net primary productivity. All biomes have a

805 response ratio of greater than 1 in both the model and observations which means that adding extra N to the system will enhance the NPP achieved. Globally the response ratio is lower than the observations but for the majority of the biomes including the tropical forests and the tundra the model response ratios fall within the range of uncertainties of the observations. However, LeBauer and Treseder (2008) suggests the tropical forest is somewhat N limited, whereas in JULES-CN tropical

810 forest is not a N limited biome. Phosphorus has long been considered as the most limiting nutrient in tropical regions (Yang et al., 2014), therefore we expect JULES to simulate a larger response ratio in the future once a phosphorus cycle is added.

In the model the soil C decomposition can be limited when the N available in the soil is less than

815 the N required by decomposition. This process does not play a major role in our simulations.

**5.4 Nitrogen stocks and fluxes**

The zonal profile of soil organic nitrogen (Figure 11) shows a similar distribution to the soil organic C (Figure 5) reflecting the relatively consistent C to N ratio of the soil within the model. CN$_{soil}$ -

820  the C to N ratio of the HUM and BIO pools - is a spatially constant parameter set to 10 in these
     simulations. The observed soil N content is slightly higher at all latitudes than simulated by JULES-
     CN particularly in the northern tundra region. This is likely caused by the turnover times of the soil
     being too fast (Figure 8) leading to not enough soil N. In addition the C to N ratios in JULES-CN
     are too small for the northern high latitudes (mean of ∼14) whereas up to 25% of soils in tundra
825  regions are peat with C to N ratios of around 30 (Hugelius et al., 2020). In contrast to the zonal
     distribution of soil organic nitrogen, the soil inorganic nitrogen in JULES-CN is larger in the tropics
     than in the northern high latitudes. Figure 12 shows the net soil N mineralisation fluxes are large in
     the tropics and smaller in the northern regions. This is reflected in the spatial distribution of the N
     uptake. As might be expected the spatial distribution of the N uptake as a fraction of N demand is
830  similar to the N limitation shown in Figure 9. Biological N fixation and N gas losses are an order
     of magnitude smaller than the N uptake and net N mineralisation. However, again the spatial pat-
     terns are very comparable. N leaching is generally very small except in parts of south America and
     south-east Asia. Figure 13 shows a slight increase in the N demand and N uptake over the twentieth
     century associated with the increase in vegetation growth (Figure 6). Similarly there is an increase
835  in the BNF which is parameterised such that it is proportional to the NPP.

**5.5  Impact of vertical discretisation of soil biochemistry**

This section discusses the differences between JULES-CN and JULES-CN$_{layer}$. In general over the
tropics and southern latitudes, JULES-CN$_{layer}$ is very comparable to JULES-CN. The majority of
840  the differences occur in the northern regions where there is soil freezing–either permafrost or sea-
     sonally frozen soils. The reduction in global mean tree covered area seen in Figure 3 is caused by
     a reduction in the boreal regions which have a larger proportion of shrubs and grasses in JULES-
     CN$_{layer}$. In the higher latitudes the soil in JULES-CN$_{layer}$ also has more organic C (Figure 5). This
     increase in soil organic C represents a store of permafrost carbon more comparable to the carbon
845  found by Batjes (2014) and Carvalhais et al. (2014). This build up of carbon in JULES-CN$_{layer}$
     occurs because the decomposition deeper in the soil is reduced with the lower soil temperatures at
     depth - the soil C in JULES-CN only respond to the soil temperatures near the surface which are
     warmer. This also causes in increase in the residence time of the soil carbon shown in Figure 8(b).
     The modelled soil C residence time in JULES-CN$_{layer}$ is now much longer and more comparable to
850  that observed.

The spatial distributions of N fluxes in JULES-CN$_{layer}$ (not shown) are very similar to those
of JULES-CN. In addition, the time series of changes in N fluxes over the twentieth century are
also comparable (Figure 13). The main differences are in the N gas loss which is larger in JULES-
855  CN$_{layer}$ and the N leaching which is larger in JULES-CN. Figure 11 shows an increase in both

*[Handwritten margin note:] ← Doesn't this mean that the model has more N ⚖ than obs per unit C? How can this be used as an argument for not enough N at northern high latitudes.*

*[Handwritten annotation near line 826:] (Fig 11b)*

[revised manuscript text omitted]

---

## Author Response (AR2)

JULES-CN: a coupled terrestrial Carbon-Nitrogen Scheme (JULES vn5.1)

Response to Will Weider

*My main outstanding question is how much of the reduction in global productivity, biomass, and IAV with the new C-N model comes from actual N limitation on living biomass and the intended declines in CUE associated with N limitation vs. simply killing off plants by imposing N limitation. Specifically, the N model makes the bare ground bias in JULES even stronger (relative to the C-only model (Fig. 3). Thus, are lower C flux and stocks simulated by JULES-CN simply a product of having more dead plants in the model? It seems like these two points matter, especially if JULES-CN is going to be included in UKESM1 simulations where potential biases in bare ground fraction may influence the land energy and water balance in ways that feedback on the atmosphere and climate.*

**This paper documents the addition of a nitrogen cycle to JULES and how it modifies the carbon cycle. This is just once component of the configuration of JULES that is used in UKESM (JULES-ES). Other additions include a new competition scheme and additional PFTs. The combination of these three components are included in UKESM. Here, we have not done any model specific tuning to improve the land cover fractions  and UKESM has very good representation of the land cover fraction (UKESM1: Description and Evaluation of the U.K. Earth System Model - Sellar - 2019 - Journal of Advances in Modeling Earth Systems - Wiley Online Library – Figure 15). Therefore, the bare ground bias was reduced by the model specific tuning for UKESM1. It is common practice (HadCM3LC, HadGEM2-ES, UKESM1) to tune these uncertain disturbance parameters to achieve a good vegetation distribution.**

**In this description of the component nitrogen model the changes described in productivity and biomass are a direct from implementing Nitrogen. The changes in biomass are a direct result of reduced net productivity. As noted, reduced biomass generally leads to a reduction in productivity. As JULES-CN is a fully dynamic model it is near impossible to separate the initial change in productivity from a biomass response. However, as no vegetation parameters are adjusted it is the case that any changes are driven by the productivity response to the inclusion of an N scheme.**

*On a related note, I appreciate the synthesis of many results in zonal mean plots and annual time series, but wonder if there are insights or information that could be gained from showing full spatial maps of initial results (e.g. vegetation cover, GPP, or LAI in the CN model and their difference from the C-only results?). Maybe it makes sense to put these in supplemental material, if the authors are worried about cluttering up their results.*

**At this stage of the review process we have not undertaken this task. Full spatial maps of the full UKESM configuration are being made available in the literature via model evaluation and assessments as part of CMIP6.**

*Line 80 –* **new GCP paper has been cited.**

*It should also be noted that comparisons with GCP2019 estimates of NEE and IAV are themselves a modelled product from the TRNEDY simulations, of which JULES-CN participated. These are quasi-independent estimates, not results residual of bookkeeping methods, as previously presented –* The following text is in the paper - **To avoid the circularity of using GCP estimates of NEE which are themselves derived from land-surface models, we instead calculate the GCP estimates of NEE as the residual of the best estimates of the total emissions from fossil fuel ($FF$) plus land-use change ($LU$), and the rate of increase of the carbon content of the atmosphere ($F_a$) plus the ocean ($F_o$.** and the observational-based IAV is not dependent on the TRENDY

Response to Reviewer 3

Major comments:

1. *N demand vs uptake*

**Thanks for pointing this out. The code is implemented correctly but unfortunately the documentation needs correcting. This has been clarified in the manuscript with Dt, the given model timestep added to ensure the units are consistent. N{avail}/Dt is indeed a maximum uptake of inorganic nitrogen in that timestep. In this implementation it is assumed to be the equivalent of the entire pool taken up over the model timestep.**

**The text now reads:**

**{In JULES-CN the N available for plant uptake for each PFT $i$ ($N_{avail,i}$ in $kg\,[N]\,m^{-2}$) is the the inorganic soil N pool ($N_{in}$ in $kg\,[N]\,m^{-2}$) split equitably between the PFTs assuming there is no differential ability between PFTs to acquire N and the whole pool is available for uptake during the model timestep. The available N in JULES-CN$_{layer}$ is more complicated and takes into account the soil profile. This is discussed in Section \ref{sec:ninorgvert}.}**

**And:**

**If the N demand is less than the available N in a given timestep ($\Delta t$) ($\Phi_{g,i}$ $<$ (1-$\lambda_i$) $N_{avail,i} / \Delta t$) then growth is unlimited and the fluxes can be updated accordingly. Where N is limiting, growth N uptake is set equal to the available N ($\Phi_{g,i}$ $=$ (1-$\lambda_i$) $N_{avail,i} / \Delta t$) and the excess C for growth $\Psi_{g,i}$ can be derived.**

**See also the response L450 and Eq 40.**

2. *Errors in equations*

   a. *Units in equation 14:* **Units associated with Eq 14 are updated to make clear they are fluxes.**
   b. *response to query about equation 50*: **Eq 50 is now revised in the revised manuscript.**
   c. *response to query about equation 55:* **Equation 55 has been updated to reflect the fact the Ndep is only added to the top layer.**

*All minor textural changes suggested in the handwritten annotated document have been made.*

*Line 12 - N limitation can slow soil decomposition* - Perhaps it is later but of C to N or soil organic matter is fixed how does this work? – **Indeed this is discussed later. The C:N ratio of two of the soil carbon pools is fixed, but the C:N ratio of the litter pools will vary, which means the overall C:N ratio of the soil carbon will vary.**

*Line 21 - The introduction of a N cycle improves the representation of interannual variability of global net ecosystem exchange which was much too pronounced in the C cycle only versions of JULES (JULES-C)* – compared to what? Changed to : **The introduction of a N cycle improves the representation of interannual variability of global net ecosystem exchange which was more pronounced in the C cycle only versions of JULES (JULES-C) than shown in estimates from the Global Carbon Project.**

*Line 25 - The abstract doesn't mention the effect of layered soil biogeochemistry* – added: **JULES-CN$_{layer}$ improves the representation of soil biogeochemistry including turnover times in the northern high latitudes.**

*Line 23 - It also reduces the present-day CUE from a global mean value of 0.45 for JULES-C to 0.41 for JULES-CN and 0.40 for JULES-CN1ayer* – Is this less or more realistic? Isn't present day CUE more like 0.5? This paper: Intercomparison of Terrestrial Carbon Fluxes and Carbon Use Efficiency Simulated by CMIP5 Earth System Models | SpringerLink provides estimates of the CUE from other models and from MODIS data. Figure 10 in this paper shows the CUE is latitudinally dependent and varies between 0.2 and 0.7 with the majority of estimates between 0.4 and 0.6. Uncertainties in the observed CUE are considerable so the JULES estimates of CUE fall within this range of uncertainties. Text changed to: **It also reduces the present-day CUE from a global mean value of 0.45 for JULES-C to 0.41 for JULES-CN and 0.40 for JULES-CN$_{layer}$ all of which fall within the observational range.**

*Line 46 - In a changing climate Co2 drives an increase in the land C uptake and hence an increase in the gross primary productivity (GPP).* – Text changed to **Any increase in atmospheric CO$_{2}$ drives an increase in the land C uptake and hence an increase in the gross primary productivity (GPP).**

*Line 85 - The philosophy behind the developments described here is to produce a parsimonious model to capture the established first order emergent response of N addition on growth which translates into leaf area index (LAI) and biomass without the complex and uncertain impacts on leaf physiology* – unclear what is meant here – changed to: **The philosophy behind the developments described here is to produce a parsimonious model to capture the established first order emergent response of N addition on growth which translates into leaf area index (LAI) and biomass.**

*Line 90 - At the core of surface exchange in JULES is a coupled stomatal conductance photosynthesis scheme parameterised in terms of the maximum rate of Rubisco carboxylation, $V_{cmax}$ (mol CO$_2$ $m^{-2}$ $s^{-1}$). $V_{cmax}$ has a dependency on the leaf N concentration.-* photosynthesis-stomatal conductance coupling in general is not tied to vcmax. How photosynthesis is determined is related to (Ca-ci) gradient not vcmax – changed to - **At the core of surface exchange in JULES is a coupled stomatal conductance photosynthesis scheme with a dependency on the leaf N concentration.**

*Line 94 - Implicit within JULES, even in simulations excluding the N cycle is the parameterisation of plant tissue level N concentrations and associated allometry \citep{gmd-13-483-2020}* - please say that this is explained later – changed to: **Implicit within JULES, even in simulations excluding the N cycle is the parameterisation of plant tissue level N concentrations and associated allometry (discussed further in Section \ref{sec:allocup} and by \cite{gmd-13-483-2020})**

*Line 100 - Each exchange of C is associated with a corresponding flux of organic N.* – need a bit more elaboration- changed to - **At the ecosystem level, the C and N cycles are closely coupled with each flux of C associated with a corresponding flux of N linked through the N to N ratios.**

*In JULES nutrient limitation operates through two mechanisms; the available C for vegetation uptake is reduced, -* unclear C for veg uptake is in the atmosphere changed to - **Firstly, the vegetation cannot uptake as much C -- any C that the plants cannot uptake is denoted excess C.** Excess carbon is defined here and used subsequently throughout the document.

*Lie 103 - and the decomposition of litter C is slowed.* - please clarify, make the connections to N- changed to -**decomposition of litter C is slowed because there is insufficient N present**

*Line 105 - This is achieved by explicitly representing the demand for N within the vegetation and soil modules and then reducing plant net C gain to match available nutrients.* – please define gain – changed to - **reducing plant net primary productivity to match available nutrients**

*Line 103 - In the soil module an additional decomposition rate modifier is introduced that slows decomposition to match available nutrients.* – what is the process involved? – changed to - **In the soil module an additional decomposition rate modifier is introduced that slows respiration by microbes to match available nutrients.**

*Line 111 - In reality the excess C ($\Psi$)* – undefined what is this?- changed to - **In reality the C the plants are unable to use because of insufficient N ($\Psi$)**

*Line 112 – goes to non structural carbohydrates* – not strictly a loss term – changed to - **becomes to non structural carbohydrates, root exudates or biogenic volatile organic compounds**

*Line 120 - This is consistent with field experiments enhancing N fertilisation that find increases in growth but no corresponding change in photosynthetic capacity \citep{brix1969effects,wang2012impact}.* - what about Field and Mooney, 1986, Mcguire et al., 1995 - leaf level N is related to photosynthetic capacity. – **added both of these references.**

*Line 125 - Within the fully coupled Earth Systems Models used in the Coupled Climate Carbon Cycle Model Intercomparison Project (C4MIP) for quantifying C feedbacks only four models include a N cycle representation* – out of how many total? – added - **out of eleven models**

*Line 139 - Within JULES, C dynamics in soils and vegetation and dynamic vegetation are provided by Top-Down Representation of Interactive Foliage and Flora Including Dynamics (TRIFFID) \citep{cox2001}* – changed to - **Within JULES, C stocks and fluxes in and between the soils and vegetation along with competition between different vegetation types are modelled by the Top-Down Representation of Interactive Foliage and Flora Including Dynamics (TRIFFID) \citep{cox2001}**

*Line 166 - TRIFFID employs fixed allometry such that the split between leaf, root and stem C* – changed to - **TRIFFID employs fixed allometry such that the split of vegetation carbon between leaf, root and stem**

*Line 168 - Biomass density increases via growth and is reduced by litter production and competition.* - Changed to - **Biomass density increases via growth and is reduced by litter production and competition with other PFTs \citep{clarketal2011}.**

*Line 232 - The total vegetation N increases with canopy height and biomass (Figure ~\ref{fig:npool}).* – I don't understand why? – changed to - **Equations 1-8 show that the total vegetation N increases with canopy height and biomass (Figure ~\ref{fig:npool}).**

*Line 256 - $n_{lc,i}$ is the mean canopy N content* – units? – changed to – **concentration kg[N]/kg[C] Firstly, the vegetation cannot uptake as much C -- any C that the plants cannot uptake is denoted excess C. Secondly the decomposition of litter C is slowed because there is insufficient N present.**

Line 264 – changed to - **The formulation of the labile pool, in this configuration, means that around half of the N required for full leaf-out is taken from leaf retranslocation with a further quarter acquired during the dormant phase while the rest is acquired during the leaf-out period.**

Line 270 - changed to - **where $k_{n,i}$ is a constant representing the profile of N density and $d$ represents the fraction of canopy above the layer. Based on observed N profiles in the Amazon basin \citep{carswell2000photosynthetic}, a value of 0.78 for $k_{n,i}$ was found \citep{mercado2007improving}. Equation \ref{eq:kn} is independent of leaf area and therefore equates to a constant of proportionality relating PFT-specific top leaf N to the mean canopy N concentration.**

Line 292 - *What does subscript g mean?* – 'growth' – this is now defined.

Line 294 - *I think its incorrect to refer to this C as excess* – this has now been properly defined earlier in the document and should now make more sense.

Line 302 on – N-avail is a pool and Dt representing the timestep has been added to compare it to the fluxes.

Line 302 – why is phi not a flux – units changed to /s.

Line 311 – retranslocation now defined as kappa not lambda so as to not confuse with other lambda

Line 320 – explained above what excess C is.

Line 330 now reads: **In JULES-CN, on a PFT basis, the N available for plant uptake ($N_{avail,i}$ in $kg\,[N]\,m^{-2}s^{-1}$) is the inorganic soil N pool ($N_{in}$) split equitably between the PFTs assuming there is no differential ability between PFTs to acquire N and the whole pool is available for uptake during the model timestep**. This should resolve the confusion between pools and fluxes for lines 319 to lines 331.

Line 355 now reads: - **The C and N allocated to spreading allow the vegetation to expand onto bare ground. Where space is limiting the PFTs compete for space.**

Line 366 – *aggregated tile* replaced by **grid box**

Equation 22 – Pi and Psi are defined with reference to other Equations in the document.

Line 380 – *pool being turned over replaced* by **the relative amount of stem, leaves and roots being turned over**

Line 383 - *The soil biogeochemistry in JULES-CN operates on aggregated tiles* – replaced by **- The soil biogeochemistry in JULES-CN operates at a grid box level**

Line 384 - *follows the Roth-C soil C model \citep{jenkinson1990,cj1999} used in JULES-C on the TRIFFID timestep, with the addition of a prognostic soil N model* – changed to - **The soil biogeochemistry in JULES-CN operates at a grid box level and is an extension of the Roth-C soil C model \citep{jenkinson1990,cj1999}. Roth-C is used in JULES-C - here we describe the addition of a prognostic soil N model.**

Line 450 – FN is also applied to immobilisation and mineralisation. This has been added: **The nitrogen limited mineralisation and immobilisation of the $DPM$ and $RPM$ pools (Equations \ref{eq:ipot} and \ref{eq:mpot}) are now effectively a function of $R_{p}$.**

Equation 40 - |Delta_t has been added which represents the timestep length.

Line 467 – have now shown that mineralisation and immobilisation can be reduced by N limitation.

Line 490 – tau has been changed to xi.

Line 504 – added - **Do(m2s–1) varies spatially depending on the freeze/thaw state of the soil**

Line 520 – f_lit is independent of PFT – text changed to **- Here $f_{lit}(z)$ is independent of the PFt type and hence the root distribution**

Line 513 -added - **and z is the mid-point of each layer**

Equation 52 – this is an additional gas loss term so the total gas loss is equation 42 plus equation 52. This is stated in line 545.

Equation 58 and 59 – the rate change equation (59) is an additional change that happens after the plant uptake, so it is an additional modification to equation 58 after the plant uptake has occurred which moves the accessible N away from equilibrium.

Equation 60 – this is now discrete.

Line 625 - *The total leaching is then the sum of all N that leaves the soil* -changed to - **The total leaching is then the sum of all N that leaves the soil both laterally from each layer or from the bottom of the soil profile**

Line 631 – added - **which include time varying climate, CO$_2$, and N deposition but pre-industrial land use.**

Line 669 - *Then we show spatial distributions and time series of the N stocks and fluxes* – changed to - **Then we show spatial distributions and time series of the N stocks and fluxes**

Line 678 – *CCI observations* – changed to - **The Climate Change Initiative (CCI) land cover observations**

Line 675 to 681 – The purpose if this paper is to document the nitrogen scheme. We have specifically not gone into too much detail of the vegetation distribution. This is because the configuration of JULES adopted here has been superseded by JULES-ES used in UKESM1. This includes additional improvements including a revised competition scheme and new PFTs. The text has been changed to - **In general, the models all tend to over-estimate the shrubs and underestimate the grass. However, \cite{ukesm1} shows that once the additional PFTs are included the model does a good job of representing the vegetation distribution.**

Figure 4- the litter fluxes have been added

Line 708 on – Figure 5 (now figure 6) shows the latitude vertically on the plot as might be expected on a spatial plot. I think this is a good way to show the zonal means rather than having the latitude horizontally.

Line 728 – The considerable uncertainties in the estimates of the CUE mean that it is hard to draw the conclusion that reducing NPP and not GPP may or may not be a good strategy in response to N limitation. The following is added: **However, considerable uncertainties remain in these estimates.**

Line 790 – the colorscales on the figure have been modified to make things clearer. We have left analysis using response ratio and NPPpot/NPPacheived because this is the way LeBauer and Treseder (2008) report their results.

Line 799 – Figure 9c implies no N limitation on 1860-1900 can you comment if this is realistic? - Figure 9c is an anomaly from the mean pre-industrial state which ahs a global mean value of 1.2-1.3. So there is nitrogen limitation in the pre-industrial state.

Line 825 – I agree, this is a confusing message, I have removed the lines.

Line 884 – this now reads: **Allowing for flexible stoichometry may lead to a lower litter quality but a comparable amount of litter. This reduction in litter quality will strengthen the soil turnover response possibly leading to an overall increase in soil organic matter.**